# Investigating Language-Specific Calibration For Pruning Multilingual Large Language Models

## Abstract

Recent advances in large language model (LLM) pruning have shown state-of-the-art (SotA) compression results in post-training and retraining-free settings while maintaining high predictive performance. However, previous research mainly considered calibrating based on English text, despite the multilingual nature of modern LLMs and their frequent use in non-English languages. In this paper, we set out to investigate calibrating the pruning of multilingual language models for monolingual applications. We present the first comprehensive empirical study, comparing different calibration languages for pruning multilingual models across diverse languages, tasks, models, and SotA pruning techniques. Our results offer practical suggestions, for example, calibrating in the target language can efficiently retain the language modeling capability but does not necessarily benefit downstream tasks. Through further analysis of latent subspaces, pruning masks, and individual neurons within pruned models, we find that while pruning generally preserves strong language-specific features, it may fail to retain language-specific neuron activation patterns and subtle, language-agnostic features associated with knowledge and reasoning that are needed for complex tasks.

## 1 Introduction

State-of-the-art language models often rely on over-parameterization with millions or billions of parameters, resulting in significant memory and computational demands (Zhang et al., 2017; Allen-Zhu et al., 2019; Xu & Du, 2023). Pruning is a model compression technique that removes unimportant weights to reduce memory footprint and inference computation (Gholami et al., 2022; Hoefler et al., 2021; Kuzmin et al., 2023). Recent pruning methods for language models, such as SparseGPT (Frantar & Alistarh, 2023) and Wanda (Sun et al., 2024), demonstrated SotA performance in a post-training and retraining-free setting (Zhu et al., 2023). The pruning process involves passing a small number of examples, i.e. calibration data, to the model to determine the importance of weights for subsequent pruning (Kuzmin et al., 2022; Frantar & Alistarh, 2023; Kuzmin et al., 2023).

Notably, existing studies typically use calibration data in English and evaluate the post-pruning performance in English. On the other hand, most SotA LLMs are multilingual and frequently employed for tasks in non-English languages (Touvron et al., 2023; Achiam et al., 2023; Jiang et al., 2023). **It remains unclear how to calibrate pruning to optimize the post-pruning performance of multilingual LLMs on tasks in non-English languages.** In other words, the current literature provides little insight into efficient calibration strategies for multilingual LLMs targeting specific non-English languages. For example, if we aim to prune a multilingual LLM and use the pruned LLM for tasks in German, should we calibrate the compression process with text in English, German, or both?

This research, as the first, presents an extensive study investigating the impact of calibration data language on pruning LLMs for language-specific tasks. Further, we performed a set of analyses at the latent subspace level, the matrix level, and the neuron level, within the pruned models, to investigate the underlying causes for our observations. We summarize our key findings as follows:

- Calibrating on the target language consistently yields the lowest perplexity (Section 4.1, but will not guarantee optimal performance on downstream tasks (Section 4.2).

- Pruning generally impairs the language-agnostic features of multilingual models, such as reasoning capability (Section 4.2) and the storage and retrieval of knowledge (Section 4.3), which are essential for downstream tasks (Section 5).
- Pruning generally affects language-agnostic features, which is potentially associated with reasoning and knowledge storage, more than it impacts language-specific features (Section 5.1); Pruning struggles to consistently identify essential neurons in the attention output and FFN down projection, potentially responsible for reasoning (Section 5.2); pruning also struggles to retain language-specific neurons with low activation probability (Section 5.3).

## 2 RELATED WORK

### 2.1 MULTILINGUAL LANGUAGE MODELS

Most SotA LMs, such as Llama-3 (Meta AI, 2024) and Phi 3(Abdin et al., 2024), are trained on multilingual data, enabling them to understand and generate text in multiple languages (Huang et al., 2023; Holmström et al., 2023; Xu et al., 2024; Meta AI, 2024). Although these multilingual LMs follow the general training paradigm of training monolingual LMs, they are often found to behave differently or show unique features from monolingual models (Xu et al., 2024). For example, Deng et al. (2024) reveals that multilingual LMs are prone to generate unsafe outputs on malicious instructions in non-English languages, i.e. multilingual jailbreak. Wang et al. (2023) also identify that all LLMs tested in their study produce significantly more unsafe responses for non-English queries than English ones. Furthermore, previous work on model explanation finds that multilingual models are different from counterpart monolingual models in terms of their internal processes, i.e., different importance distributions on the same input (Jørgensen et al., 2022; Zhao & Aletras, 2024). A general conclusion drawn from these studies is that findings from monolingual settings, particularly those in English, are unlikely to hold in multilingual settings involving non-English languages.

### 2.2 CALIBRATION OF POST-TRAINING PRUNING

Unlike sparse training (Yuan et al., 2021; Hoang & Liu, 2023; Zhang et al., 2023) or pruning-aware training (Liu et al., 2021), which require iterative training to achieve sparsity, post-training pruning (Frantar & Alistarh, 2023; Sun et al., 2024) does not require training but eliminates redundant weights based on the importance of weights that are calculated with calibration data. The retraining-free feature of post-training pruning makes it a more efficient approach for LLMs.

Prior research has primarily focused on calculating the importance of weight to optimize the performance of pruned models, examining importance metrics and importance ranking ranges (Frantar & Alistarh, 2023; Sun et al., 2024; Zhang et al., 2024). Two studies have investigated the impact of calibration data, looking at the quantity of calibration data (Zhang et al., 2024) and the sources of the data (Williams & Aletras, 2023). These studies have been limited to English. The effects of pruning on the multilingual capabilities of language models, and the impact of the language of calibration data on performance in other target languages remain unknown.

## 3 METHODOLOGY

To investigate the impact of the language of calibration data on the pruned model, we compare the performance (pruning error, Signal-to-noise ratio, perplexity, and downstream tasks) in a range of languages between pruned models and their counterpart full-sized models, where different calibration data are applied.

**Models** Our experiments use two SotA LLM families: Llama-3 (Meta, 2024), the open-source SotA at the time of writing, and Aya-23 (Aryabumi et al., 2024), renowned for its multilingual pre-training.As our evaluation focuses on instructed generation tasks, we employ the instruction-tuned versions of both families (Chrysostomou et al., 2023). Consequently, our experimental setup includes Llama-3-instruction models in 8B and 70B parameter sizes, alongside Aya-instruction models in 8B and 35B parameter sizes.

**Languages** We include seven languages in our study: Arabic (AR), German (DE), English (EN), Spanish (ES), Russian (RU), Swahili (SW), and Chinese (ZH). This selection spans six language

families, four writing systems, and encompasses both high and mid-resource languages. We include Swahili as an outlier calibration language, as neither Llama-3 or Aya-23 includes it in their pre-training corpora. A summary of languages used in our paper is given in Table 5 in Appendix B.

**Pruning and Calibration** We construct calibration sets in seven different languages from mC4 (Raffel et al., 2019). Specifically, following Frantar & Alistarh (2023); Sun et al. (2024), we randomly sample 128 sequences of 8192 tokens for each language, from the mC4 split for that language.[1]

We focus on two SotA post-training *pruning methods*, Wanda (Sun et al., 2024) and SparseGPT (Frantar & Alistarh, 2023). Unless stated otherwise, we prune for 50% unstructured sparsity to impose fewer restrictions while keeping all other hyperparameters as in the original paper.

**Evaluation Downstream Tasks** We compare the performance of the pruned models calibrated on different languages using the *perplexity* on a subset of the mC4 validation set, and a selection of downstream tasks in different target languages: MMLU (Hendrycks et al., 2021), MKQA (Longpre et al., 2020), and Belebele (Bandarkar et al., 2023). To better isolate the impact of the calibration language, we also chose mARC and mHellaSwag (Dac Lai et al., 2023) to compare with their original versions, ARC (Clark et al., 2018) and HellaSwag (Zellers et al., 2019) in English for their low sensitivity to the choice of calibration samples (Williams & Aletras, 2023). These tasks primarily assess commonsense reasoning, reading comprehension, and question answering using multiple choice questions. We present further details of each task in Appendix C. Unless stated otherwise, we evaluate in a zero-shot setting.

**Implementation Details** We adopt the code from Sun et al. (2024) for implementing model pruning. We use EleutherAI Evaluation Harness (Gao et al., 2024) for a robust and reproducible evaluation. We use Huggingface (Wolf et al., 2020) for loading datasets and models. All experiments are conducted with at most two NVIDIA A100 GPUs.

# 4 RESULTS

## 4.1 PRUNING RESULTS

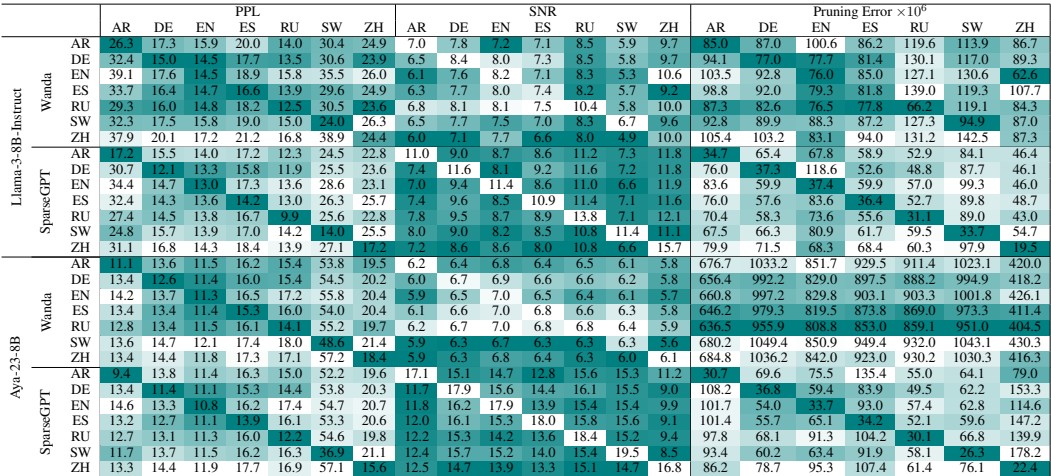

| | | | PPL | | | | | | | SNR | | | | | | | Pruning Error ×10⁶ | | | | | | |
|---|---|---|---|---|---|---|---|---|---|---|---|---|---|---|---|---|---|---|---|---|---|---|---|
| | | | AR | DE | EN | ES | RU | SW | ZH | AR | DE | EN | ES | RU | SW | ZH | AR | DE | EN | ES | RU | SW | ZH |
| Llama-3-8B-Instruct | Wanda | AR | 26.3 | 17.3 | 15.9 | 20.0 | 14.0 | 30.4 | 24.9 | 7.0 | 7.8 | 7.2 | 7.1 | 8.5 | 5.9 | 9.7 | 85.0 | 87.0 | 100.6 | 86.2 | 119.6 | 113.9 | 86.7 |
| | | DE | 32.4 | 15.0 | 14.5 | 17.7 | 13.5 | 30.6 | 23.9 | 6.5 | 8.4 | 8.0 | 7.3 | 8.5 | 5.8 | 9.7 | 94.1 | 77.0 | 77.7 | 81.4 | 130.1 | 117.0 | 89.3 |
| | | EN | 39.1 | 17.6 | 14.5 | 18.9 | 15.8 | 35.5 | 26.0 | 6.1 | 7.6 | 8.2 | 7.1 | 8.3 | 5.3 | 10.6 | 103.5 | 92.8 | 76.0 | 85.0 | 127.1 | 130.6 | 62.6 |
| | | ES | 33.7 | 16.4 | 14.7 | 16.6 | 13.9 | 29.6 | 24.9 | 6.3 | 7.7 | 8.0 | 7.4 | 8.2 | 5.7 | 9.2 | 98.8 | 92.0 | 79.3 | 81.8 | 139.0 | 119.3 | 107.7 |
| | | RU | 29.3 | 16.0 | 14.8 | 18.2 | 12.5 | 30.5 | 23.6 | 6.8 | 8.1 | 8.1 | 7.5 | 10.4 | 5.8 | 10.0 | 87.3 | 82.6 | 75.5 | 77.8 | 66.2 | 119.1 | 84.3 |
| | | SW | 32.3 | 17.5 | 15.8 | 19.0 | 15.0 | 24.0 | 26.3 | 6.5 | 7.7 | 7.5 | 7.0 | 8.3 | 6.7 | 9.6 | 92.8 | 89.9 | 88.3 | 87.2 | 127.3 | 94.9 | 87.0 |
| | | ZH | 37.9 | 20.1 | 17.2 | 21.2 | 16.8 | 38.9 | 24.4 | 6.0 | 7.1 | 7.7 | 6.6 | 8.0 | 4.9 | 10.0 | 105.4 | 103.2 | 83.1 | 94.0 | 131.2 | 142.5 | 87.3 |
| | SparseGPT | AR | 17.2 | 15.5 | 14.0 | 17.2 | 12.3 | 24.5 | 22.8 | 11.0 | 9.0 | 8.7 | 8.6 | 11.2 | 7.3 | 11.8 | 34.7 | 65.4 | 67.8 | 58.9 | 52.9 | 84.1 | 46.4 |
| | | DE | 30.7 | 12.1 | 13.3 | 15.8 | 11.9 | 25.5 | 23.6 | 7.4 | 11.6 | 8.1 | 9.2 | 11.6 | 7.2 | 11.8 | 76.0 | 37.3 | 118.6 | 52.6 | 48.8 | 87.7 | 46.1 |
| | | EN | 34.4 | 14.7 | 13.0 | 17.3 | 13.6 | 28.6 | 23.1 | 7.0 | 9.4 | 11.4 | 8.6 | 11.0 | 6.6 | 11.9 | 83.6 | 59.9 | 37.4 | 59.9 | 57.0 | 99.3 | 46.0 |
| | | ES | 32.4 | 14.3 | 13.6 | 14.2 | 13.0 | 26.3 | 25.7 | 7.4 | 9.6 | 8.5 | 10.9 | 11.4 | 7.1 | 11.6 | 76.0 | 57.6 | 83.6 | 36.4 | 52.7 | 89.8 | 48.7 |
| | | RU | 27.4 | 14.5 | 13.8 | 16.7 | 9.9 | 25.6 | 22.8 | 7.8 | 9.5 | 8.7 | 8.9 | 13.8 | 7.1 | 12.1 | 70.4 | 58.3 | 73.6 | 55.6 | 31.1 | 89.0 | 43.0 |
| | | SW | 24.8 | 15.7 | 13.9 | 17.0 | 14.2 | 14.0 | 25.5 | 8.0 | 9.0 | 8.2 | 8.5 | 10.8 | 11.4 | 11.1 | 67.5 | 66.3 | 80.9 | 61.7 | 59.5 | 33.7 | 54.7 |
| | | ZH | 31.1 | 16.8 | 14.3 | 18.4 | 13.9 | 27.1 | 17.2 | 7.2 | 8.6 | 8.6 | 8.0 | 10.8 | 6.6 | 15.7 | 79.9 | 71.5 | 68.3 | 68.4 | 60.3 | 97.9 | 19.5 |
| Aya-23-8B | Wanda | AR | 11.1 | 13.6 | 11.6 | 16.2 | 15.4 | 53.8 | 19.5 | 6.2 | 6.4 | 6.8 | 6.4 | 6.5 | 6.1 | 5.8 | 676.7 | 1033.2 | 851.7 | 929.5 | 911.4 | 1023.1 | 420.0 |
| | | DE | 13.4 | 12.6 | 11.4 | 16.0 | 15.4 | 54.5 | 20.2 | 6.0 | 6.7 | 6.9 | 6.6 | 6.6 | 6.2 | 5.8 | 656.4 | 992.2 | 829.0 | 897.5 | 888.2 | 994.9 | 418.2 |
| | | EN | 14.2 | 13.7 | 11.3 | 16.5 | 17.2 | 55.8 | 20.4 | 5.9 | 6.5 | 7.0 | 6.5 | 6.4 | 6.1 | 5.7 | 660.8 | 997.2 | 829.8 | 903.1 | 903.3 | 1001.8 | 426.1 |
| | | ES | 13.4 | 13.4 | 11.4 | 15.3 | 16.0 | 54.0 | 20.4 | 6.1 | 6.6 | 7.0 | 6.8 | 6.6 | 6.3 | 5.8 | 646.2 | 979.3 | 819.5 | 873.8 | 869.0 | 973.3 | 411.4 |
| | | RU | 12.8 | 13.4 | 11.5 | 16.1 | 14.1 | 55.2 | 19.7 | 6.2 | 6.7 | 7.0 | 6.8 | 6.8 | 6.4 | 5.9 | 636.5 | 955.9 | 808.8 | 853.0 | 859.1 | 951.0 | 404.5 |
| | | SW | 13.6 | 14.7 | 12.1 | 17.4 | 18.0 | 48.6 | 21.4 | 5.9 | 6.3 | 6.7 | 6.3 | 6.3 | 6.3 | 5.6 | 680.2 | 1049.4 | 850.9 | 949.4 | 932.0 | 1043.1 | 430.3 |
| | | ZH | 13.4 | 14.4 | 11.8 | 17.3 | 17.1 | 57.2 | 18.4 | 5.9 | 6.4 | 6.3 | 6.0 | 6.3 | 6.0 | 6.1 | 684.8 | 1036.2 | 842.0 | 923.0 | 930.2 | 1030.3 | 416.3 |
| | SparseGPT | AR | 9.4 | 13.8 | 11.4 | 16.3 | 15.0 | 52.2 | 19.6 | 17.1 | 15.1 | 14.7 | 12.8 | 15.6 | 15.3 | 11.2 | 30.7 | 69.6 | 75.5 | 135.4 | 55.0 | 64.1 | 79.0 |
| | | DE | 13.4 | 11.4 | 11.1 | 15.3 | 14.4 | 53.8 | 20.3 | 11.7 | 17.9 | 15.6 | 14.4 | 16.1 | 15.5 | 9.0 | 108.2 | 36.8 | 59.4 | 83.9 | 49.5 | 62.2 | 153.3 |
| | | EN | 14.6 | 13.3 | 10.8 | 16.2 | 17.4 | 54.7 | 20.7 | 11.8 | 16.2 | 17.9 | 13.9 | 15.4 | 15.4 | 9.9 | 101.7 | 54.0 | 33.7 | 93.0 | 57.4 | 62.8 | 114.6 |
| | | ES | 13.2 | 12.7 | 11.1 | 13.9 | 16.1 | 53.3 | 20.6 | 12.0 | 16.1 | 15.3 | 18.0 | 15.8 | 15.6 | 9.1 | 101.4 | 55.7 | 65.1 | 34.2 | 52.1 | 59.6 | 147.2 |
| | | RU | 12.7 | 13.1 | 11.3 | 16.0 | 12.2 | 54.6 | 19.8 | 12.2 | 15.3 | 14.3 | 13.4 | 18.4 | 15.2 | 9.4 | 97.8 | 68.1 | 71.3 | 104.2 | 30.1 | 66.8 | 139.9 |
| | | SW | 11.7 | 13.7 | 11.5 | 16.2 | 16.3 | 36.9 | 21.1 | 12.4 | 15.7 | 15.2 | 14.0 | 15.4 | 19.5 | 8.5 | 93.4 | 60.2 | 63.4 | 91.9 | 58.1 | 26.3 | 178.2 |
| | | ZH | 13.3 | 14.4 | 11.9 | 17.7 | 16.9 | 57.1 | 15.6 | 12.5 | 14.7 | 13.9 | 13.3 | 15.1 | 14.7 | 16.8 | 86.2 | 78.7 | 95.3 | 107.4 | 61.4 | 76.1 | 22.4 |

Table 1: Language-specific perplexity (PPL), Signal-to-noise ratio (SNR) and pruning error of hidden states for 50% pruned Llama-3 8B, averaged over three pruning runs. The leftmost columns show the model, the pruning technique, and the calibration language. For PPL and pruning errors, the smaller the value (the darker), the better; while for SNR, the greater the value (the lighter), the better. This evaluation uses 128 inputs sampled from the mC4 validation set (1048576 tokens).

---

[1]In the original implementation of Wanda and SparseGPT, each sample contains 2,048 tokens, matching the maximum context length of Llama 1. We increase the length to 8,192 tokens to match the maximum context of Llama-3. This is to avoid potential context length-dependence pruning behavior, although Sun et al. (2024) reports a negligible return from increased calibration set sizes. For mixing different calibration languages, we mix in equal shares and keep the budget of 128 samples.

Table 1 presents the perplexity (PPL), Signal-to-noise ratio (SNR) and pruning error of hidden states for models pruned to 50% sparsity and calibrated on different languages. SNR and pruning error estimate the pruning accuracy (pruning performance); ideal pruning is able to preserve activations identical to the full model. The formal expression and detailed explanation of SNR and pruning error are presented in the Appendix D.1. Perplexity reflects the general language modeling capability. The column headers denote the evaluation languages, while the row headers indicate the calibration languages.

Overall, our findings reveal that no single calibration language consistently outperforms others across perplexity, pruning error, or SNR metrics. The optimal calibration language varies depending on the target evaluation language. Notably, *calibrating on the target language itself generally yields the best pruning performance and the lowest perplexity*, as evidenced by the diagonal pattern in Table 1. There are a few exceptions. For instance, when evaluating Llama-3 pruned with Wanda on Chinese, Russian calibration performs optimally on perplexity (23.6), slightly outperforming Chinese calibration (24.4).

## 4.2 Downstream Task Performance

*How to select the calibration language to optimize performance in downstream tasks?*

Table 2 shows the models' performance on downstream tasks. The column headers denote the evaluation languages, while the row headers indicate the calibration languages. Given a downstream task in a specific language, we analyze the results column-wise; the lighter the color in a column, the better the performance on the specified downstream task.

First, for both pruning methods, the calibration language affects downstream task performance. This impact is particularly pronounced when pruning Llama-3 8B model with SparseGPT or Aya-23 models with either Wanda or SparseGPT, as evidenced by the greater color difference in the table, compared to pruning Llama-3 8B with Wanda. For example, when evaluating Llama-3 8B model on Belebele in English, pruning with Wanda results in performance ranging from 58.5 with Spanish calibration to 65.0 with English calibration. In contrast, when pruning Llama-3 8B with SparseGPT the performance varies from 50.0 with Chinese calibration to 71.4 with English calibration.

Unlike the perplexity evaluation in Table 1, calibration with the target language does not reliably result in good performance. For instance, on MMLU, the pruned Llama-3 8B model mainly achieves higher accuracy on evaluation languages other than on its calibration languages. Overall, there are 50 evaluation groups, 25 columns for Wanda and 25 for SparseGPT, in 26 comparison cases. Calibration with the target language yields the best performance, 8 for Wanda and 18 for SparseGPT (e.g.

| Model | Method | Calib | ARC[acc] AR | DE | EN-E | EN-C | ES | RU | ZH | Belebele[acc] AR | DE | EN | ES | RU | SW | ZH | MMLU[acc] AR | DE | EN | ES | RU | ZH | HellaSwag[acc] AR | DE | EN | ES | RU |
|---|---|---|---|---|---|---|---|---|---|---|---|---|---|---|---|---|---|---|---|---|---|---|---|---|---|---|---|
| Llama-3-8B-Instruct | | - | 31.7 | 36.6 | 75.9 | 48.5 | 37.9 | 36.9 | 34.8 | 55.3 | 69.4 | 58.2 | 66.3 | 62.9 | 47.3 | 44.8 | 27.1 | 38.2 | 58.6 | 43.0 | 30.3 | 25.9 | 36.2 | 43.2 | 53.3 | 46.1 | 41.7 |
| | Wanda | AR | 22.9 | 26.7 | 68.0 | 40.1 | 30.6 | 28.7 | 29.0 | 40.6 | 51.5 | 64.1 | 48.3 | 50.9 | 28.7 | 54.9 | 26.6 | 30.8 | 42.1 | 30.0 | 27.0 | 29.1 | 31.3 | 36.2 | 46.7 | 39.4 | 35.5 |
| | | DE | 22.6 | 27.9 | 68.9 | 40.4 | 30.3 | 28.3 | 27.8 | 42.8 | 52.6 | 64.5 | 50.3 | 53.6 | 29.3 | 54.2 | 25.7 | 29.8 | 41.6 | 30.4 | 27.7 | 27.3 | 30.8 | 37.0 | 48.1 | 40.0 | 36.0 |
| | | EN | 21.9 | 27.8 | 69.3 | 40.2 | 30.5 | 27.9 | 27.7 | 40.7 | 49.7 | 65.0 | 49.4 | 52.3 | 30.1 | 52.2 | 25.9 | 29.5 | 43.1 | 31.2 | 26.9 | 28.5 | 30.2 | 36.1 | 48.0 | 39.9 | 35.3 |
| | | ES | 21.7 | 27.0 | 68.1 | 41.4 | 31.4 | 29.3 | 28.6 | 38.1 | 45.4 | 58.5 | 45.3 | 47.1 | 28.8 | 47.7 | 25.8 | 29.9 | 37.9 | 29.2 | 28.3 | 28.8 | 30.8 | 36.4 | 47.5 | 40.1 | 35.8 |
| | | RU | 23.0 | 28.3 | 69.2 | 40.3 | 30.9 | 28.9 | 28.3 | 40.7 | 51.0 | 62.2 | 47.0 | 52.0 | 30.3 | 52.9 | 26.6 | 30.8 | 41.7 | 32.8 | 28.9 | 30.8 | 31.1 | 36.5 | 47.8 | 40.0 | 36.4 |
| | | SW | 21.7 | 26.4 | 64.7 | 38.0 | 28.6 | 26.7 | 26.2 | 34.8 | 44.1 | 60.3 | 39.5 | 45.4 | 29.2 | 49.7 | 24.7 | 26.5 | 41.0 | 26.7 | 25.1 | 26.1 | 30.5 | 35.2 | 46.0 | 38.9 | 34.8 |
| | | ZH | 20.4 | 24.0 | 65.9 | 37.9 | 27.6 | 25.7 | 27.0 | 28.9 | 40.2 | 62.2 | 41.0 | 40.3 | 23.5 | 41.2 | 23.2 | 24.0 | 40.5 | 26.3 | 23.4 | 24.1 | 29.5 | 34.5 | 44.7 | 38.0 | 34.1 |
| | SparseGPT | AR | 26.6 | 27.6 | 64.8 | 39.5 | 31.9 | 30.1 | 29.3 | 41.0 | 44.4 | 50.9 | 41.3 | 43.2 | 25.6 | 41.8 | 28.6 | 31.8 | 42.6 | 33.0 | 29.8 | 29.9 | 33.1 | 37.3 | 48.1 | 40.6 | 36.9 |
| | | DE | 23.6 | 30.5 | 68.4 | 40.5 | 31.7 | 31.1 | 29.4 | 40.4 | 57.2 | 67.1 | 53.3 | 52.4 | 30.3 | 50.3 | 26.8 | 36.7 | 48.6 | 36.5 | 31.0 | 30.7 | 31.2 | 39.1 | 49.1 | 41.3 | 37.5 |
| | | EN | 23.0 | 28.7 | 67.9 | 41.4 | 31.9 | 29.8 | 30.1 | 43.1 | 57.9 | 71.4 | 55.5 | 55.9 | 31.0 | 57.0 | 28.4 | 35.9 | 49.1 | 36.7 | 30.7 | 33.7 | 30.7 | 37.9 | 49.6 | 41.2 | 36.7 |
| | | ES | 24.0 | 28.5 | 66.6 | 40.6 | 34.9 | 30.4 | 28.3 | 45.1 | 56.6 | 68.3 | 58.7 | 56.8 | 31.9 | 53.1 | 29.7 | 37.8 | 47.8 | 40.9 | 34.3 | 33.8 | 31.1 | 37.9 | 48.7 | 42.3 | 37.1 |
| | | RU | 24.0 | 28.9 | 67.6 | 40.9 | 32.6 | 32.1 | 30.1 | 34.4 | 48.7 | 62.1 | 45.5 | 50.8 | 27.7 | 46.3 | 30.0 | 38.1 | 50.2 | 38.0 | 34.7 | 34.7 | 31.7 | 38.1 | 49.2 | 41.2 | 38.7 |
| | | SW | 23.9 | 28.2 | 66.7 | 38.4 | 31.3 | 29.1 | 27.9 | 40.8 | 47.3 | 57.9 | 45.5 | 49.5 | 36.4 | 46.1 | 26.4 | 30.5 | 41.6 | 30.0 | 26.8 | 26.9 | 31.4 | 37.1 | 48.4 | 40.5 | 36.3 |
| | | ZH | 23.8 | 26.7 | 58.9 | 36.8 | 29.8 | 27.4 | 30.1 | 34.9 | 38.8 | 50.0 | 36.4 | 37.6 | 28.4 | 40.5 | 26.2 | 30.2 | 43.8 | 28.6 | 27.4 | 29.4 | 30.8 | 36.5 | 47.8 | 39.4 | 36.1 |
| Aya-23-8B | | - | 37.0 | 42.4 | 75.1 | 45.1 | 43.9 | 38.6 | 35.9 | 66.2 | 75.3 | 77.8 | 72.4 | 71.6 | 32.3 | 73.3 | 44.5 | 48.7 | 54.5 | 50.5 | 47.5 | 46.9 | 41.0 | 45.7 | 58.3 | 49.4 | 43.5 |
| | Wanda | AR | 31.1 | 33.6 | 70.7 | 40.8 | 35.5 | 34.4 | 32.3 | 56.6 | 66.7 | 75.7 | 39.1 | 64.8 | 31.4 | 64.2 | 37.7 | 42.8 | 47.6 | 43.5 | 41.3 | 40.3 | 37.5 | 41.9 | 51.5 | 45.1 | 39.6 |
| | | DE | 29.4 | 33.6 | 70.3 | 41.6 | 35.9 | 34.2 | 31.3 | 58.3 | 69.0 | 75.7 | 62.4 | 65.0 | 31.9 | 66.0 | 37.6 | 43.0 | 48.1 | 44.4 | 41.0 | 40.7 | 37.3 | 42.0 | 51.5 | 44.9 | 39.8 |
| | | EN | 28.9 | 33.8 | 71.3 | 41.9 | 35.9 | 34.5 | 32.0 | 56.7 | 69.7 | 76.9 | 65.3 | 67.0 | 29.9 | 67.6 | 36.9 | 42.5 | 48.6 | 44.7 | 41.1 | 40.8 | 37.1 | 41.9 | 51.5 | 44.8 | 39.3 |
| | | ES | 30.5 | 33.0 | 70.4 | 41.7 | 35.8 | 34.7 | 31.7 | 58.7 | 69.0 | 75.6 | 62.5 | 66.0 | 31.0 | 65.7 | 37.2 | 43.0 | 47.8 | 44.1 | 41.4 | 40.7 | 37.2 | 42.1 | 51.3 | 45.5 | 40.1 |
| | | RU | 30.9 | 34.4 | 70.9 | 42.4 | 36.3 | 33.8 | 32.0 | 53.0 | 68.9 | 74.3 | 58.7 | 65.5 | 30.6 | 63.4 | 37.5 | 42.9 | 47.9 | 43.6 | 41.3 | 40.2 | 37.5 | 42.0 | 51.7 | 45.6 | 40.1 |
| | | SW | 27.7 | 32.4 | 69.5 | 39.1 | 33.0 | 31.6 | 30.7 | 59.4 | 65.8 | 75.4 | 63.1 | 63.6 | 32.2 | 63.4 | 36.5 | 41.6 | 45.4 | 42.6 | 40.4 | 38.9 | 36.9 | 40.9 | 49.9 | 43.8 | 38.8 |
| | | ZH | 28.6 | 33.3 | 70.0 | 39.3 | 35.1 | 32.5 | 30.8 | 54.1 | 64.0 | 73.4 | 60.7 | 63.1 | 31.6 | 63.7 | 36.8 | 41.3 | 47.1 | 43.2 | 40.4 | 40.6 | 36.8 | 41.0 | 50.1 | 43.9 | 38.9 |
| | SparseGPT | AR | 31.5 | 33.6 | 68.6 | 40.4 | 35.7 | 32.2 | 31.2 | 58.8 | 70.3 | 77.1 | 65.1 | 65.6 | 30.9 | 64.3 | 39.0 | 41.9 | 46.5 | 44.2 | 41.3 | 40.6 | 38.4 | 42.0 | 51.4 | 45.1 | 39.8 |
| | | DE | 28.7 | 35.6 | 69.5 | 41.1 | 34.4 | 32.5 | 30.9 | 57.6 | 65.9 | 73.7 | 62.7 | 62.6 | 30.3 | 65.3 | 37.3 | 43.0 | 46.6 | 44.3 | 41.3 | 39.8 | 37.3 | 42.9 | 51.9 | 45.2 | 40.1 |
| | | EN | 29.6 | 33.7 | 70.8 | 41.8 | 35.6 | 32.0 | 31.0 | 48.0 | 66.2 | 72.1 | 59.9 | 61.6 | 28.8 | 63.9 | 36.6 | 42.6 | 45.8 | 44.0 | 40.9 | 40.0 | 37.3 | 42.2 | 52.7 | 45.3 | 39.9 |
| | | ES | 28.7 | 33.9 | 69.6 | 40.4 | 36.3 | 33.5 | 30.9 | 53.1 | 65.3 | 76.4 | 61.3 | 62.7 | 30.0 | 63.4 | 36.9 | 42.2 | 47.0 | 44.1 | 41.8 | 40.9 | 37.5 | 42.7 | 52.3 | 46.5 | 40.8 |
| | | RU | 29.7 | 34.2 | 69.6 | 41.3 | 35.4 | 33.7 | 31.0 | 53.2 | 64.4 | 74.5 | 58.9 | 62.5 | 27.4 | 65.5 | 37.6 | 42.6 | 44.1 | 41.9 | 41.1 | 40.9 | 37.4 | 42.5 | 51.3 | 45.3 | 40.9 |
| | | SW | 29.4 | 32.8 | 68.2 | 39.8 | 33.0 | 31.0 | 30.6 | 53.0 | 65.7 | 75.7 | 63.4 | 60.6 | 31.4 | 62.9 | 36.6 | 42.2 | 44.7 | 43.1 | 40.4 | 39.2 | 37.3 | 41.2 | 50.9 | 44.8 | 39.2 |
| | | ZH | 29.3 | 32.2 | 67.5 | 39.2 | 34.4 | 30.5 | 31.7 | 54.4 | 65.3 | 72.6 | 61.7 | 61.3 | 29.8 | 66.7 | 36.9 | 40.0 | 44.7 | 42.2 | 40.2 | 40.6 | 36.5 | 40.6 | 49.2 | 43.4 | 38.8 |

Table 2: Evaluation task performance with 50% unstructured sparsity averaged over three pruning runs. The leftmost columns show the model, the pruning technique, and the calibration language used for pruning. A "-" indicates the unpruned reference model. Each column shows the accuracy of the pruned models on a specific evaluation language. The lighter the color, the better.

Arabic on ARC and MMLU). For the remaining cases, calibration with the target language achieves the second-best or comparable performance. Therefore, calibrating using the target language typically results in acceptable, though not consistently the best performance. Considering the standard deviations shown in Table 7 in Appendix E, *calibration with the target language is a reasonable choice out of downstream performance*.

In terms of downstream performance across different languages, the full-sized baseline models exhibit the strongest performance in English, followed by other Latin languages, with Russian next in line. Arabic and Chinese downstream tasks generally are the most challenging for models. Further, pruning can sometimes alter the original ranking of languages observed in the baseline models. For example, on the Belebele benchmark, the Llama-3 8B model achieves a baseline accuracy of 55.3 for Arabic and 44.8 for Chinese, but the pruned models reverse this trend and achieve an accuracy of 45.1 for Arabic and 57.0 for Chinese. That is, *pruning can shift which languages the model performs best or worst on.*

*Calibrating on an outlier language or a similar one could benefit downstream tasks in non-English?*

We include Swahili as an outlier language, which is out-of-domain for the model as it is not included in the pre-training corpora of Llama-3 and Aya-23. In column-wise comparison, the SW cell is consistently the darkest or relatively dark one. That is, in most cases, calibration with Swahili results in the worst or the second-worst performance across different calibration language strategies, given a downstream task.

Furthermore, there is no consistent pattern related to the similarity of calibration-evaluation language pairs. For instance, Latin language pairs such as English-Spanish (calibrating in English and evaluating in Spanish) or pairs from the same language family, like English-German, do not always yield optimal performance. Conversely, pairs with different writing systems, such as Russian-English or Spanish-Arabic, do not consistently perform poorly. On the other hand, calibration with a dissimilar language, i.e. Chinese, often results in particularly low accuracy across many tasks and evaluation languages, as demonstrated by the darker row of "ZH". In summary, we observe *no benefit from calibrating with an outlier language or a similar language.*

*Does the model or the pruning method matter?*

Between Llama-3 8B and Aya-23 8B, despite their similar decoder-only architecture, Table 2 highlights distinct performance patterns between the two models under investigation. With its larger vocabulary size in the six non-English languages, Aya-23 8B generally outperforms the Llama-3 8B model in most evaluation languages and tasks, both before and after pruning. Notably, Aya-23 8B experiences less performance drop after pruning but shows less stable results, often performing better on languages other than the one used for calibration. Overall, these results suggest that vocabulary size and pre-training data impact baseline performance and accuracy after pruning in a multilingual context.

Between Wanda and SparseGPT, Llama-3 8B's performance degrades less after pruning with SparseGPT pruning, while the performance of the pruned Aya-23 8B model varies, that is, excelling with Wanda or SparseGPT depends on the task. Specifically, SparseGPT produces a distinct diagonal pattern for Aya-23 8B on ARC and HellaSwag, while Wanda often yields superior performance for calibration with non-target languages on Belebele and MMLU. No pruning technique consistently performs best in all tasks.

### 4.3 OPEN DOMAIN QUESTION ANSWERING WITHOUT CONTEXT

*How does pruning impact the knowledge stored in multilingual LLMs?*

Table 3 shows the open-domain question-answering performance of Llama-3 8B and Aya-23 8B across various calibration-evaluation language pairs. This "closed-book" task, MKQA, provides no context in the prompt, and relies on the model's internal knowledge to generate answers. To ensure fair cross-languages comparisons, the MKQA dataset is fully parallel and primarily consists of entity-based and structured "atomic" answer types. See Appendix C for further details. First, we observe significant performance differences among the evaluation languages for the full-size models. While Latin languages perform best, Arabic and Chinese perform worst. Further, we notice a significant accuracy drop by pruning, even for performance in English. In summary, *pruning sub-*

*stantially impacts the storage and retrieval of knowledge in a multilingual model across different languages.*

## 4.4 MULTIPLE CALIBRATION LANGUAGES

*Using more languages in the calibration, i.e. bilingual and multilingual calibration set, will benefit the performance of pruned multilingual models?*

We repeat the experiments but include more languages in the calibration set. As English is the dominant language in pre-training data, we first test the combination of English and the target language for a bilingual calibration setup. We further experiment with including all seven languages in the calibration, i.e. multilingual setup. For all setups, the total calibration sample number remains the same, i.e. 128.

Comparing Table 8 with Table 2, we observe that bilingual and multilingual calibration sets generally outperform monolingual calibration for non-English target languages. For example, for Chinese on ARC, the seven-lingual calibration set, AR-DE-EN-ES-RU-SW-ZH, often yields higher accuracy than calibration with Chinese, i.e. the target language, across models and pruning methods. That is, AR-DE-EN-ES-RU-SW-ZH achieves 28.3 (Wanda, Llama-3 8B), 30.0 (SparseGPT, Llama-3 8B), 32.0 (Wanda, Aya-23 8B) and 30.9 (SparseGPT, Aya-23 8B), consistently higher or comparable to monolingual calibration set, i.e. 27.0, 30.1, 30.8 and 31.7 respectively. On the other hand, multilingual calibration pruning often degrades the downstream performance in English, particularly on ARC when pruning Aya-23 with SparseGPT and on MMLU when pruning Llama-3 with Wanda. Interestingly, the best bilingual combination does not necessarily contain the target language but English. For example, for Russian on ARC when pruning with Wanda, calibration with EN-ES and AR-EN leads to the optimal performance for Llama-3 8B and Aya-23 8B.

In summary, *employing bilingual and multilingual calibration sets occasionally improves performance on downstream tasks, compared to monolingual calibration.* However, there is no clear pattern identifying which specific language combinations are most effective for a given target downstream language.

| | | | MKQA[f1] | | | | |
|---|---|---|---|---|---|---|---|
| | | AR | DE | EN | ES | RU | ZH |
| Llama-3-8B-Instruct | - - | 8.9 | 26.8 | 38.3 | 27.2 | 16.4 | 2.6 |
| Wanda | AR | 0.9 | 6.3 | 18.4 | 11.4 | 5.6 | 2.0 |
| | DE | 0.3 | 6.0 | 19.5 | 11.5 | 5.7 | 2.1 |
| | EN | 0.2 | 6.3 | 19.4 | 11.6 | 4.5 | 1.5 |
| | ES | 0.1 | 6.6 | 19.6 | 11.8 | 5.4 | 1.7 |
| | RU | 0.4 | 7.0 | 19.2 | 12.0 | 5.5 | 2.1 |
| | ZH | 0.7 | 4.8 | 18.0 | 10.1 | 3.5 | 2.1 |
| SparseGPT | AR | 3.5 | 8.0 | 19.8 | 11.3 | 6.4 | 0.9 |
| | DE | 0.3 | 10.5 | 19.8 | 11.2 | 6.1 | 1.6 |
| | EN | 1.0 | 9.7 | 20.2 | 12.1 | 5.8 | 1.5 |
| | ES | 0.8 | 8.4 | 20.3 | 12.0 | 6.3 | 2.4 |
| | RU | 0.9 | 9.0 | 19.9 | 11.9 | 7.3 | 1.8 |
| | ZH | 0.1 | 7.4 | 18.1 | 9.3 | 5.1 | 0.7 |
| Aya-23-8B | - - | 14.8 | 28.8 | 46.8 | 31.0 | 22.7 | 0.3 |
| Wanda | AR | 6.3 | 8.8 | 16.0 | 10.5 | 7.3 | 0.1 |
| | DE | 5.8 | 8.4 | 18.0 | 10.9 | 7.8 | 0.1 |
| | EN | 5.2 | 8.1 | 17.5 | 10.2 | 7.0 | 0.1 |
| | ES | 5.6 | 8.1 | 16.8 | 10.6 | 7.6 | 0.1 |
| | RU | 5.9 | 7.7 | 16.4 | 10.6 | 7.7 | 0.1 |
| | ZH | 5.3 | 8.5 | 16.8 | 10.4 | 6.8 | 0.1 |
| SparseGPT | AR | 5.4 | 8.4 | 18.0 | 10.6 | 6.8 | 0.1 |
| | DE | 4.9 | 7.8 | 19.4 | 8.6 | 6.4 | 0.0 |
| | EN | 3.7 | 7.5 | 17.3 | 9.9 | 6.3 | 0.1 |
| | ES | 3.6 | 7.7 | 16.6 | 8.8 | 5.7 | 0.2 |
| | RU | 5.1 | 7.1 | 18.3 | 9.8 | 6.3 | 0.4 |
| | ZH | 4.6 | 6.6 | 14.9 | 8.6 | 5.5 | 0.0 |

Table 3: MKQA F1 accuracy over seven evaluation languages for the Llama-3 8B and Aya-23 8B models pruned with Wanda and SparseGPT for 50% unstructured sparsity.

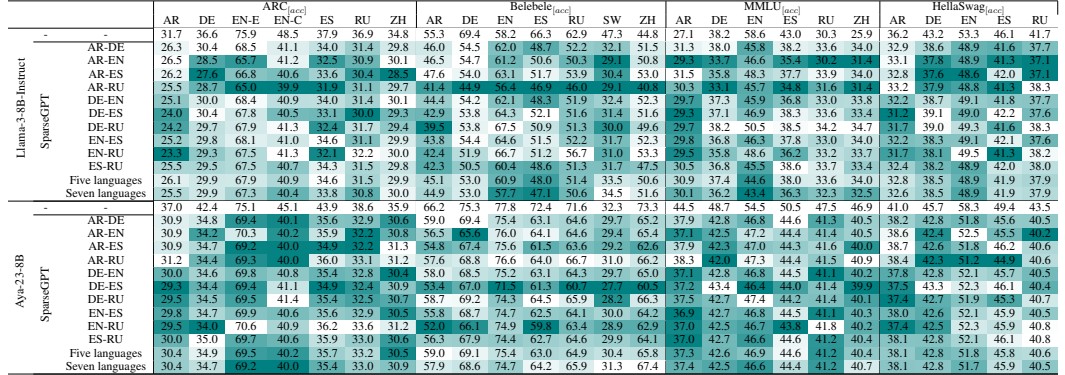

| | | ARC[acc] | | | | | | | Belebele[acc] | | | | | | | MMLU[acc] | | | | | | HellaSwag[acc] | | | | |
|---|---|---|---|---|---|---|---|---|---|---|---|---|---|---|---|---|---|---|---|---|---|---|---|---|---|---|
| | | AR | DE | EN-E | EN-C | ES | RU | ZH | AR | DE | EN | ES | RU | SW | ZH | AR | DE | EN | ES | RU | ZH | AR | DE | EN | ES | RU |
| Llama-3-8B-Instruct SparseGPT | - | 31.7 | 36.6 | 75.9 | 48.5 | 37.9 | 36.9 | 34.8 | 55.3 | 58.2 | 66.3 | 62.9 | 47.3 | | 44.8 | 27.1 | 38.2 | 58.6 | 43.0 | 30.3 | 25.9 | 36.2 | 43.2 | 53.3 | 46.1 | 41.7 |
| | AR-DE | 26.3 | 30.4 | 68.5 | 41.1 | 34.0 | 31.4 | 29.8 | 46.0 | 54.5 | 62.0 | 48.7 | 52.2 | 32.1 | 51.5 | 31.3 | 38.0 | 45.8 | 38.2 | 33.6 | 34.0 | 32.9 | 38.6 | 48.9 | 41.6 | 37.7 |
| | AR-EN | 26.5 | 28.5 | 65.7 | 41.2 | 32.5 | 30.9 | 30.1 | 46.5 | 54.7 | 61.2 | 50.6 | 50.3 | 29.1 | 50.8 | 29.3 | 33.7 | 46.6 | 35.6 | 30.2 | 31.4 | 33.1 | 37.8 | 48.9 | 41.3 | 37.1 |
| | AR-ES | 26.2 | 27.6 | 66.8 | 40.6 | 33.6 | 30.4 | 28.5 | 47.6 | 54.0 | 63.1 | 51.7 | 53.9 | 30.4 | 53.0 | 31.5 | 35.8 | 48.3 | 37.7 | 33.9 | 34.0 | 32.8 | 37.6 | 48.6 | 42.0 | 37.1 |
| | AR-RU | 25.5 | 28.7 | 65.0 | 39.9 | 31.9 | 31.1 | 29.7 | 41.4 | 44.9 | 56.4 | 46.9 | 46.0 | 29.1 | 40.8 | 30.3 | 33.1 | 45.7 | 34.8 | 31.6 | 31.4 | 33.2 | 37.9 | 48.8 | 41.3 | 38.3 |
| | DE-EN | 25.1 | 30.0 | 68.4 | 40.9 | 34.0 | 31.4 | 30.1 | 44.4 | 54.2 | 62.1 | 48.3 | 51.9 | 32.4 | 52.3 | 29.7 | 37.3 | 45.9 | 36.8 | 33.0 | 33.8 | 32.2 | 38.7 | 49.1 | 41.8 | 37.7 |
| | DE-ES | 24.0 | 30.4 | 67.8 | 40.5 | 33.1 | 30.0 | 29.3 | 42.9 | 53.8 | 64.3 | 52.1 | 51.6 | 31.4 | 51.6 | 29.3 | 37.1 | 46.9 | 38.3 | 33.6 | 33.4 | 33.2 | 39.1 | 49.0 | 42.2 | 37.6 |
| | DE-RU | 24.2 | 29.7 | 67.9 | 41.3 | 32.4 | 31.7 | 29.4 | 39.5 | 53.8 | 67.5 | 50.9 | 51.3 | 30.0 | 49.6 | 29.7 | 38.2 | 50.5 | 38.5 | 34.2 | 34.7 | 31.7 | 39.0 | 49.3 | 41.6 | 38.3 |
| | EN-ES | 25.2 | 29.6 | 68.1 | 41.0 | 34.6 | 31.1 | 29.9 | 43.8 | 54.4 | 64.6 | 51.5 | 52.2 | 31.7 | 52.3 | 29.8 | 36.8 | 46.3 | 37.8 | 33.0 | 34.0 | 32.2 | 38.2 | 48.9 | 42.1 | 37.6 |
| | EN-RU | 23.5 | 29.3 | 67.5 | 41.3 | 32.1 | 32.2 | 30.0 | 42.4 | 51.9 | 66.7 | 51.2 | 56.7 | 31.0 | 53.3 | 29.5 | 35.8 | 48.6 | 36.2 | 33.2 | 33.7 | 31.7 | 38.1 | 49.5 | 41.3 | 38.2 |
| | ES-RU | 25.5 | 29.5 | 67.5 | 40.7 | 34.3 | 31.5 | 29.8 | 42.3 | 50.5 | 60.4 | 48.6 | 51.3 | 31.7 | 47.5 | 30.5 | 36.8 | 45.5 | 38.6 | 33.7 | 33.4 | 32.8 | 38.2 | 48.9 | 41.9 | 37.9 |
| | Five languages | 26.1 | 29.9 | 67.9 | 40.9 | 34.6 | 31.5 | 29.9 | 45.1 | 53.0 | 60.9 | 48.0 | 51.4 | 33.5 | 50.6 | 30.9 | 37.4 | 44.6 | 38.0 | 33.6 | 34.0 | 32.6 | 38.5 | 48.9 | 41.9 | 37.9 |
| | Seven languages | 25.5 | 29.9 | 67.3 | 40.4 | 33.8 | 30.8 | 30.0 | 44.9 | 53.0 | 57.7 | 47.1 | 50.6 | 34.5 | 51.6 | 30.1 | 36.2 | 43.4 | 36.3 | 32.3 | 32.5 | 32.6 | 38.5 | 48.9 | 41.9 | 37.9 |
| Aya-23-8B SparseGPT | - | 37.0 | 42.4 | 75.1 | 45.1 | 43.9 | 38.6 | 35.9 | 66.2 | 75.3 | 77.8 | 72.4 | 71.6 | 32.3 | 73.3 | 44.5 | 48.7 | 54.5 | 50.5 | 47.5 | 46.9 | 36.2 | 43.2 | 53.3 | 49.4 | 43.5 |
| | AR-DE | 30.9 | 34.8 | 69.4 | 40.1 | 35.6 | 32.9 | 30.6 | 59.0 | 69.4 | 75.4 | 63.1 | 64.6 | 29.7 | 65.2 | 37.9 | 42.8 | 46.8 | 44.6 | 41.3 | 40.5 | 38.2 | 42.8 | 51.8 | 45.6 | 40.5 |
| | AR-EN | 30.9 | 34.2 | 70.3 | 40.2 | 35.9 | 32.2 | 30.8 | 56.5 | 65.6 | 76.0 | 64.1 | 64.6 | 29.4 | 65.4 | 37.1 | 42.5 | 47.2 | 44.4 | 41.4 | 40.5 | 38.6 | 42.4 | 52.5 | 45.5 | 40.2 |
| | AR-ES | 30.9 | 34.7 | 69.2 | 40.0 | 34.9 | 32.2 | 31.3 | 54.8 | 67.4 | 75.6 | 61.5 | 63.6 | 29.2 | 62.6 | 37.9 | 42.3 | 47.0 | 44.3 | 41.6 | 40.0 | 38.7 | 42.6 | 51.8 | 46.2 | 40.6 |
| | AR-RU | 31.2 | 34.4 | 69.3 | 40.0 | 36.0 | 33.1 | 31.2 | 57.6 | 68.8 | 76.6 | 64.0 | 66.7 | 31.0 | 66.2 | 38.4 | 42.3 | 47.3 | 44.4 | 40.9 | 40.4 | 38.4 | 42.3 | 51.2 | 44.9 | 40.6 |
| | DE-EN | 30.0 | 34.6 | 69.8 | 40.8 | 35.4 | 32.8 | 30.4 | 58.0 | 68.5 | 75.2 | 63.1 | 64.3 | 29.7 | 65.0 | 37.1 | 42.8 | 46.8 | 44.5 | 41.1 | 40.2 | 37.8 | 42.8 | 52.1 | 45.7 | 40.5 |
| | DE-ES | 29.3 | 34.4 | 69.4 | 41.1 | 34.9 | 32.4 | 30.9 | 53.4 | 67.0 | 71.5 | 60.7 | 60.7 | 27.7 | 60.5 | 37.2 | 43.4 | 46.4 | 44.0 | 41.4 | 39.9 | 37.5 | 43.3 | 52.3 | 46.1 | 40.4 |
| | DE-RU | 29.5 | 34.5 | 69.5 | 41.4 | 35.4 | 32.5 | 30.7 | 58.7 | 69.2 | 74.3 | 64.5 | 65.9 | 28.2 | 66.3 | 36.9 | 42.7 | 47.4 | 44.2 | 41.4 | 40.1 | 37.4 | 42.7 | 51.9 | 45.3 | 40.7 |
| | EN-ES | 29.8 | 34.7 | 69.9 | 40.6 | 35.6 | 32.9 | 30.5 | 55.8 | 68.7 | 74.7 | 62.5 | 64.1 | 30.0 | 64.2 | 37.0 | 42.7 | 46.8 | 44.5 | 41.1 | 40.3 | 37.8 | 42.6 | 52.1 | 45.9 | 40.5 |
| | EN-RU | 29.5 | 34.0 | 70.6 | 40.9 | 36.2 | 33.6 | 31.2 | 52.0 | 66.1 | 74.9 | 59.8 | 63.4 | 28.9 | 62.9 | 37.0 | 42.5 | 46.7 | 43.8 | 41.8 | 40.2 | 37.4 | 42.5 | 52.3 | 45.9 | 40.8 |
| | ES-RU | 30.0 | 35.0 | 69.7 | 40.6 | 35.3 | 33.0 | 30.6 | 56.3 | 67.9 | 74.4 | 62.7 | 64.6 | 29.9 | 64.1 | 37.0 | 42.7 | 46.6 | 44.6 | 41.2 | 40.4 | 38.2 | 42.1 | 52.1 | 46.1 | 40.8 |
| | Five languages | 30.4 | 34.9 | 69.5 | 40.2 | 35.7 | 33.2 | 30.5 | 59.0 | 69.1 | 75.4 | 63.0 | 64.9 | 30.4 | 65.8 | 37.3 | 42.6 | 46.9 | 44.6 | 41.2 | 40.4 | 38.1 | 42.8 | 51.8 | 45.8 | 40.6 |
| | Seven languages | 30.4 | 34.7 | 69.2 | 40.0 | 35.4 | 33.0 | 30.9 | 57.9 | 68.6 | 74.7 | 64.2 | 65.9 | 31.3 | 67.4 | 37.4 | 42.5 | 46.6 | 44.4 | 41.2 | 40.7 | 38.1 | 42.8 | 51.7 | 45.9 | 40.5 |

Table 4: Pruned Model Performance averaged over three pruning runs. The lighter, the less drop, the better. Five languages: AR, DE, EN, ES, RU. Seven languages: AR, DE, EN, ES, RU, SW, ZH.

### 4.5 Impact of Model Sizes

To investigate the scaling impact towards pruning behavior, we repeat experiments in Table 7 on the larger Llama-3 70B and Aya-23 35B models, the results of which are reported in Appendix E. Overall, pruning performance increases with higher baseline accuracy of the full-sized models. However, we observe that ***the performance patterns and findings from the smaller models, mentioned above, do not consistently hold true on their bigger counterparts***.

### 4.6 Quantization

We further explore the impact of the calibration language in quantization on downstream task performance across different languages. We use GPTQ (Frantar et al., 2023) to quantize weights to 4 bits with a group size of 128 and 8 bits with a group size of 128 (equivalent to 50% sparsity in pruning) on LLaMA-3-8B. We follow our pruning setup for downstream tasks and languages for calibration and evaluation. The results are present in Table 14 and 15 in Appendix F, revealing several key findings that are consistent with our findings on pruning: (1) calibrating with the target language tends to yield better downstream performance; (2) pruning can alter which languages the model performs best or worst on; and (3) calibrating with an outlier or a linguistically similar language does not provide any notable advantage.

## 5 Analysis

In reviewing Table 1 and Table 7 together, we poist that calibrating on the target language effectively preserves language-related features but not language-agnostic features, such as knowledge memories and reasoning abilities. This implies that the smallest weights or weights of smallest activations contribute to the model's nuanced reasoning or knowledge memorization and retrieval processes. The rationale here is that both SNR and pruning error measure the extent of model change after pruning, which is dominated by large values, that pruning aims to preserve. On the other hand, perplexity reflects the overall language modeling capability. The similar diagonal patterns across the three metrics, SNR, pruning error and perplexity, as shown in Table 1, suggest that calibrating on the target language, i.e. identifying and pruning the smallest values, preserves the next-token prediction capability (perplexity) in the target language, along with the optimal pruning performance in terms of pruning the smallest values (SNR and pruning errors).

However, optimal performance in downstream tasks requires more than just surface-level language modeling; it also depends on robust reasoning capabilities. The suboptimal performance observed in downstream tasks may be attributed to the degradation of language-agnostic reasoning capabilities. This phenomenon suggests that removing the weights or weights of the smallest activations might inadvertently compromise the model's ability to process reasoning, underscoring the reasoning role of these pruned weights, i.e. the smallest weights or weights of the smallest activations.

To validate our hypothesis, we investigate the internal changes of pruned models on three different levels: subspace level, matrix level, neuron level (columns in a matrix, followed by non-linearity). Previous studies on multilingualism of LLMs have sought to separate language-specific features from language-agnostic features, either at the neuron level (Tang et al., 2024; Zhao et al., 2024; Wang et al., 2024) or from an extracted subspace perspective (Xie et al., 2022). To comprehend the underlying mechanisms driving the disparities in pruning metrics and downstream performance when using different calibration languages, we examine the shifts in language-specific and agnostic features and neurons induced by the pruning process.

### 5.1 Language-specific Subspace Representations

The core idea of Low-rank Subspace for language-Agnostic Representations (LSAR) by Xie et al. (2022) is to construct a mean embedding matrix $M \in \mathbb{R}^{d \times L}$ by concatenating L language embeddings. Subsequently, LSAR decomposes $M$ into a vector $\mu$ representing shared signals across languages and a matrix $M_s$ specifying a low-rank subspace on which different languages express different linguistic signals. This decomposition process is achieved via singular value decomposition on solving: $\min_{\mu, M_s, \Gamma} \left\| M - \mu \mathbf{1}^\top - M_s \Gamma^\top \right\|_F^2$ s.t. $\mu \perp \text{Span}(M_s)$ with the orthogonality

constraint. $\boldsymbol{\Gamma} \in \mathbb{R}^{L \times r}$ represents the coordinates of language-specific signals along the subspace's $r$ components, and $\mathbf{1} \in \mathbb{R}^d$ is a vector of all ones. We can extract language-specific features $\boldsymbol{a}$ from a sentence embedding $\boldsymbol{e}$ using $\boldsymbol{M}_s$ by projecting $\boldsymbol{e}$ into and back from the low-rank, language signal retaining subspace, $\boldsymbol{a} = \boldsymbol{e} - \boldsymbol{s} = \boldsymbol{e} - \boldsymbol{M}_s \boldsymbol{M}_s^T \boldsymbol{e}$. See Appendix D.2 for more details and the experiment setup.

We employ LSAR to decomposite the token-wise averaged embedding output of each layer of SparseGPT-pruned Llama-3 8B into language-specific and language-agnostic features, and estimate their changes after pruning. Figure 1 shows the layer-wise magnitude of differences ($\Delta$ magnitude) of (a) language-agnostic features and (b) language-specific features after pruning. From a high-level perspective, the greater $\Delta$ magnitude suggests the greater changes in terms of hidden state after pruning, thus the greater pruning error and the worse pruning performance. Unlike the pruning error presented in Table 1, here we separately analyze the pruning error for language-agnostic and language-specific features. See Figure 5 in Appendix E for layer-wise full plots.

First, calibrating on the target language leads to relatively smaller pruning errors in terms of language-specific features as shown in the sub Figure (b) in Figure 1. This phenomenon is observed across layers, particularly noticeable in layer 32, as pinpointed by stars in Figure 1. This potentially explains the findings in Section 4.1 that calibration on the target language leads to the lowest perplexity, which is associated with a robust language-specific linguistic modeling capability. On the other hand, as indicated by the relatively flat horizontal lines across languages and layers in sub Figure (a), the pruning error on the language-agnostic features remains similar regardless of the calibration languages. This pattern helps explain the sub-optimal downstream task performance observed in Table 7, where no single calibration language consistently yields optimal performance across downstream tasks, including cases where the calibration is performed on the target language. That is, ***the selection of calibration language is unlikely to impact the language-agnostic features associated with understanding, reasoning, and knowledge retrieval. In contrast, the preservation of language-specific features, such as modeling capabilities related to perplexity, depends on the selection of calibration languages.***

Second, the middle layers (as shown in the second and third columns in Figure 1) exhibit greater $\Delta$ magnitude on language-agnostic feature representations and smaller $\Delta$ magnitude on language-specific feature representations. This indicates that pruning errors can be predominantly attributed to the pruning errors on language-agnostic features, with less pruning error arising from language-specific features. Therefore, we conclude ***pruning generally affects language-agnostic features—potentially associated with reasoning and knowledge storage—more significantly than it impacts language-specific features.***

## 5.2 PRUNING MASK SIMILARITY

To better understand the internal changes within the model after pruning, we conduct a matrix-level analysis by calculating the Intersection over Union (IoU) of pruning masks across different

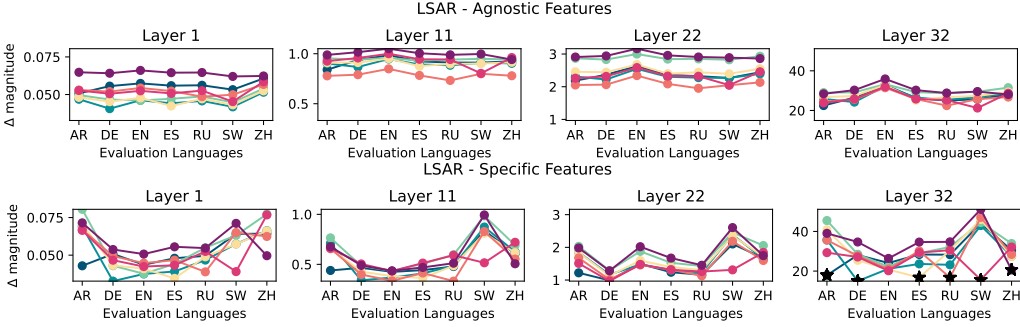

Figure 1: Magnitude difference of language-agnostic and specific features after pruning, averaged over 200 data per language from validation sets. The X-axis represents the evaluation language. The calibration languages are indicated by the color as: AR, DE, ES, EN, RU, SW, ZH. The greater $\Delta$ magnitude, the greater pruning error on language-agnostic features (Fig.b) or language-specific features (Fig.c).

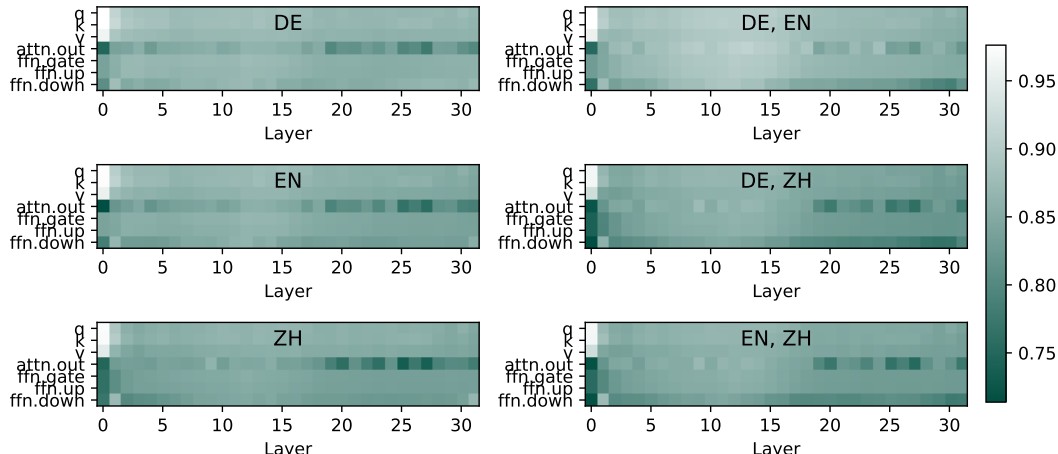

Figure 2: Pruning similarities (IoU) between using different calibration languages. Left: IoU of pruning masks for three calibration sets of the same language. Right: IoU between pruning masks for different calibration languages. The higher IoU (indicated as a lighter color), the more similar pruning masks between different calibration languages.

calibration sets, obtaining a measure of pruning mask similarity. The formal expression and details explaining IoU are presented in Appendix D.3. We use the pruning masks from the Llama-3 8B model pruned with SparseGPT for 50% unstructured sparsity. To reduce calibration set-dependent noise as prevalent in the downstream tasks, we first compute the intersection $M_I^l$ of pruned neuron indices $M_i^l$ across three pruned models calibrated with different seeds $i$ but in the same language $l$. This intersection represents more stable neuron indices.

The IoU of the left side of Figure 2 depicts the proportion of $M_I^l$ with respect to all pruned neuron indices. It reveals high pruning mask similarity in the attention query, key and value in the first layer, while the attention output projection varies more significantly, especially after the 20th layer. This suggests that ***pruning struggles to consistently identify essential neurons in the attention output projection, partly responsible for the models reasoning capabilities***.

The right plot compares the IoU of intersected neuron indices $M_i^l$ from the left plot across calibration in German, English, and Chinese. Notably, the attention query, key, and value in the first layer consistently achieve high IoUs above 0.95 across all languages, indicating that these components handle inputs similarly, irrespective of language differences. However, the attention output and FFN down projection show lower IoU, especially in early layers, with similarity peaking in middle layers (3rd to 15th) before decreasing again in later layers.

This pattern suggests that the attention output and FFN down projection in early and late layers handle language-specific signals, while middle layers process language-agnostic signals, supporting Figure 1. In other words, ***early layers focus on language comprehension, middle layers on language-independent reasoning, and later layers on generating language-specific predictions.*** This aligns with Zhao et al. (2024), who proposes that LLMs first comprehend queries by converting multilingual inputs into English in the early layers, reason in English in the intermediate layers, and then generate responses aligned with the input language in the final layers.

### 5.3 LANGUAGE-SPECIFIC NEURONS

This section investigates neuron-level changes after pruning using Language Activation Probability Entropy (LAPE) as introduced by Tang et al. (2024). We focus on neurons of the up projections of the feed-forward layers (FFNs) followed by the non-linearity. A neuron is considered activated when the non-linearity output is greater than zero. LAPE measures the likelihood of individual neurons activating across different language inputs, taking neurons with high activation probability for one or two languages and lower probabilities for all others (i.e. low LAPE score) as language-specific. Details on the LAPE calculation and experiment settings are provided in Appendix D.4.

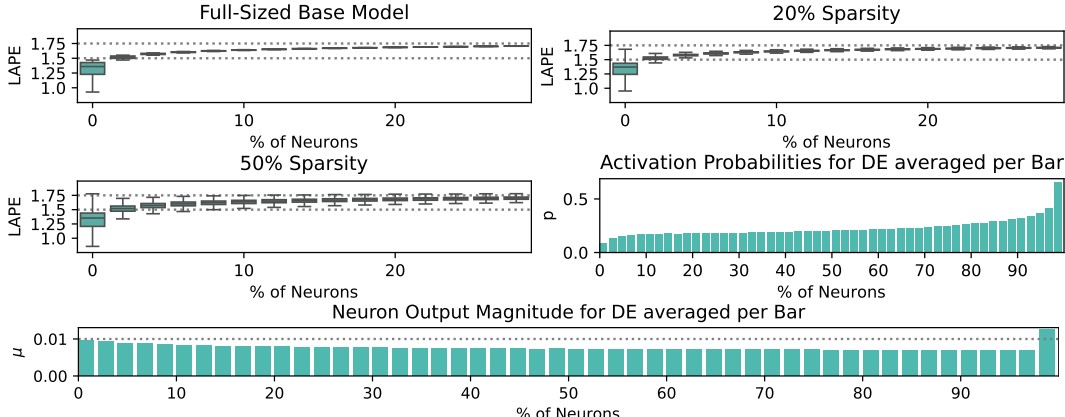

Figure 3: Language Entropy for FFN neurons of Llama-3 8B and its SparseGPT-pruned versions calibrated for DE. The bottom-right plot visualizes the activation likelihood of neurons on DE. The bottom subfigure shows the mean absolute neuron outputs for DE input. *The lower the LAPE score, the more specialized the neuron is for a particular language.* See Appendix G for the activation probability and neuron output magnitudes across all languages.

In Figure 3, each bar and boxplot corresponds to the same group of neurons, comprising 2% of the total neuron population. Neurons in each plot are sorted in ascending order based on their LAPE score in the full-sized model, enabling a comparison of changes in LAPE scores after pruning. We only plot the neuron groups of the 30% lowest LAPE score due to the rapidly diminishing variance for boxplots of higher LAPE scores. This suggests a lower impact of pruning on the activation frequency of high LAPE neurons, i.e. language-agnostic neurons. On the other hand, for the neuron group with the lowest LAPE scores (the leftmost boxplot), the higher the model-sparsity is, the longer whiskers (higher entropy variance) these neurons have. This indicates that ***pruning introduces LAPE noise, shifting the LAPE score distribution and creating new language-specific (low LAPE) and agnostic (high LAPE) neurons***. The distribution shift of language-agnostic neurons in FFNs may contribute to performance degradation in downstream tasks. This hypothesis aligns with previous causal tracing studies, which have frequently identified FFNs are crucial for knowledge retention and retrieval (Meng et al., 2022; 2023).

Moreover, by analysis of boxplots and the bottom-right plot together, we find that the neuron group of the lowest LAPE has the lowest activation probabilities, while neurons with the highest LAPE scores activate more frequently. This indicates that ***pruning struggles to retain low activation probability of language-specific neurons***. We also examine the absolute neuron output magnitude for these neglected neurons. The bottom plot in Figure 3 shows that despite the low activation probability, low LAPE neurons tend to have high output magnitudes. Pruned models show no significant differences in output magnitudes, leading us to conclude that pruning retains neuron output magnitudes in FFN modules but may fail to preserve activation frequency.

## 6 CONCLUSION

In this paper, we empirically demonstrate that the choice of calibration language influences the downstream performance of pruned models across various evaluation languages. Particularly, calibration with the target language mainly benefits preserving perplexity performance but does not necessarily help maintain the downstream task performance. Our analysis of changes within the model after pruning indicates that calibrating in the test language does not reliably benefit layers associated with semantic understanding. Additionally, we recommend practitioners do not depend on perplexity for accessing the pruned model, or performance in English to estimate the performance on target languages.

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

## A    LIMITATIONS

**Generality of Findings.**    Due to resource constraints, we predominantly experimented with the small versions of Llama-3 and Aya-23, and validated our findings with fewer pruning runs on their counterpart large version. Since our results translate between model families, and to bigger model sizes, we assume a certain degree of generalization. Nonetheless, other models trained with different techniques or for other tasks might show different behavior. Given the pace of this research field, it is also unclear whether these results translate to future models.

**Underrepresented Languages.**    Our experiments focused on languages with sufficient model and downstream task support. However, this selection does not encompass all languages of interest, particularly mid and low-resource languages that are underrepresented in the pre-training, and challenging to evaluate due to the lack of benchmark support. Future research could benefit from including more languages to explore the interplay between different language families or writing systems and performance after pruning.

## B    CALIBRATION AND TEST LANGUAGES

| Language | Language family | Writing system |
|---|---|---|
| Modern Standard Arabic (AR) | Afro-Asiatic | Arabic |
| German (DE) | Germanic | Latin |
| English (EN) | Germanic | Latin |
| Spanish (ES) | Romance | Latin |
| Russian (RU) | Balto-Slavic | Cyrillic |
| Swahili (SW) | Atlantic-Congo | Latin |
| Chinese (simplified) (ZH) | Sino-Tibetan | Simplified Han |

Table 5: Languages used for calibration and testing throughout this paper with their corresponding language family and writing system.

## C    DOWNSTREAM DATASETS

Throughout the paper we used the following widely employed datasets for automated benchmarking. All evaluations were conducted in a zero-shot fashion and employ the chat-template of the respective instruction-tuned model.

**ARC:**    The AI2 Reasoning Challenge (ARC) dataset introduced by Clark et al. (2018) tests the reasoning and knowledge capabilities through natural, grad-school multiple choice science questions originally authored for standardized human tests. The dataset comprises a total of 7787 questions in english divided into a Challenge set (ARC-C) of hard to answer questions and an Easy set (ARC-E) of questions.

For evaluation in English, we use the original datasets (e.g. ARC-c & ARC-e), for all other languages the translated version from  Dac Lai et al. (2023) is utilized.

**Belebele:**    This carefully curated dataset evaluates 4-way multiple-choice machine reading comprehension among 122 language options, broadly focussing on high-, medium-, and low-resource languages (Bandarkar et al., 2023). Each of the 900 samples is based on an English FLORES-200 passage that has been translated into the respective target language by fluent expert speakers. Hence, the dataset is fully parallel, allowing direct performance comparison across all languages.

**HellaSwag:**    The HellaSwag dataset by Zellers et al. (2019) comprises 10042 english samples testing commonsense natural language inference on event descriptions that need to be continued/completed in a multiple-choice fashion. Though easily answerable by humans, such paragraph continuation questions still pose a challenge for state-of-the-art LLMs.

**MKQA:** Multilingual Knowledge Questions and Answers (MKQA) (Longpre et al., 2020) is an open-domain question-answering evaluation set of 10000 samples aligned across 26 languages by human translators. Its question-answer pairs were filtered from the Google Natural Questions dataset (Kwiatkowski et al., 2019), annotating real Google search user questions with answers found on Wikipedia. Given a question, the task is to predict the correct answer or give no answer without any additional context provided. Hence, this dataset tests the knowledge retrieval capabilities of models. For our evaluation, we remove all unanswerable and questions requiring overly long answers for simplicity, yielding a total 6758 remaining samples.

There are a total of 6,758 evaluation samples included in our experiment, after eliminating those with unanswerable and long, imprecise answers.

**MMLU:** The Massive Multitask Language Understanding (MMLU) dataset (Hendrycks et al., 2021) is an english benchmark designed to evaluate a model's ability to handle diverse subjects across multiple domains. It contains a total of 14042 question-answer pairs covering 57 task categories, ranging from high school and college-level subjects to professional and specialized knowledge. Each task includes multiple-choice questions, and the dataset measures both the model's factual knowledge and reasoning abilities.

**Translated Datasets from Okapi:** The Okapi framework, introduced by Dac Lai et al. (2023), focuses on instruction tuning LLMs using reinforcement learning from human feedback (RLHF) across multiple languages. As part of its resources, it includes translated versions of the ARC, HellaSwag, and MMLU datasets, generated using ChatGPT. We leverage these translations to complement the evaluation of the original English datasets in multiple languages.

# D    MATH & SETTINGS

## D.1    PRUNING ERROR AND SNR

We resort to the definition of Kuzmin et al. (2023) for computing the Pruning Error and SNR. Since originally both of them relate to model weights, which are input and therefore language independent, we compute Pruning Error and SNR with respect to the outputs of each layer, i.e. the hidden states. To cope with different hidden state magnitudes per layer, all hidden states are normalized layer-wise by their average vector norm, yielding the following pruning error:

$$\mathbb{E}[(\boldsymbol{H}^{(k)} - \widetilde{\boldsymbol{H}}^{(k)})^2] = \frac{1}{Nd}\sum_{i=1}^{N}\sum_{j=1}^{d}\left(\frac{h_{i,j}^{(k)} - \widetilde{h}_{i,j}^{(k)}}{\mu^{(k)}}\right)^2, \mu^{(k)} = \frac{1}{N}\sum_{i=1}^{N}||\boldsymbol{h}_i^{(k)}||_2 \tag{1}$$

with $\boldsymbol{H}^{(k)} \in \mathbb{R}^{N \times d}$ denoting the hidden states of all $N$ tokens in layer $k$ before and $\widetilde{\boldsymbol{H}}^{(k)} \in \mathbb{R}^{N \times d}$ after pruning, while $h_{i,j}^{(k)}$ refers to the hidden state of the $j$-th feature element of the $i$-th token in $H^{(k)}$. Subsequently, the SNR of a single layer $k$ is computed as

$$SNR_{dB}^{(k)} = 10 \cdot log_{10}(\mathbb{E}(\boldsymbol{H}^{(k)})/\mathbb{E}[(\boldsymbol{H}^{(k)} - \widetilde{\boldsymbol{H}}^{(k)})^2]), \text{ with } \mathbb{E}(\boldsymbol{H}^{(k)}) = \frac{1}{Nd}\sum_{i=1}^{N}\sum_{j=1}^{d}\left(\frac{h_{i,j}^{(k)}}{\mu^{(k)}}\right)^2. \tag{2}$$

The final model SNR is the average over all layer-wise $SNR_{dB}^{(k)}$.

## D.2    LOW-RANK SUBSPACE FOR LANGUAGE-AGNOSTIC REPRESENTATIONS (LSAR)

### D.2.1    LOW-RANK SUBSPACE COMPUTATION

Embeddings posess language identity components that can be extracted via Low-rank Subspace for language-Agnostic Representations (LSAR) as introduced by Xie et al. (2022). They project sentence embeddings (i.e. averaged embeddings of a sentence/short input) into a low-dimensional

subspace via a projection matrix $M_s \in \mathbb{R}^{d \times r}$. $M_s$ describes a low-rank subspace of the original multilingual latent space, spanned by $r$ components. *The following describes how to obtain $M_s$.*

First, the mean embeddings $\boldsymbol{\mu}_l = \frac{1}{n} \sum_{i=1}^{n} \boldsymbol{e}_l^i$ of each language $l$ are computed to then concatenate all $\boldsymbol{\mu}_l$ column-wise into matrix $M \in \mathbb{R}^{d \times L}$. To capture the relationships among the mean embedding vectors in $M$, it is decomposed into the matrix $M_s \in \mathbb{R}^{d \times r}$, spanning the low-rank subspace for linguistic, language-specific signals, and a vector $\boldsymbol{\mu}$ orthogonal to $\mathrm{Span}(M_s)$ representing shared, language-agnostic feature elements. This poses an optimization problem

$$\min_{\boldsymbol{\mu}, M_s, \boldsymbol{\Gamma}} \| M - \boldsymbol{\mu} \mathbf{1}^\top - M_s \boldsymbol{\Gamma}^\top \|^2$$
$$\text{s.t. } \boldsymbol{\mu} \perp \mathrm{Span}(M_s)$$

with $\mathbf{1} \in \mathbb{R}^d$ containing all ones, that can be solved optimally using a Singular Value Decomposition (SVD). The following omits the proof and only shows how to compute $M_s$, for more details please refer to the original paper by Xie et al. (2022).

$M_s$ is obtained in a two step process. First, $M$ is approximated in low-dimensional space by centering $M$ with $M_C = M - \boldsymbol{\mu}' \mathbf{1}^T = M - \frac{1}{d} M \mathbf{1}$ and computing the SVD decomposition of $M_C$ for the $r$ largest singular values as follows:

$$M_s', \boldsymbol{s}, \boldsymbol{\Gamma}' = \text{Top-}r\text{SVD}(M_C) \tag{3}$$

$M_s'$ and $\boldsymbol{\Gamma}'$ are the left and right singular vectors. Then $M$ is reconstructed from $M_s'$, $\boldsymbol{s}$ and $\boldsymbol{\Gamma}$ via

$$M' = \boldsymbol{\mu}' \mathbf{1}^T + M_s' (\boldsymbol{\Gamma}' \mathbf{I} \boldsymbol{s})^T \tag{4}$$

with $\mathbf{I} \in \mathbb{R}^{r \times r}$ as identity matrix. The second step is to enforce the orthogonality constraint $\boldsymbol{\mu} \perp \mathrm{Span}(M_s)$. This involves re-centering $M'$ again for SVD preparation to

$$M_C' = M' - \boldsymbol{\mu} = M' - \frac{1}{\|M'^{-1} \mathbf{1}\|^2} M'^{-1} \mathbf{1}. \tag{5}$$

Afterwards, the SVD for the reconstructed $M'$ is computed, obtaining $M_s$ as left singular values as follows:

$$M_s, {}_-, \boldsymbol{\Gamma} = \text{Top-}r\text{SVD}(M_C') \tag{6}$$

Finally, $M_s$ is the projection matrix into the low-rank latent space of language-specific signals. Conversely, $M_s$ allows extracting language-agnostic features for an embedding $\boldsymbol{e}_l$ by projection onto the null space of $M_s$. That is, projecting $\boldsymbol{e}_l$ into the low-rank subspace, retaining language-specific information in $\mathbb{R}^r$ only, followed by re-projection into the original embedding space, obtaining the language-specific feature components in $\mathbb{R}^d$. Then, obtaining language-agnostic $\boldsymbol{a}_l$ runs as follows:

$$\boldsymbol{a}_l = (I - M_s (M_s^T M_s)^{-1} M_s^T) \boldsymbol{e}_l = \boldsymbol{e}_l - M_s M_s^T \boldsymbol{e}_l \tag{7}$$

with $I$ as the identity matrix.

### D.2.2 LSAR EXPERIMENTAL SETTINGS

Since an analysis of token-embeddings via the Pruning Error and SNR does not predict downstream task performance well, we revert to sentence-embeddings or rather prompt-embeddings (i.e. the average of all token-embeddings, omitting special tokens from the chat template) to capture more high-level semantic and syntactic information. For separating language-agnostic from specific feature, we calculate a separate low-rank projection matrix $M_s^{(m)}$ per model $m$. Since we use LSAR to obtain pruning error metrics and do not demand any generalization to unseen data, $M_s^{(m)}$ is computed on the same samples used for evaluation.

### D.3 PRUNING MASK COMPARISON VIA IOU

We compare the similarity of pruning masks for different calibration languages by IoU in a two step process. First, we eliminate calibration set dependent noise by computing the intersection $M_I^l$ of the sets $M_i^l$ of pruned neuron indices for different calibration sets $i \in \{0, \ldots, N\}$ of the same language $l \in \{0, \ldots, L\}$. Then, $M_i^l$ is computed as follows:

$$M_I^l = \bigcap_{i=0}^{N} M_i^l, \qquad M_U^l = \bigcup_{i=0}^{N} M_i^l \tag{8}$$

$M_I^l$ only contains neuron indices that were pruned for all calibration sets and can therefore be deemed more stable. For comparing the similarity of pruning masks between different calibration languages $l_1$ and $l_2$ the IoU is utilized, running as follows:

$$IoU_{l_1,l_2} = \frac{\left| M_I^{l_1} \bigcap M_I^{l_2} \right|}{\left| M_I^{l_1} \bigcup M_I^{l_2} \right|} \tag{9}$$

For validation, in Figure 2 of the main text we also plot the $IoU_l$ for pruning masks $M_i^l$ of the same calibration language, visualizing pruning certainty.

### D.4 LANGUAGE ACTIVATION PROBABILITY ENTROPY (LAPE) SCORE

In our analysis we utilize the LAPE score introduced by Tang et al. (2024) as a measure for language-specific neuron activations in the FFN. Let the FFN of the $i$-th layer of a Llama-3 model be described by the following formula:

$$\boldsymbol{h}^{(i)} = (SwiGLU(\boldsymbol{h}_{attn}^{(i)} \cdot \boldsymbol{W}_1^{(i)}) \otimes \boldsymbol{W}_2^{(i)}) \cdot \boldsymbol{W}_3^{(i)}, \tag{10}$$

where $\boldsymbol{h}^{(i)} \in \mathbb{R}^d$ depicts the hidden state and $\boldsymbol{h}_{attn}^{(i)}$ the attention output of layer $i$ of a specific token, with $\boldsymbol{W}_1^{(i)}, \boldsymbol{W}_2^{(i)} \in \mathbb{R}^{d \times 3.5d}$ as up and gate projection matrices, $\boldsymbol{W}_3^{(i)} \in \mathbb{R}^{3.5d \times d}$ as down projection matrix and $\otimes$ as element-wise multiplication.

For simplicity Tang et al. (2024) consider the $j$-th neuron of the FFN to be **activated** if the corresponding output of the SwiGLU activation function is greater than zero. Then, the **activation probability** of the $j$-th neuron in the $i$-th layer for input prompts in language $k$ is defined as:

$$p_{i,j}^{(k)} = \mathbb{E}(\mathbf{1}(SwiGLU(h_{attn}^{(i)} \cdot \boldsymbol{W}_1^{(i)}))_j > 0 \mid \text{language } k), \tag{11}$$

with $\mathbf{1}$ as the identity function to count the times the neuron is activated and subsequently estimate the likelihood of neuron activation. Repeating this process for all languages and appyling L1 normalization yields the distribution $\widetilde{\boldsymbol{p}}_{i,j} = (p_{i,j}^{(1)}, \ldots, p_{i,j}^{(k)}, \ldots, p_{i,j}^{(L)}) \div ||(p_{i,j}^{(1)}, \ldots, p_{i,j}^{(k)}, \ldots, p_{i,j}^{(L)})||_1$. Finally, this allows for computing the **language activation probability entropy**:

$$LAPE_{i,j} = \sum_{k=1}^{L} \widetilde{p}_{i,j}^{(k)} \cdot log(\widetilde{p}_{i,j}^{(k)}) \tag{12}$$

A low LAPE score denotes neurons that are activated more often for inputs in specific languages than for all other languages.

Unlike Tang et al. (2024), who focus on the lowest 1% of LAPE scores and exclude those with negligible activation probability, our analysis seeks broader trends. Hence, by neglecting any thresholds we relax the original definition.

## E SUPPLEMENTARY TEST RESULTS

| | | | PPL | | | | | | | SNR | | | | | | | Pruning Error ×10⁶ | | | | | | |
|---|---|---|---|---|---|---|---|---|---|---|---|---|---|---|---|---|---|---|---|---|---|---|---|
| | | | AR | DE | EN | ES | RU | SW | ZH | AR | DE | EN | ES | RU | SW | ZH | AR | DE | EN | ES | RU | SW | ZH |
| Llama-3-8B-Instruct Wanda | AR | | 26.3 | 17.3 | 15.9 | 20.0 | 14.0 | 30.4 | 24.9 | 7.0 | 7.8 | 7.2 | 7.1 | 8.5 | 5.9 | 9.7 | 85.0 | 87.0 | 100.6 | 86.2 | 119.6 | 113.9 | 86.7 |
| | DE | | 32.4 | 15.0 | 14.5 | 17.7 | 13.5 | 30.6 | 23.9 | 6.5 | 8.4 | 8.0 | 7.3 | 8.5 | 5.8 | 9.7 | 94.1 | 77.0 | 77.7 | 81.4 | 130.1 | 117.0 | 89.3 |
| | EN | | 39.1 | 17.6 | 14.5 | 18.9 | 15.8 | 35.5 | 26.0 | 6.1 | 7.6 | 8.2 | 7.1 | 8.3 | 5.3 | 10.6 | 103.5 | 92.8 | 76.0 | 85.0 | 127.1 | 130.6 | 62.6 |
| | ES | | 33.7 | 16.4 | 14.7 | 16.6 | 13.9 | 29.6 | 24.9 | 6.3 | 7.7 | 8.0 | 7.4 | 8.2 | 5.7 | 9.2 | 98.8 | 92.0 | 79.3 | 81.8 | 139.0 | 119.3 | 107.7 |
| | RU | | 29.3 | 16.0 | 14.8 | 18.2 | 12.5 | 30.5 | 23.6 | 6.8 | 8.1 | 8.1 | 7.5 | 10.4 | 5.8 | 10.0 | 87.3 | 82.6 | 76.9 | 76.5 | 77.8 | 66.2 | 84.3 |
| | SW | | 32.3 | 17.5 | 15.8 | 19.0 | 15.0 | 24.0 | 26.3 | 6.5 | 7.7 | 7.5 | 7.0 | 8.3 | 6.7 | 9.6 | 92.8 | 89.9 | 88.3 | 87.2 | 127.3 | 94.9 | 87.0 |
| | ZH | | 37.9 | 20.1 | 17.2 | 21.2 | 16.8 | 38.9 | 24.4 | 6.0 | 7.1 | 7.7 | 6.6 | 8.0 | 4.9 | 10.0 | 105.4 | 103.2 | 83.1 | 94.0 | 131.2 | 142.5 | 87.3 |
| SparseGPT | AR | | 17.2 | 15.5 | 14.0 | 17.2 | 12.3 | 24.5 | 22.8 | 11.0 | 9.0 | 8.7 | 8.6 | 11.2 | 7.3 | 11.8 | 34.7 | 65.4 | 67.8 | 58.9 | 52.9 | 84.1 | 46.4 |
| | DE | | 30.7 | 12.1 | 13.3 | 15.8 | 11.9 | 25.5 | 23.6 | 7.4 | 11.6 | 8.1 | 9.2 | 11.6 | 7.2 | 11.8 | 76.0 | 37.3 | 118.6 | 52.6 | 48.8 | 87.7 | 46.1 |
| | EN | | 34.4 | 14.7 | 13.0 | 17.3 | 13.6 | 28.6 | 23.1 | 7.0 | 9.4 | 11.4 | 8.6 | 11.0 | 6.6 | 11.9 | 83.6 | 59.9 | 37.4 | 59.9 | 57.0 | 99.3 | 46.0 |
| | ES | | 32.4 | 14.3 | 13.6 | 14.2 | 13.0 | 26.3 | 25.7 | 7.4 | 9.6 | 8.5 | 10.9 | 11.4 | 7.1 | 11.6 | 76.0 | 57.6 | 83.6 | 36.4 | 52.7 | 89.8 | 48.7 |
| | RU | | 27.4 | 14.5 | 13.9 | 16.7 | 9.9 | 25.6 | 22.8 | 7.8 | 9.5 | 8.7 | 8.9 | 13.8 | 7.1 | 12.1 | 70.4 | 58.3 | 73.6 | 55.6 | 31.1 | 89.0 | 43.0 |
| | SW | | 24.8 | 15.7 | 13.9 | 17.0 | 14.2 | 14.0 | 25.5 | 8.0 | 9.0 | 8.2 | 8.5 | 10.8 | 11.4 | 11.1 | 67.5 | 66.3 | 80.9 | 61.7 | 59.5 | 33.7 | 54.7 |
| | ZH | | 31.1 | 16.8 | 14.3 | 18.4 | 13.9 | 27.1 | 17.2 | 7.2 | 8.6 | 8.6 | 8.0 | 10.8 | 6.0 | 15.7 | 79.9 | 71.5 | 68.4 | 60.3 | 97.9 | 68.3 | 19.5 |
| Aya-23-8B Wanda | AR | | 11.1 | 13.6 | 11.5 | 16.2 | 15.4 | 53.8 | 19.5 | 6.2 | 6.4 | 6.8 | 6.4 | 6.5 | 6.1 | 5.8 | 676.7 | 1033.2 | 851.7 | 929.5 | 911.4 | 1023.1 | 420.0 |
| | DE | | 13.4 | 12.6 | 11.4 | 16.0 | 15.4 | 54.5 | 20.2 | 6.0 | 6.7 | 6.9 | 6.6 | 6.6 | 6.2 | 5.8 | 656.4 | 992.2 | 829.0 | 897.5 | 888.2 | 994.9 | 418.2 |
| | EN | | 14.2 | 13.7 | 11.3 | 16.5 | 17.2 | 55.8 | 20.4 | 5.9 | 6.5 | 7.0 | 6.6 | 6.4 | 6.1 | 5.7 | 660.8 | 997.2 | 829.8 | 903.1 | 903.3 | 1001.8 | 426.1 |
| | ES | | 13.3 | 13.4 | 11.4 | 15.3 | 16.0 | 54.0 | 20.4 | 6.1 | 6.6 | 7.0 | 6.8 | 6.6 | 6.3 | 5.8 | 646.2 | 979.3 | 819.5 | 873.8 | 869.0 | 973.3 | 411.4 |
| | RU | | 12.8 | 13.4 | 11.5 | 16.1 | 14.1 | 55.2 | 19.7 | 6.2 | 6.7 | 7.0 | 6.8 | 6.8 | 6.4 | 5.9 | 636.5 | 955.9 | 808.8 | 853.0 | 859.1 | 951.0 | 404.5 |
| | SW | | 13.6 | 14.7 | 12.1 | 17.4 | 18.0 | 48.6 | 21.4 | 5.9 | 6.5 | 6.7 | 6.3 | 6.3 | 6.3 | 5.6 | 680.2 | 1049.4 | 850.9 | 949.4 | 932.0 | 1043.1 | 430.3 |
| | ZH | | 13.4 | 14.4 | 11.8 | 17.3 | 17.1 | 57.2 | 18.4 | 5.9 | 6.3 | 6.6 | 6.3 | 6.6 | 6.0 | 6.1 | 684.8 | 1036.2 | 842.0 | 923.0 | 930.2 | 1030.3 | 416.3 |
| SparseGPT | AR | | 9.4 | 13.8 | 11.4 | 16.3 | 15.0 | 52.2 | 19.6 | 17.1 | 15.1 | 14.7 | 12.8 | 15.6 | 15.3 | 11.2 | 30.7 | 69.6 | 75.5 | 135.4 | 55.0 | 64.1 | 79.0 |
| | DE | | 13.4 | 11.4 | 11.1 | 15.3 | 15.0 | 53.8 | 20.3 | 11.7 | 17.9 | 15.4 | 14.4 | 16.1 | 15.5 | 9.0 | 108.2 | 36.8 | 59.4 | 83.9 | 49.5 | 62.2 | 153.3 |
| | EN | | 14.6 | 13.3 | 10.8 | 16.2 | 17.4 | 54.7 | 20.7 | 11.8 | 16.2 | 17.9 | 13.9 | 15.4 | 15.4 | 9.9 | 101.7 | 54.0 | 33.7 | 93.0 | 57.4 | 62.8 | 114.6 |
| | ES | | 13.2 | 12.7 | 11.1 | 13.9 | 16.1 | 53.3 | 20.6 | 12.0 | 16.1 | 15.3 | 18.0 | 15.8 | 15.6 | 9.1 | 101.4 | 55.7 | 65.1 | 34.2 | 52.1 | 59.6 | 147.2 |
| | RU | | 12.7 | 13.1 | 11.3 | 16.0 | 12.2 | 54.6 | 19.8 | 12.2 | 15.3 | 14.2 | 13.6 | 18.4 | 15.2 | 9.4 | 97.8 | 68.1 | 91.3 | 104.2 | 30.1 | 66.8 | 139.9 |
| | SW | | 11.7 | 13.7 | 11.5 | 16.2 | 16.3 | 36.9 | 21.1 | 12.4 | 15.7 | 15.2 | 14.0 | 15.4 | 19.5 | 8.5 | 93.4 | 60.2 | 63.4 | 91.9 | 58.1 | 26.3 | 178.2 |
| | ZH | | 13.3 | 14.4 | 11.9 | 17.7 | 16.9 | 57.1 | 15.6 | 12.5 | 14.7 | 13.9 | 13.3 | 15.1 | 14.7 | 16.8 | 86.2 | 78.7 | 95.3 | 107.4 | 61.4 | 76.1 | 22.4 |

Table 6: Language-specific perplexity (PPL), Signal-to-noise ratio (SNR) and pruning error for pruned Llama 3 8B and Aya 23 8B models, averaged over three pruning runs. The leftmost columns show the model, the pruning technique, and the calibration language. For PPL and pruning errors, the smaller the value (the darker), the better; while for SNR, the greater the value (the lighter), the better. *Note the diverging color pattern for the Pruning Errors of the Aya-23 8B model pruned with Wanda. Even after repeating the evaluation several times the results look identical. We leave them in for transparency.*

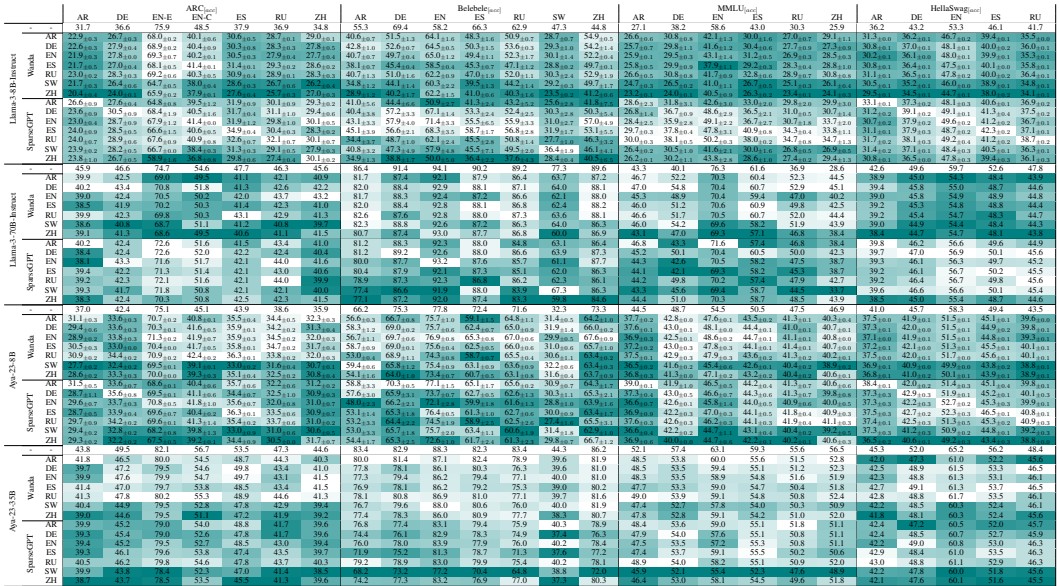

Table 7: Evaluation task performance of different model sizes (Llama-3 8B/70B and Aya-23 8B/35B) pruned for 50% unstructured sparsity. Due to resource constraints, only the 8B model variants were pruned with multiple seeds. The results are averaged over three pruning runs, with standard deviation given in the subscript. The leftmost columns show the model, the pruning technique, and the calibration language used for pruning. A "-" indicates the unpruned reference model. Each column shows the perplexity score of the pruned models on a specific evaluation language. For evaluation in English we use the original datasets (e.g. ARC-c & ARC-e), for all other languages the translated version from Dac Lai et al. (2023) is utilized.

| | | | ARC[acc] | | | | | | | Belebele[acc] | | | | | | | MMLU[acc] | | | | | | HellaSwag[acc] | | | | |
|---|---|---|---|---|---|---|---|---|---|---|---|---|---|---|---|---|---|---|---|---|---|---|---|---|---|---|---|
| | | | AR | DE | EN-E | EN-C | ES | RU | ZH | AR | DE | EN | ES | RU | SW | ZH | AR | DE | EN | ES | RU | ZH | AR | DE | EN | ES | RU |
| Llama-3-8B-Instruct | | - | 31.7 | 36.6 | 75.9 | 48.5 | 37.9 | 36.9 | 34.8 | 55.3 | 69.4 | 58.2 | 66.3 | 62.9 | 47.3 | 44.8 | 27.1 | 38.2 | 58.6 | 43.0 | 30.3 | 25.9 | 36.2 | 43.2 | 53.3 | 46.1 | 41.7 |
| | Wanda | AR-DE | 22.8 | 27.3 | 68.6 | 40.7 | 31.2 | 28.8 | 28.4 | 41.1 | 50.3 | 63.9 | 47.3 | 50.9 | 29.6 | 52.9 | 26.4 | 29.9 | 40.2 | 29.8 | 27.6 | 28.3 | 31.2 | 36.6 | 47.7 | 39.9 | 35.9 |
| | | AR-EN | 22.6 | 28.1 | 69.2 | 41.0 | 31.1 | 28.3 | 27.9 | 41.9 | 50.0 | 63.7 | 48.2 | 49.9 | 31.5 | 54.3 | 27.0 | 30.8 | 41.0 | 30.7 | 27.9 | 30.2 | 31.2 | 36.4 | 47.8 | 39.7 | 35.6 |
| | | AR-ES | 22.2 | 26.8 | 68.1 | 40.6 | 30.4 | 28.7 | 28.3 | 40.1 | 49.8 | 61.9 | 46.6 | 49.4 | 29.9 | 53.1 | 26.4 | 31.1 | 39.6 | 28.4 | 28.4 | 29.4 | 31.3 | 36.1 | 47.4 | 39.8 | 35.7 |
| | | AR-RU | 23.4 | 27.4 | 68.7 | 40.1 | 31.0 | 28.1 | 28.3 | 38.4 | 49.0 | 61.8 | 46.0 | 49.0 | 28.5 | 52.4 | 26.1 | 30.0 | 39.0 | 29.0 | 27.6 | 29.1 | 31.5 | 36.4 | 47.4 | 39.7 | 36.0 |
| | | DE-EN | 22.6 | 27.5 | 68.8 | 40.7 | 31.1 | 28.8 | 28.3 | 41.1 | 50.4 | 63.1 | 46.4 | 50.8 | 29.9 | 53.4 | 26.3 | 29.7 | 40.3 | 29.5 | 27.7 | 28.8 | 30.9 | 36.6 | 47.9 | 40.0 | 35.9 |
| | | DE-ES | 22.0 | 27.1 | 68.6 | 40.8 | 31.3 | 28.4 | 27.8 | 39.0 | 47.6 | 60.4 | 44.7 | 51.2 | 27.6 | 51.7 | 25.7 | 29.9 | 39.1 | 28.7 | 27.7 | 27.0 | 30.8 | 36.8 | 47.9 | 40.4 | 36.1 |
| | | DE-RU | 22.9 | 27.8 | 68.6 | 40.2 | 30.6 | 28.7 | 28.3 | 43.3 | 51.6 | 63.3 | 47.2 | 53.3 | 28.4 | 53.6 | 26.1 | 30.2 | 42.2 | 30.4 | 27.4 | 28.4 | 31.1 | 36.8 | 47.9 | 40.1 | 36.2 |
| | | EN-ES | 22.4 | 27.3 | 68.6 | 41.0 | 31.0 | 29.1 | 28.3 | 42.0 | 49.2 | 62.4 | 45.9 | 50.1 | 30.1 | 52.8 | 26.3 | 30.0 | 39.8 | 29.7 | 28.0 | 29.1 | 31.0 | 36.7 | 47.7 | 40.0 | 35.8 |
| | | EN-RU | 22.5 | 28.0 | 69.1 | 40.8 | 30.1 | 28.4 | 28.3 | 42.0 | 49.6 | 63.8 | 48.4 | 52.3 | 30.1 | 53.2 | 26.3 | 30.4 | 40.6 | 30.1 | 28.0 | 28.5 | 30.8 | 36.5 | 47.9 | 40.0 | 36.0 |
| | | ES-RU | 22.7 | 27.6 | 68.6 | 40.7 | 31.2 | 28.8 | 28.3 | 39.6 | 48.7 | 61.3 | 45.0 | 49.4 | 29.2 | 52.1 | 26.2 | 29.7 | 39.2 | 29.5 | 27.8 | 28.9 | 31.1 | 36.5 | 47.7 | 40.0 | 36.0 |
| | | Five languages | 22.7 | 27.5 | 68.6 | 40.9 | 31.4 | 29.1 | 28.2 | 40.1 | 49.6 | 62.5 | 45.1 | 49.7 | 30.0 | 52.8 | 26.3 | 29.7 | 39.3 | 28.9 | 27.6 | 28.8 | 31.1 | 36.4 | 47.7 | 39.9 | 35.9 |
| | | Seven languages | 22.3 | 27.3 | 68.3 | 40.6 | 31.2 | 28.7 | 28.3 | 39.7 | 49.4 | 61.6 | 44.6 | 49.3 | 30.5 | 52.9 | 26.3 | 29.2 | 39.1 | 28.8 | 27.4 | 28.7 | 31.0 | 36.3 | 47.4 | 39.7 | 35.7 |
| | SparseGPT | AR-DE | 26.3 | 30.4 | 68.5 | 41.1 | 34.0 | 31.4 | 29.8 | 46.0 | 54.5 | 62.0 | 48.7 | 52.2 | 32.1 | 51.5 | 31.3 | 38.0 | 45.8 | 38.2 | 33.6 | 34.0 | 32.9 | 38.6 | 48.9 | 41.6 | 37.7 |
| | | AR-EN | 26.5 | 28.5 | 65.7 | 41.2 | 32.5 | 30.9 | 30.1 | 46.5 | 54.7 | 61.2 | 50.6 | 50.3 | 29.1 | 50.8 | 29.3 | 33.7 | 46.6 | 35.4 | 30.2 | 31.4 | 33.1 | 37.8 | 48.9 | 41.3 | 37.1 |
| | | AR-ES | 26.2 | 27.6 | 66.8 | 40.6 | 33.6 | 30.4 | 28.5 | 47.6 | 54.0 | 63.1 | 51.7 | 53.9 | 30.4 | 53.0 | 31.5 | 35.8 | 48.3 | 37.7 | 33.9 | 34.0 | 32.8 | 37.6 | 48.6 | 42.0 | 37.1 |
| | | AR-RU | 25.5 | 28.7 | 65.9 | 39.9 | 31.9 | 31.1 | 29.7 | 41.4 | 44.9 | 56.4 | 46.9 | 46.0 | 29.1 | 40.8 | 30.3 | 33.1 | 45.7 | 34.8 | 31.6 | 31.4 | 33.2 | 37.9 | 48.8 | 41.3 | 38.3 |
| | | DE-EN | 25.1 | 30.0 | 68.4 | 40.9 | 34.0 | 31.4 | 30.1 | 44.4 | 54.2 | 62.1 | 48.3 | 51.9 | 32.4 | 52.3 | 29.7 | 37.3 | 45.9 | 36.8 | 33.0 | 33.8 | 32.2 | 38.7 | 49.1 | 41.8 | 37.7 |
| | | DE-ES | 24.0 | 30.4 | 67.8 | 40.5 | 33.1 | 30.0 | 29.3 | 42.9 | 53.8 | 64.3 | 52.1 | 51.6 | 31.4 | 51.6 | 29.3 | 37.1 | 46.9 | 38.3 | 33.6 | 33.4 | 31.2 | 39.1 | 49.0 | 42.2 | 37.6 |
| | | DE-RU | 24.2 | 29.7 | 67.9 | 41.3 | 32.4 | 31.7 | 29.4 | 39.5 | 53.8 | 67.5 | 50.9 | 51.3 | 30.0 | 49.6 | 29.7 | 38.2 | 50.5 | 38.5 | 34.2 | 34.7 | 31.7 | 39.0 | 49.3 | 41.6 | 38.3 |
| | | EN-ES | 25.2 | 29.8 | 68.1 | 41.0 | 34.6 | 31.1 | 29.9 | 43.8 | 54.4 | 64.6 | 51.5 | 52.2 | 31.7 | 52.3 | 29.8 | 36.8 | 46.3 | 37.8 | 33.0 | 34.0 | 32.2 | 38.3 | 49.1 | 42.1 | 37.6 |
| | | EN-RU | 23.3 | 30.7 | 67.5 | 41.3 | 32.1 | 32.2 | 30.0 | 42.4 | 51.9 | 66.7 | 51.2 | 56.7 | 31.0 | 53.3 | 29.5 | 35.8 | 48.6 | 36.2 | 33.2 | 33.7 | 31.7 | 38.1 | 49.5 | 41.3 | 38.2 |
| | | ES-RU | 25.5 | 29.5 | 67.5 | 40.7 | 34.3 | 31.5 | 29.8 | 42.3 | 50.5 | 60.4 | 48.6 | 51.3 | 31.7 | 47.5 | 30.5 | 36.8 | 45.5 | 38.6 | 33.7 | 33.4 | 32.4 | 38.2 | 48.9 | 42.0 | 38.0 |
| | | Five languages | 26.1 | 29.9 | 67.9 | 40.9 | 34.6 | 31.5 | 29.9 | 45.1 | 53.0 | 60.9 | 48.0 | 51.4 | 33.5 | 50.6 | 30.9 | 37.4 | 44.6 | 38.0 | 33.6 | 34.0 | 32.8 | 38.5 | 48.9 | 41.9 | 37.9 |
| | | Seven languages | 25.5 | 29.9 | 67.3 | 40.4 | 33.8 | 30.8 | 30.0 | 44.9 | 53.0 | 57.7 | 47.1 | 50.6 | 34.5 | 51.6 | 30.1 | 36.2 | 43.4 | 36.3 | 32.3 | 32.5 | 32.6 | 38.5 | 48.9 | 41.9 | 37.9 |
| Aya-23-8B | | - | 37.0 | 42.4 | 75.1 | 45.1 | 43.9 | 38.6 | 35.9 | 66.2 | 75.3 | 77.8 | 72.4 | 71.6 | 32.3 | 73.3 | 44.5 | 48.7 | 54.5 | 50.5 | 47.5 | 46.9 | 41.0 | 45.7 | 58.3 | 49.4 | 43.5 |
| | Wanda | AR-DE | 31.1 | 33.3 | 70.6 | 41.6 | 36.0 | 34.2 | 32.0 | 58.2 | 68.7 | 75.9 | 62.4 | 65.0 | 31.9 | 66.0 | 37.6 | 43.0 | 47.9 | 44.2 | 41.0 | 40.4 | 37.7 | 42.0 | 51.4 | 45.1 | 39.8 |
| | | AR-EN | 30.9 | 34.0 | 70.5 | 41.7 | 36.4 | 35.5 | 31.9 | 60.3 | 67.7 | 76.0 | 61.8 | 65.4 | 33.0 | 66.9 | 37.9 | 42.7 | 48.5 | 44.2 | 40.4 | 40.4 | 37.7 | 41.7 | 51.5 | 45.1 | 39.8 |
| | | AR-ES | 31.9 | 33.8 | 70.7 | 41.5 | 35.9 | 34.2 | 31.9 | 57.2 | 68.0 | 73.9 | 61.3 | 65.0 | 30.8 | 64.4 | 37.5 | 43.1 | 47.4 | 43.9 | 41.2 | 40.0 | 37.9 | 42.1 | 51.4 | 45.4 | 39.8 |
| | | AR-RU | 32.1 | 33.6 | 70.8 | 41.5 | 36.4 | 34.3 | 32.1 | 56.8 | 67.6 | 74.0 | 59.6 | 64.9 | 32.3 | 64.1 | 37.4 | 42.8 | 47.7 | 43.9 | 41.1 | 40.4 | 37.8 | 41.8 | 51.4 | 45.2 | 39.9 |
| | | DE-EN | 30.4 | 33.5 | 70.7 | 41.8 | 36.1 | 34.1 | 31.7 | 58.0 | 68.8 | 76.3 | 62.9 | 66.5 | 32.0 | 66.1 | 37.7 | 43.0 | 48.1 | 44.3 | 40.9 | 40.5 | 37.5 | 42.0 | 51.4 | 45.1 | 39.7 |
| | | DE-ES | 29.7 | 33.9 | 70.5 | 42.1 | 36.2 | 34.5 | 31.1 | 59.7 | 69.2 | 75.5 | 63.1 | 65.9 | 31.6 | 66.4 | 37.9 | 43.3 | 48.2 | 44.1 | 41.7 | 40.6 | 37.4 | 41.9 | 51.5 | 45.5 | 39.8 |
| | | DE-RU | 30.5 | 33.7 | 70.3 | 41.8 | 36.3 | 34.3 | 31.9 | 58.1 | 69.7 | 75.5 | 61.8 | 64.6 | 31.3 | 63.7 | 37.7 | 43.1 | 47.8 | 44.1 | 40.9 | 40.3 | 37.4 | 41.9 | 51.7 | 45.1 | 39.9 |
| | | EN-ES | 30.6 | 33.4 | 70.7 | 41.7 | 35.9 | 34.3 | 31.9 | 58.9 | 68.7 | 76.2 | 62.9 | 65.5 | 31.4 | 66.3 | 37.5 | 43.0 | 48.2 | 44.4 | 41.0 | 40.5 | 37.4 | 41.9 | 51.4 | 45.2 | 39.8 |
| | | EN-RU | 30.2 | 33.5 | 70.7 | 42.3 | 35.8 | 34.2 | 31.7 | 55.2 | 68.5 | 76.0 | 62.4 | 66.5 | 31.2 | 65.7 | 37.6 | 42.8 | 48.4 | 44.5 | 41.0 | 40.4 | 37.4 | 41.9 | 51.7 | 45.2 | 39.9 |
| | | ES-RU | 30.9 | 33.2 | 70.7 | 41.9 | 36.0 | 34.4 | 32.0 | 58.4 | 68.4 | 75.8 | 62.7 | 65.1 | 31.6 | 65.9 | 37.4 | 43.1 | 47.9 | 44.2 | 41.1 | 40.5 | 37.6 | 42.0 | 51.4 | 45.2 | 39.8 |
| | | Five languages | 31.0 | 33.2 | 70.7 | 41.7 | 36.0 | 34.1 | 32.1 | 58.6 | 68.6 | 76.4 | 63.0 | 65.0 | 32.0 | 66.3 | 37.6 | 43.0 | 48.0 | 44.3 | 41.0 | 40.6 | 37.6 | 41.9 | 51.3 | 45.1 | 39.7 |
| | | Seven languages | 31.0 | 33.0 | 70.8 | 42.0 | 36.1 | 33.9 | 32.0 | 59.0 | 68.1 | 76.4 | 63.7 | 65.0 | 32.6 | 67.4 | 37.5 | 42.8 | 47.9 | 44.2 | 41.0 | 40.6 | 37.6 | 41.9 | 51.2 | 44.9 | 39.6 |
| | SparseGPT | AR-DE | 30.9 | 34.8 | 69.4 | 40.1 | 35.6 | 32.9 | 30.6 | 59.0 | 69.4 | 75.4 | 63.1 | 64.6 | 29.7 | 65.2 | 37.9 | 42.8 | 46.8 | 44.6 | 41.3 | 40.5 | 38.2 | 42.8 | 51.8 | 45.6 | 40.5 |
| | | AR-EN | 30.9 | 34.2 | 70.3 | 40.2 | 35.9 | 32.2 | 30.8 | 56.5 | 65.6 | 76.0 | 64.1 | 64.6 | 29.4 | 65.4 | 37.1 | 42.5 | 47.2 | 44.4 | 41.4 | 40.5 | 38.6 | 42.4 | 52.5 | 45.5 | 40.2 |
| | | AR-ES | 30.9 | 34.7 | 69.2 | 40.0 | 34.9 | 32.2 | 31.3 | 54.8 | 67.4 | 75.6 | 61.5 | 63.6 | 29.2 | 62.6 | 37.9 | 42.3 | 47.0 | 44.3 | 41.4 | 40.0 | 38.5 | 42.6 | 51.8 | 46.2 | 40.6 |
| | | AR-RU | 31.2 | 34.4 | 69.3 | 40.0 | 35.0 | 33.1 | 31.2 | 57.6 | 68.8 | 76.6 | 64.0 | 66.7 | 31.0 | 66.2 | 38.3 | 42.0 | 47.3 | 44.4 | 41.5 | 40.9 | 38.4 | 42.3 | 51.2 | 44.9 | 40.6 |
| | | DE-EN | 30.0 | 34.6 | 69.8 | 40.8 | 35.4 | 32.8 | 30.4 | 58.0 | 68.5 | 75.2 | 63.1 | 64.3 | 29.7 | 65.0 | 37.1 | 42.8 | 46.6 | 44.5 | 41.1 | 40.2 | 37.8 | 42.8 | 52.1 | 45.7 | 40.5 |
| | | DE-ES | 29.3 | 34.4 | 69.4 | 41.1 | 34.9 | 32.4 | 30.9 | 53.4 | 67.0 | 71.5 | 61.3 | 60.7 | 27.7 | 60.5 | 37.2 | 43.4 | 46.4 | 44.0 | 41.4 | 39.9 | 37.5 | 43.3 | 52.3 | 46.1 | 40.4 |
| | | DE-RU | 29.5 | 34.5 | 69.5 | 41.4 | 35.2 | | 30.7 | 58.7 | 69.2 | 74.3 | 64.5 | 55.9 | 28.2 | 66.3 | 37.5 | 42.7 | 47.4 | 44.1 | 40.1 | 37.4 | 42.7 | 51.9 | 45.3 | 40.7 |
| | | EN-ES | 29.8 | 34.7 | 69.9 | 40.6 | 35.6 | 32.9 | 30.5 | 55.8 | 68.7 | 74.7 | 62.5 | 64.1 | 30.0 | 64.2 | 36.9 | 42.7 | 46.8 | 44.5 | 41.1 | 40.3 | 38.0 | 42.5 | 52.1 | 45.9 | 40.5 |
| | | EN-RU | 29.5 | 34.0 | 70.6 | 40.9 | 36.2 | 33.6 | 31.2 | 52.0 | 66.1 | 74.9 | 59.8 | 63.4 | 28.9 | 62.9 | 37.0 | 42.5 | 46.7 | 45.5 | 41.8 | 40.2 | 37.4 | 42.5 | 52.3 | 45.7 | 40.5 |
| | | ES-RU | 30.0 | 35.0 | 69.7 | 40.6 | 35.9 | 33.0 | 30.6 | 56.3 | 67.9 | 74.4 | 62.7 | 64.6 | 29.9 | 64.1 | 37.0 | 42.7 | 46.6 | 44.6 | 41.4 | 40.4 | 38.1 | 42.8 | 51.6 | 46.1 | 40.8 |
| | | Five languages | 30.4 | 34.9 | 69.6 | 40.2 | 35.7 | 33.2 | 30.5 | 59.0 | 69.1 | 75.4 | 63.0 | 64.9 | 30.4 | 65.8 | 37.3 | 42.6 | 46.9 | 44.6 | 41.2 | 40.4 | 38.1 | 42.8 | 51.8 | 45.8 | 40.6 |
| | | Seven languages | 30.4 | 34.7 | 69.2 | 40.0 | 35.4 | 33.0 | 30.9 | 57.9 | 68.6 | 74.7 | 64.2 | 65.9 | 31.3 | 67.4 | 37.4 | 42.5 | 46.6 | 44.4 | 41.2 | 40.7 | 38.1 | 42.8 | 51.7 | 45.9 | 40.5 |

Table 8: Pruned Model Performance averaged over three pruning runs. The lighter, the less drop the better. Five languages: AR, DE, EN, ES, RU. Seven languages: AR, DE, EN, ES, RU, SW, ZH.

| | | | ARC[acc] | | | | | | | Belebele[acc] | | | | | | | MMLU[acc] | | | | | | HellaSwag[acc] | | | | |
|---|---|---|---|---|---|---|---|---|---|---|---|---|---|---|---|---|---|---|---|---|---|---|---|---|---|---|---|
| | | | AR | DE | EN-E | EN-C | ES | RU | ZH | AR | DE | EN | ES | RU | SW | ZH | AR | DE | EN | ES | RU | ZH | AR | DE | EN | ES | RU |
| Llama-3-8B-Instruct | | - | 31.7 | 36.6 | 75.9 | 48.5 | 37.9 | 36.9 | 34.8 | 55.3 | 69.4 | 58.2 | 66.3 | 62.9 | 47.3 | 44.8 | 27.1 | 38.2 | 58.6 | 43.0 | 30.3 | 25.9 | 36.2 | 43.2 | 53.3 | 46.1 | 41.7 |
| | Wanda | AR | 20.6 | 18.6 | 50.3 | 27.7 | 20.6 | 19.4 | 22.2 | 23.2 | 23.0 | 28.1 | 22.8 | 22.7 | 23.3 | 22.9 | 23.4 | 24.0 | 24.9 | 24.8 | 23.5 | 24.5 | 27.3 | 28.3 | 36.1 | 30.6 | 28.5 |
| | | DE | 19.8 | 18.7 | 48.7 | 27.0 | 21.1 | 18.4 | 21.1 | 22.9 | 22.7 | 25.1 | 22.4 | 22.8 | 22.9 | 22.8 | 23.7 | 23.6 | 25.3 | 24.1 | 22.9 | 23.1 | 26.7 | 29.0 | 36.6 | 30.7 | 28.7 |
| | | EN | 19.5 | 18.0 | 49.5 | 28.6 | 20.9 | 19.2 | 22.1 | 22.8 | 23.0 | 29.9 | 25.0 | 24.3 | 23.4 | 25.9 | 23.8 | 25.0 | 27.9 | 26.5 | 24.2 | 24.6 | 26.5 | 28.3 | 36.9 | 30.4 | 28.3 |
| | | ES | 20.5 | 19.4 | 49.3 | 28.2 | 22.9 | 18.4 | 21.9 | 23.1 | 22.6 | 28.2 | 23.6 | 23.0 | 24.4 | 23.6 | 23.0 | 24.0 | 25.4 | 23.0 | 23.8 | 23.7 | 27.0 | 28.5 | 36.9 | 31.7 | 28.3 |
| | | RU | 21.3 | 18.9 | 49.5 | 28.4 | 22.4 | 20.4 | 21.7 | 22.9 | 22.6 | 25.7 | 22.2 | 22.9 | 23.7 | 23.1 | 22.7 | 22.9 | 25.1 | 23.2 | 22.7 | 22.9 | 26.9 | 28.9 | 37.4 | 31.3 | 29.1 |
| | | SW | 21.1 | 18.0 | 48.3 | 27.7 | 20.3 | 19.4 | 21.8 | 23.3 | 23.8 | 25.9 | 24.0 | 22.9 | 23.0 | 23.6 | 23.0 | 23.7 | 26.0 | 23.0 | 23.2 | 23.0 | 26.4 | 28.2 | 35.2 | 29.8 | 27.8 |
| | | ZH | 19.6 | 18.1 | 49.5 | 26.0 | 21.0 | 19.3 | 22.6 | 22.9 | 22.9 | 24.0 | 22.8 | 22.9 | 23.1 | 22.3 | 22.7 | 22.8 | 24.8 | 23.0 | 22.7 | 23.0 | 26.5 | 28.0 | 35.9 | 29.5 | 28.1 |
| | SparseGPT | AR | 22.8 | 20.7 | 55.6 | 32.1 | 23.8 | 22.5 | 23.2 | 29.1 | 28.7 | 37.7 | 32.3 | 36.4 | 26.9 | 30.6 | 26.3 | 26.5 | 30.3 | 26.1 | 25.7 | 25.4 | 30.1 | 30.9 | 40.2 | 33.8 | 31.1 |
| | | DE | 20.9 | 24.7 | 60.5 | 32.8 | 24.9 | 22.0 | 24.4 | 23.0 | 26.6 | 38.9 | 24.6 | 23.3 | 22.9 | 23.6 | 22.8 | 25.1 | 32.5 | 24.1 | 22.8 | 23.0 | 27.7 | 34.0 | 41.1 | 34.1 | 31.3 |
| | | EN | 20.4 | 21.7 | 58.6 | 33.3 | 24.4 | 20.7 | 24.7 | 23.6 | 28.7 | 45.0 | 28.8 | 23.6 | 23.2 | 29.2 | 23.0 | 24.7 | 36.9 | 25.0 | 23.8 | 24.3 | 26.8 | 30.3 | 41.4 | 33.2 | 29.6 |
| | | ES | 19.8 | 21.0 | 58.5 | 32.8 | 27.4 | 22.1 | 23.6 | 22.9 | 24.2 | 43.1 | 26.3 | 24.2 | 23.1 | 23.6 | 22.8 | 24.2 | 35.3 | 25.2 | 23.2 | 23.6 | 27.6 | 31.3 | 41.5 | 36.4 | 31.0 |
| | | RU | 22.0 | 21.6 | 59.8 | 34.0 | 24.7 | 25.8 | 25.6 | 25.0 | 33.9 | 45.8 | 36.0 | 33.7 | 23.4 | 26.6 | 23.6 | 29.0 | 35.0 | 28.9 | 26.0 | 25.1 | 27.0 | 32.1 | 41.8 | 35.2 | 34.0 |
| | | SW | 18.9 | 20.1 | 55.7 | 32.2 | 22.8 | 20.1 | 22.6 | 23.3 | 23.7 | 38.9 | 24.6 | 24.4 | 24.7 | 23.2 | 23.2 | 25.7 | 35.7 | 26.0 | 24.2 | 23.5 | 27.5 | 30.0 | 39.9 | 32.9 | 29.7 |
| | | ZH | 20.1 | 19.1 | 53.8 | 31.0 | 21.7 | 19.8 | 24.4 | 24.3 | 28.9 | 28.3 | 27.4 | 28.3 | 23.9 | 27.9 | 24.7 | 24.8 | 25.5 | 24.8 | 25.2 | 25.3 | 26.9 | 29.1 | 39.3 | 31.5 | 29.8 |
| Aya-23-8B | | - | 37.0 | 42.4 | 75.1 | 45.1 | 43.9 | 38.6 | 35.9 | 66.2 | 75.3 | 77.8 | 72.4 | 71.6 | 32.3 | 73.3 | 44.5 | 48.7 | 54.5 | 50.5 | 47.5 | 46.9 | 41.0 | 45.7 | 58.3 | 49.4 | 43.5 |
| | Wanda | AR | 26.5 | 26.5 | 62.9 | 33.7 | 29.0 | 25.4 | 27.9 | 30.9 | 38.9 | 42.9 | 33.2 | 33.8 | 24.9 | 34.2 | 29.7 | 30.1 | 34.7 | 32.3 | 30.3 | 30.8 | 32.9 | 35.1 | 43.2 | 38.1 | 33.8 |
| | | DE | 25.1 | 26.7 | 63.6 | 34.7 | 28.6 | 24.8 | 26.6 | 33.9 | 41.1 | 44.2 | 33.7 | 38.1 | 25.9 | 35.4 | 29.2 | 31.0 | 34.9 | 31.4 | 31.1 | 30.6 | 31.2 | 36.7 | 43.4 | 38.2 | 34.2 |
| | | EN | 24.8 | 27.5 | 64.9 | 35.4 | 28.8 | 25.4 | 26.6 | 30.8 | 38.3 | 40.6 | 33.7 | 36.1 | 25.0 | 32.1 | 27.6 | 30.6 | 33.7 | 30.9 | 29.3 | 30.2 | 31.1 | 35.3 | 43.5 | 37.8 | 33.2 |
| | | ES | 25.4 | 26.8 | 64.2 | 34.4 | 29.5 | 25.9 | 27.4 | 27.1 | 35.4 | 36.7 | 31.8 | 32.6 | 23.0 | 32.2 | 28.0 | 29.6 | 33.8 | 30.4 | 30.5 | 29.3 | 31.3 | 35.5 | 43.0 | 39.1 | 34.3 |
| | | RU | 25.2 | 26.4 | 64.7 | 34.8 | 29.1 | 26.8 | 28.4 | 32.8 | 37.2 | 45.0 | 34.6 | 37.2 | 26.3 | 34.7 | 27.3 | 28.6 | 34.1 | 30.3 | 27.9 | 29.9 | 31.9 | 35.6 | 43.2 | 38.4 | 35.2 |
| | | SW | 23.2 | 26.6 | 61.6 | 33.8 | 30.0 | 25.1 | 26.1 | 29.7 | 36.7 | 41.9 | 31.1 | 33.2 | 25.2 | 32.4 | 27.8 | 28.9 | 33.7 | 29.6 | 29.1 | 29.7 | 30.6 | 34.6 | 41.8 | 37.4 | 33.5 |
| | | ZH | 25.5 | 26.0 | 63.2 | 33.3 | 29.7 | 24.4 | 27.2 | 29.7 | 33.8 | 36.4 | 29.3 | 32.5 | 25.9 | 31.9 | 28.2 | 29.6 | 32.3 | 28.5 | 29.0 | 28.9 | 31.1 | 34.3 | 42.5 | 37.1 | 33.1 |
| | SparseGPT | AR | 25.2 | 24.7 | 57.7 | 30.0 | 26.7 | 23.4 | 24.3 | 37.2 | 37.0 | 46.3 | 35.3 | 36.6 | 22.2 | 31.2 | 32.2 | 27.2 | 32.6 | 31.4 | 30.5 | 30.6 | 34.8 | 31.9 | 40.1 | 35.6 | 32.2 |
| | | DE | 22.8 | 27.2 | 59.3 | 30.8 | 27.6 | 22.4 | 25.1 | 29.4 | 32.4 | 38.8 | 36.0 | 35.7 | 22.7 | 33.4 | 27.7 | 30.3 | 31.9 | 30.0 | 29.6 | 28.4 | 28.5 | 36.6 | 41.4 | 36.8 | 33.0 |
| | | EN | 22.8 | 23.4 | 59.9 | 32.3 | 26.3 | 21.8 | 24.4 | 26.6 | 34.6 | 50.3 | 36.3 | 37.0 | 25.2 | 33.9 | 27.9 | 30.7 | 33.8 | 30.7 | 30.6 | 29.2 | 26.9 | 32.4 | 42.2 | 34.9 | 30.6 |
| | | ES | 23.1 | 25.1 | 59.4 | 31.2 | 28.8 | 24.0 | 24.1 | 28.2 | 35.8 | 44.7 | 31.9 | 38.1 | 23.4 | 31.8 | 28.2 | 28.7 | 28.7 | 26.9 | 26.8 | 26.8 | 28.7 | 34.5 | 41.6 | 39.5 | 32.4 |
| | | RU | 22.0 | 25.7 | 60.8 | 31.5 | 27.7 | 26.7 | 24.6 | 31.9 | 36.0 | 42.2 | 34.4 | 38.1 | 28.9 | 35.3 | 27.7 | 30.9 | 34.7 | 30.2 | 32.1 | 29.4 | 29.5 | 33.9 | 41.4 | 36.8 | 36.4 |
| | | SW | 22.3 | 23.4 | 56.4 | 28.7 | 25.1 | 22.2 | 24.3 | 31.6 | 35.4 | 40.2 | 35.2 | 34.3 | 28.2 | 35.2 | 28.4 | 30.7 | 32.6 | 29.3 | 29.4 | 30.0 | 29.5 | 32.7 | 39.5 | 35.3 | 32.0 |
| | | ZH | 23.0 | 22.7 | 57.1 | 27.5 | 26.0 | 21.1 | 27.2 | 28.2 | 41.0 | 30.9 | 31.2 | 24.9 | 37.0 | | 27.6 | 27.2 | 31.5 | 29.2 | 27.6 | 30.6 | 29.4 | 32.4 | 39.9 | 34.6 | 31.3 |

Table 9: Downstream task performance with 50% 2:4 structured sparsity averaged over three pruning runs.

| | | | ARC[acc] | | | | | | | Belebele[acc] | | | | | | | MMLU[acc] | | | | | | HellaSwag[acc] | | | | |
|---|---|---|---|---|---|---|---|---|---|---|---|---|---|---|---|---|---|---|---|---|---|---|---|---|---|---|---|
| | | | AR | DE | EN-E | EN-C | ES | RU | ZH | AR | DE | EN | ES | RU | SW | ZH | AR | DE | EN | ES | RU | ZH | AR | DE | EN | ES | RU |
| Llama-3.1-8B-Instruct | | - | 32.8 | 36.4 | 77.9 | 51.6 | 41.1 | 36.9 | 37.5 | 73.9 | 72.7 | 82.4 | 76.3 | 76.3 | 51.8 | 75.7 | 41.1 | 47.6 | 63.6 | 53.3 | 43.5 | 49.5 | 38.2 | 45.7 | 57.7 | 48.3 | 44.1 |
| | Wanda | AR | 25.7 | 28.7 | 71.5 | 39.8 | 32.2 | 28.7 | 29.4 | 51.6 | 59.0 | 68.3 | 61.0 | 56.6 | 34.3 | 61.3 | 32.9 | 42.2 | 51.8 | 46.4 | 36.6 | 40.9 | 32.8 | 39.0 | 49.2 | 41.4 | 37.5 |
| | | DE | 26.4 | 30.5 | 71.3 | 43.0 | 32.6 | 28.7 | 31.0 | 48.4 | 58.7 | 70.9 | 61.7 | 59.0 | 34.4 | 63.7 | 31.2 | 39.1 | 52.1 | 44.3 | 34.0 | 39.2 | 32.1 | 39.3 | 50.3 | 41.9 | 37.3 |
| | | EN | 24.5 | 26.6 | 70.5 | 42.8 | 31.2 | 28.1 | 29.1 | 49.7 | 59.0 | 67.7 | 60.4 | 58.1 | 34.9 | 61.6 | 32.0 | 40.7 | 49.9 | 44.2 | 36.0 | 39.3 | 31.6 | 38.5 | 50.4 | 41.3 | 36.8 |
| | | ES | 25.6 | 28.6 | 71.3 | 41.6 | 33.7 | 28.3 | 29.1 | 48.2 | 58.6 | 68.8 | 59.4 | 56.7 | 30.7 | 61.7 | 31.8 | 41.0 | 51.7 | 44.1 | 36.0 | 39.4 | 32.1 | 39.0 | 50.2 | 41.8 | 37.4 |
| | | RU | 26.6 | 28.5 | 69.7 | 40.9 | 31.7 | 27.8 | 30.9 | 47.4 | 56.0 | 70.0 | 57.2 | 56.0 | 32.2 | 60.1 | 30.8 | 39.7 | 53.0 | 44.5 | 36.3 | 41.5 | 32.6 | 39.1 | 50.3 | 41.8 | 36.9 |
| | | SW | 24.4 | 27.3 | 71.3 | 41.7 | 31.7 | 27.9 | 29.1 | 46.1 | 50.8 | 67.7 | 58.4 | 58.0 | 34.3 | 59.8 | 31.2 | 38.6 | 50.8 | 43.1 | 30.8 | 39.4 | 32.0 | 38.2 | 49.3 | 41.1 | 36.9 |
| | | ZH | 23.4 | 27.6 | 71.0 | 39.3 | 31.8 | 27.5 | 28.4 | 47.4 | 53.2 | 64.9 | 53.9 | 56.4 | 35.1 | 58.7 | 33.7 | 38.9 | 50.2 | 41.6 | 36.1 | 40.2 | 31.5 | 37.5 | 48.1 | 39.7 | 36.0 |
| | SparseGPT | AR | 28.8 | 29.1 | 71.8 | 43.3 | 34.8 | 30.4 | 29.7 | 52.2 | 62.2 | 74.0 | 63.3 | 61.9 | 36.9 | 65.6 | 33.4 | 42.6 | 53.1 | 46.3 | 40.8 | 41.1 | 34.9 | 39.4 | 51.3 | 42.9 | 38.6 |
| | | DE | 27.3 | 32.2 | 74.3 | 43.4 | 34.4 | 30.5 | 31.3 | 51.4 | 61.9 | 72.8 | 61.3 | 58.3 | 36.0 | 64.9 | 31.5 | 40.9 | 53.0 | 44.9 | 36.1 | 42.2 | 32.6 | 40.7 | 52.1 | 43.7 | 39.3 |
| | | EN | 25.3 | 29.3 | 73.0 | 45.2 | 33.2 | 30.4 | 29.4 | 41.6 | 45.1 | 60.9 | 36.3 | 44.3 | 28.4 | 51.4 | 30.9 | 36.5 | 52.2 | 31.1 | 33.7 | 36.4 | 31.5 | 40.0 | 52.2 | 43.2 | 37.7 |
| | | ES | 25.7 | 31.3 | 73.0 | 44.7 | 37.9 | 31.9 | 31.0 | 48.0 | 59.4 | 71.1 | 62.3 | 56.3 | 35.8 | 63.5 | 31.8 | 40.1 | 53.1 | 43.5 | 34.8 | 40.5 | 32.5 | 39.9 | 52.2 | 44.4 | 39.2 |
| | | RU | 27.9 | 30.0 | 72.2 | 42.8 | 34.1 | 31.3 | 31.4 | 53.3 | 64.7 | 71.8 | 59.1 | 56.0 | 28.9 | 62.2 | 34.7 | 41.4 | 56.1 | 47.2 | 41.9 | 42.1 | 32.8 | 40.4 | 51.9 | 43.3 | 40.0 |
| | | SW | 27.6 | 30.8 | 72.7 | 43.9 | 33.3 | 30.4 | 29.7 | 50.7 | 62.8 | 75.9 | 63.7 | 56.8 | 40.3 | 61.9 | 31.9 | 43.6 | 53.9 | 46.4 | 34.8 | 41.5 | 32.8 | 39.1 | 51.4 | 42.5 | 38.1 |
| | | ZH | 27.2 | 27.8 | 69.9 | 42.7 | 30.9 | 30.8 | 31.5 | 44.4 | 53.4 | 68.8 | 56.2 | 56.7 | 36.8 | 65.2 | 31.1 | 38.6 | 49.3 | 41.2 | 34.0 | 41.2 | 32.0 | 38.6 | 50.4 | 41.6 | 37.7 |

Table 10: Downstream task performance of Llama 3.1 pruned by SparseGPT and Wanda towards 50% unstructured sparsity.

| | | | | ARC[acc] | | | | | | Belebele[acc] | | | | | | MMLU[acc] | | | | | HellaSwag[acc] | | | | |
|---|---|---|---|---|---|---|---|---|---|---|---|---|---|---|---|---|---|---|---|---|---|---|---|---|---|
| | | | AR | DE | EN-E | EN-C | ES | RU | ZH | AR | DE | EN | ES | RU | SW | ZH | AR | DE | EN | ES | RU | ZH | AR | DE | EN | ES | RU |
| **Aya-23-8B** | | - | 37.0 | 42.4 | 75.1 | 45.1 | 43.9 | 38.6 | 35.9 | 66.2 | 75.3 | 77.8 | 72.4 | 71.6 | 32.3 | 73.3 | 44.5 | 48.7 | 54.5 | 50.5 | 47.5 | 46.9 | 41.0 | 45.7 | 58.3 | 49.4 | 43.5 |
| | Wanda | AR | 36.5 | 41.7 | 75.3 | 45.3 | 42.6 | 39.4 | 36.1 | 68.0 | 76.0 | 77.2 | 71.7 | 70.7 | 32.6 | 72.9 | 43.9 | 48.5 | 53.5 | 50.5 | 47.1 | 46.7 | 40.6 | 45.1 | 57.6 | 48.7 | 43.2 |
| | | DE | 36.7 | 41.6 | 75.5 | 44.9 | 42.6 | 39.4 | 36.3 | 70.0 | 76.2 | 78.9 | 72.9 | 71.1 | 32.7 | 73.1 | 44.0 | 48.7 | 53.8 | 50.6 | 47.2 | 46.8 | 40.8 | 45.1 | 57.5 | 48.7 | 43.1 |
| | | EN | 37.0 | 41.7 | 75.5 | 46.0 | 42.3 | 39.5 | 36.2 | 69.8 | 75.4 | 78.4 | 73.0 | 71.1 | 33.1 | 73.4 | 43.9 | 48.9 | 53.6 | 50.3 | 47.3 | 46.9 | 40.9 | 45.2 | 57.4 | 48.6 | 43.2 |
| | | ES | 36.4 | 41.2 | 75.4 | 45.6 | 42.4 | 39.9 | 36.1 | 68.9 | 75.9 | 77.4 | 71.9 | 70.6 | 32.3 | 71.6 | 44.0 | 48.7 | 53.6 | 50.5 | 47.2 | 46.7 | 41.0 | 45.3 | 57.7 | 48.7 | 43.2 |
| | | RU | 36.5 | 41.4 | 75.5 | 45.4 | 42.6 | 39.4 | 36.3 | 69.0 | 75.3 | 78.0 | 72.2 | 70.2 | 32.2 | 72.9 | 43.9 | 48.6 | 53.7 | 50.3 | 47.4 | 46.6 | 40.7 | 45.0 | 57.6 | 48.6 | 43.1 |
| | | SW | 36.4 | 41.1 | 75.4 | 45.8 | 42.7 | 39.1 | 36.5 | 68.0 | 76.0 | 77.1 | 71.6 | 70.2 | 30.9 | 73.3 | 43.7 | 48.0 | 53.5 | 49.9 | 47.1 | 46.3 | 40.4 | 45.2 | 57.3 | 48.6 | 42.8 |
| | | ZH | 36.4 | 41.7 | 75.3 | 46.0 | 41.5 | 38.9 | 35.5 | 68.6 | 76.8 | 77.8 | 71.6 | 70.4 | 32.4 | 72.8 | 43.8 | 48.7 | 53.6 | 50.3 | 47.4 | 46.4 | 40.8 | 45.3 | 57.7 | 48.8 | 43.2 |
| | SparseGPT | AR | 37.5 | 42.1 | 75.7 | 46.1 | 42.3 | 39.8 | 36.1 | 70.9 | 76.1 | 78.7 | 72.9 | 72.0 | 32.3 | 73.6 | 44.4 | 48.7 | 53.5 | 50.6 | 47.1 | 46.9 | 40.7 | 44.9 | 57.5 | 48.6 | 43.1 |
| | | DE | 36.9 | 41.8 | 75.3 | 45.9 | 41.5 | 39.5 | 36.0 | 72.1 | 77.9 | 79.0 | 74.0 | 71.8 | 32.1 | 76.0 | 44.0 | 48.4 | 53.8 | 50.3 | 47.3 | 46.6 | 40.6 | 45.2 | 57.5 | 48.9 | 43.1 |
| | | EN | 37.0 | 41.3 | 75.7 | 45.7 | 42.1 | 39.1 | 36.4 | 70.2 | 76.8 | 78.8 | 73.6 | 72.6 | 31.9 | 74.7 | 44.0 | 48.7 | 54.0 | 50.5 | 47.6 | 46.8 | 40.6 | 45.0 | 57.5 | 48.7 | 42.9 |
| | | ES | 36.7 | 41.8 | 75.5 | 46.0 | 41.8 | 39.7 | 35.8 | 71.9 | 76.9 | 79.3 | 74.3 | 72.2 | 31.7 | 75.3 | 43.8 | 48.6 | 53.9 | 50.6 | 47.2 | 46.8 | 40.4 | 44.8 | 57.4 | 48.3 | 43.0 |
| | | RU | 36.6 | 41.8 | 75.0 | 44.9 | 41.9 | 39.9 | 35.8 | 71.6 | 76.9 | 79.9 | 73.8 | 72.6 | 31.7 | 75.3 | 43.9 | 48.9 | 53.8 | 50.4 | 47.4 | 46.9 | 40.7 | 44.9 | 57.6 | 48.8 | 43.0 |
| | | SW | 36.4 | 41.8 | 75.4 | 46.4 | 42.1 | 39.2 | 36.3 | 69.1 | 77.8 | 80.2 | 75.0 | 71.3 | 32.6 | 75.4 | 44.0 | 48.9 | 53.8 | 50.5 | 47.0 | 46.9 | 40.6 | 44.8 | 57.4 | 48.5 | 43.1 |
| | | ZH | 36.7 | 41.5 | 75.5 | 45.9 | 42.2 | 38.8 | 36.2 | 70.1 | 78.2 | 79.6 | 74.6 | 72.6 | 31.9 | 76.3 | 43.6 | 48.3 | 54.1 | 50.1 | 47.1 | 46.3 | 40.5 | 45.0 | 57.3 | 48.4 | 43.1 |
| **Aya-23-35B** | | - | 43.8 | 49.5 | 82.1 | 56.7 | 53.5 | 47.3 | 44.6 | 83.4 | 82.9 | 88.3 | 82.3 | 83.4 | 44.3 | 86.2 | 52.1 | 57.4 | 63.1 | 59.3 | 55.6 | 56.5 | 45.3 | 52.0 | 65.2 | 56.2 | 48.4 |
| | Wanda | AR | 21.6 | 19.8 | 27.4 | 17.3 | 19.8 | 18.9 | 22.9 | 27.3 | 27.2 | 27.6 | 26.9 | 23.9 | 26.2 | 27.4 | 25.5 | 25.6 | 25.5 | 25.6 | 27.1 | 24.9 | 26.0 | 26.0 | 26.4 | 26.8 | 25.7 |
| | | DE | 21.6 | 19.9 | 27.6 | 17.2 | 20.1 | 19.0 | 22.6 | 25.7 | 27.3 | 27.3 | 27.3 | 23.6 | 25.8 | 25.7 | 25.7 | 25.5 | 25.5 | 25.8 | 26.7 | 24.8 | 25.2 | 26.2 | 26.4 | 27.0 | 25.6 |
| | | EN | 20.4 | 19.4 | 28.3 | 18.0 | 19.8 | 19.3 | 22.3 | 26.7 | 27.1 | 27.3 | 27.1 | 23.1 | 27.9 | 25.4 | 25.1 | 25.6 | 25.5 | 25.9 | 27.0 | 25.0 | 24.9 | 26.2 | 26.7 | 27.2 | 25.6 |
| | | ES | 21.4 | 20.4 | 27.4 | 17.2 | 20.9 | 19.2 | 22.5 | 27.4 | 27.3 | 27.3 | 27.3 | 22.1 | 27.4 | 22.3 | 25.1 | 25.7 | 25.5 | 25.5 | 27.0 | 25.3 | 25.0 | 26.1 | 26.5 | 27.0 | 25.6 |
| | | RU | 21.0 | 20.4 | 27.8 | 16.9 | 19.7 | 18.3 | 22.6 | 26.3 | 27.3 | 27.2 | 27.4 | 22.4 | 28.9 | 27.9 | 25.9 | 25.6 | 25.5 | 25.8 | 25.5 | 24.9 | 25.6 | 26.2 | 26.4 | 26.9 | 26.0 |
| | | SW | 21.4 | 19.4 | 28.3 | 17.3 | 20.8 | 18.9 | 22.1 | 26.0 | 27.3 | 27.4 | 27.7 | 22.2 | 27.1 | 24.4 | 25.6 | 25.7 | 25.5 | 25.7 | 26.9 | 26.0 | 25.3 | 26.1 | 26.6 | 27.0 | 25.7 |
| | | ZH | 21.2 | 20.1 | 28.2 | 16.5 | 21.1 | 19.2 | 22.4 | 27.1 | 27.2 | 26.4 | 28.3 | 23.0 | 26.6 | 27.0 | 25.5 | 25.5 | 25.5 | 25.6 | 25.1 | 24.6 | 25.5 | 26.1 | 26.6 | 26.9 | 25.5 |
| | SparseGPT | AR | 21.2 | 19.2 | 32.3 | 19.4 | 19.7 | 19.7 | 21.5 | 27.0 | 26.7 | 27.4 | 27.7 | 28.4 | 23.0 | 27.1 | 24.9 | 24.8 | 23.7 | 24.8 | 24.8 | 24.8 | 28.5 | 26.2 | 28.0 | 26.9 | 26.1 |
| | | DE | 21.6 | 21.3 | 33.7 | 18.3 | 19.7 | 20.2 | 22.0 | 24.7 | 24.3 | 24.3 | 24.0 | 28.2 | 24.3 | 25.0 | 24.7 | 25.1 | 24.5 | 25.0 | 25.1 | 25.0 | 25.5 | 28.5 | 28.8 | 27.6 | 26.3 |
| | | EN | 21.5 | 19.1 | 33.6 | 19.2 | 19.6 | 20.2 | 20.9 | 28.9 | 27.7 | 29.4 | 29.1 | 28.2 | 27.7 | 26.7 | 25.0 | 25.6 | 25.3 | 25.6 | 25.8 | 24.8 | 25.4 | 26.4 | 28.8 | 27.0 | 26.4 |
| | | ES | 22.5 | 19.5 | 32.6 | 18.3 | 19.7 | 20.4 | 20.3 | 24.3 | 26.6 | 26.1 | 26.0 | 28.8 | 27.9 | 27.1 | 24.9 | 25.2 | 24.5 | 25.1 | 25.4 | 25.3 | 25.3 | 26.7 | 28.5 | 28.8 | 26.2 |
| | | RU | 22.2 | 20.4 | 36.4 | 19.1 | 21.3 | 21.6 | 21.3 | 23.2 | 23.0 | 25.8 | 22.6 | 22.8 | 24.4 | 22.7 | 22.9 | 23.2 | 24.2 | 23.3 | 23.6 | 25.2 | 26.1 | 28.2 | 27.2 | 29.3 | |
| | | SW | 21.6 | 18.0 | 30.7 | 17.0 | 18.9 | 20.4 | 20.3 | 26.0 | 22.6 | 25.1 | 23.1 | 26.6 | 23.4 | 22.8 | 25.1 | 24.6 | 25.1 | 24.7 | 24.9 | 24.9 | 25.6 | 26.6 | 27.5 | 27.2 | 26.2 |
| | | ZH | 21.2 | 19.2 | 31.0 | 18.2 | 20.3 | 20.5 | 21.2 | 22.9 | 22.9 | 22.9 | 22.9 | 22.9 | 22.9 | 22.9 | 22.7 | 22.7 | 22.9 | 22.7 | 22.7 | 22.8 | 24.9 | 26.0 | 26.9 | 26.3 | 26.0 |

Table 11: Aya-23 models of size 8B and 35B pruned towards unstructured sparsity of 20% and 80% respectively.

| | | | | | ARC[acc] | | | |
|---|---|---|---|---|---|---|---|---|
| | | | AR | DE | EN-E | EN-C | ES | RU | ZH |
| **Aya-23-8B** | | - | - | 37.0 | 42.4 | 75.1 | 45.1 | 43.9 | 38.6 | 35.9 |
| | SparseGPT | AR | 31.4 | 34.5 | 69.5 | 42.0 | 36.7 | 33.4 | 32.0 |
| | | DE | 29.8 | 35.8 | 69.9 | 41.0 | 35.4 | 33.5 | 32.0 |
| | | EN | 29.8 | 34.5 | 71.4 | 42.0 | 36.2 | 34.1 | 31.7 |
| | | ES | 30.3 | 34.8 | 70.3 | 41.9 | 36.8 | 34.2 | 32.0 |
| | | RU | 30.5 | 35.8 | 70.3 | 42.2 | 36.6 | 33.7 | 31.5 |
| | | SW | 29.3 | 34.2 | 68.9 | 40.6 | 34.0 | 31.9 | 31.8 |
| | | ZH | 30.0 | 32.6 | 67.8 | 40.4 | 35.0 | 32.8 | 31.7 |

Table 12: First and last layer of SparseGPT-pruned Aya-23 8B model replaced with full-sized weights (46.875% sparsity).

| | | | | | ARC[acc] | | | |
|---|---|---|---|---|---|---|---|---|
| | | | AR | DE | EN-E | EN-C | ES | RU | ZH |
| **Aya-23-8B** | | - | - | 37.0 | 42.4 | 75.1 | 45.1 | 43.9 | 38.6 | 35.9 |
| | SparseGPT | AR | 31.5 | 33.6 | 68.6 | 40.4 | 35.7 | 32.2 | 31.2 |
| | | DE | 28.7 | 35.6 | 69.5 | 41.1 | 34.4 | 32.5 | 30.9 |
| | | EN | 29.6 | 33.7 | 70.8 | 41.8 | 35.6 | 32.0 | 31.0 |
| | | ES | 28.7 | 33.9 | 69.6 | 40.4 | 36.3 | 33.5 | 30.9 |
| | | RU | 29.7 | 34.2 | 69.6 | 41.3 | 35.4 | 33.7 | 31.0 |
| | | SW | 29.4 | 32.8 | 68.2 | 39.8 | 33.0 | 31.0 | 30.6 |
| | | ZH | 29.3 | 32.2 | 67.5 | 39.2 | 34.4 | 30.5 | 31.7 |

Table 13: Aya-23 8B model pruned with SparseGPT for 50% unstructured sparsity as reference for the previous table.

# F QUANTIZATION

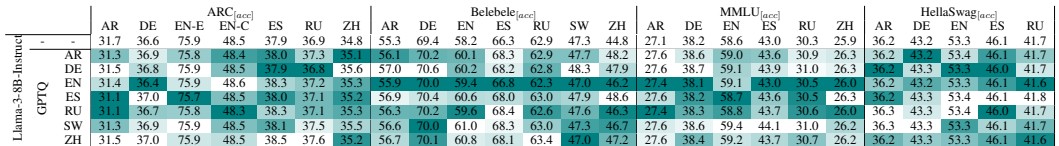

| | | | ARC[acc] | | | | | | Belebele[acc] | | | | | | | MMLU[acc] | | | | | | HellaSwag[acc] | | | | |
|---|---|---|---|---|---|---|---|---|---|---|---|---|---|---|---|---|---|---|---|---|---|---|---|---|---|---|
| | | AR | DE | EN-E | EN-C | ES | RU | ZH | AR | DE | EN | ES | RU | SW | ZH | AR | DE | EN | ES | RU | ZH | AR | DE | EN | ES | RU |
| | - | 31.7 | 36.6 | 75.9 | 48.5 | 37.9 | 36.9 | 34.8 | 55.3 | 69.4 | 58.2 | 66.3 | 62.9 | 47.3 | 44.8 | 27.1 | 38.2 | 58.6 | 43.0 | 30.3 | 25.9 | 36.2 | 43.2 | 53.3 | 46.1 | 41.7 |
| AR | | 28.1 | 32.4 | 72.2 | 44.9 | 33.5 | 34.0 | 30.4 | 47.6 | 58.4 | 34.3 | 65.3 | 46.7 | 29.8 | 28.8 | 27.2 | 35.8 | 55.0 | 41.2 | 26.6 | 23.9 | 34.6 | 41.1 | 52.4 | 44.5 | 40.5 |
| DE | | 29.2 | 34.8 | 72.9 | 45.6 | 36.0 | 34.2 | 32.6 | 33.6 | 58.4 | 34.9 | 59.2 | 42.3 | 24.6 | 26.9 | 24.3 | 32.6 | 52.8 | 37.8 | 25.2 | 23.7 | 34.0 | 41.6 | 52.3 | 44.5 | 40.2 |
| EN | | 26.9 | 31.0 | 72.9 | 45.9 | 34.0 | 33.0 | 32.3 | 42.6 | 52.0 | 45.7 | 55.7 | 46.4 | 24.9 | 34.4 | 25.7 | 32.2 | 54.4 | 40.8 | 26.4 | 24.9 | 33.2 | 40.9 | 52.2 | 44.0 | 40.1 |
| ES | | 27.3 | 33.7 | 72.3 | 47.0 | 36.2 | 35.1 | 30.6 | 24.8 | 31.3 | 26.9 | 45.0 | 33.0 | 23.3 | 23.8 | 22.8 | 26.0 | 56.4 | 34.3 | 23.6 | 22.9 | 34.0 | 41.7 | 51.6 | 44.2 | 40.2 |
| RU | | 27.6 | 31.9 | 71.6 | 45.1 | 34.4 | 32.2 | 32.8 | 30.7 | 54.3 | 42.6 | 54.8 | 46.3 | 25.2 | 30.3 | 22.9 | 27.9 | 52.4 | 29.8 | 23.3 | 22.9 | 33.8 | 41.3 | 52.3 | 44.4 | 40.4 |
| SW | | 26.3 | 30.9 | 71.8 | 44.4 | 34.2 | 33.0 | 32.5 | 42.8 | 55.7 | 50.3 | 56.9 | 47.4 | 33.1 | 35.7 | 27.5 | 36.0 | 56.8 | 40.6 | 27.1 | 24.2 | 33.8 | 41.2 | 52.0 | 44.2 | 39.9 |
| ZH | | 26.7 | 34.2 | 71.9 | 44.9 | 35.9 | 32.9 | 32.1 | 56.3 | 61.2 | 59.1 | 56.4 | 60.6 | 27.7 | 38.2 | 27.9 | 34.8 | 58.8 | 39.8 | 27.8 | 24.9 | 34.4 | 41.2 | 52.5 | 44.2 | 40.3 |

Table 14: Downstream performance of Llama-3 8B after GPTQ Quantization to 4-bit weights and with a group-size of 128.

| | | | ARC[acc] | | | | | | Belebele[acc] | | | | | | | MMLU[acc] | | | | | | HellaSwag[acc] | | | | |
|---|---|---|---|---|---|---|---|---|---|---|---|---|---|---|---|---|---|---|---|---|---|---|---|---|---|---|
| | | AR | DE | EN-E | EN-C | ES | RU | ZH | AR | DE | EN | ES | RU | SW | ZH | AR | DE | EN | ES | RU | ZH | AR | DE | EN | ES | RU |
| | - | 31.7 | 36.6 | 75.9 | 48.5 | 37.9 | 36.9 | 34.8 | 55.3 | 69.4 | 58.2 | 66.3 | 62.9 | 47.3 | 44.8 | 27.1 | 38.2 | 58.6 | 43.0 | 30.3 | 25.9 | 36.2 | 43.2 | 53.3 | 46.1 | 41.7 |
| AR | | 31.3 | 36.9 | 75.8 | 48.4 | 38.0 | 37.3 | 35.1 | 56.1 | 70.2 | 60.1 | 68.3 | 62.9 | 47.7 | 48.2 | 27.6 | 38.6 | 59.0 | 43.6 | 30.9 | 26.3 | 36.2 | 43.2 | 53.4 | 46.1 | 41.7 |
| DE | | 31.5 | 36.8 | 75.9 | 48.5 | 37.9 | 36.8 | 35.6 | 57.0 | 70.6 | 60.2 | 68.2 | 62.8 | 48.3 | 47.9 | 27.6 | 38.7 | 59.1 | 43.9 | 31.0 | 26.3 | 36.2 | 43.3 | 53.3 | 46.0 | 41.7 |
| EN | | 31.4 | 36.4 | 75.9 | 48.6 | 38.3 | 37.2 | 35.3 | 55.9 | 70.0 | 59.4 | 66.8 | 62.3 | 47.0 | 46.2 | 27.4 | 38.1 | 59.1 | 43.0 | 30.5 | 26.0 | 36.2 | 43.2 | 53.3 | 46.1 | 41.6 |
| ES | | 31.1 | 37.0 | 75.7 | 48.5 | 38.0 | 37.1 | 35.2 | 56.9 | 70.4 | 60.6 | 68.0 | 63.0 | 47.9 | 48.6 | 27.6 | 38.2 | 58.7 | 43.6 | 30.5 | 26.3 | 36.2 | 43.3 | 53.4 | 46.1 | 41.8 |
| RU | | 31.1 | 36.7 | 75.8 | 48.3 | 38.3 | 37.1 | 35.3 | 56.3 | 70.2 | 59.6 | 68.4 | 62.6 | 47.6 | 46.3 | 27.4 | 38.3 | 58.8 | 43.7 | 30.6 | 26.0 | 36.3 | 43.3 | 53.4 | 46.0 | 41.7 |
| SW | | 31.3 | 36.9 | 75.9 | 48.5 | 38.1 | 37.5 | 35.5 | 56.6 | 70.0 | 61.0 | 68.3 | 63.0 | 47.3 | 46.7 | 27.6 | 38.6 | 59.4 | 44.1 | 31.0 | 26.2 | 36.3 | 43.3 | 53.3 | 46.1 | 41.7 |
| ZH | | 31.5 | 37.0 | 75.9 | 48.5 | 38.5 | 37.6 | 35.2 | 56.7 | 70.1 | 60.8 | 68.1 | 63.4 | 47.0 | 47.2 | 27.6 | 38.4 | 59.2 | 43.7 | 30.7 | 26.2 | 36.2 | 43.3 | 53.3 | 46.1 | 41.6 |

Table 15: Downstream performance of Llama-3 8B after GPTQ Quantization to 8-bit weights and with a group-size of 128.

# G    SUPPLEMENTARY ANALYSIS

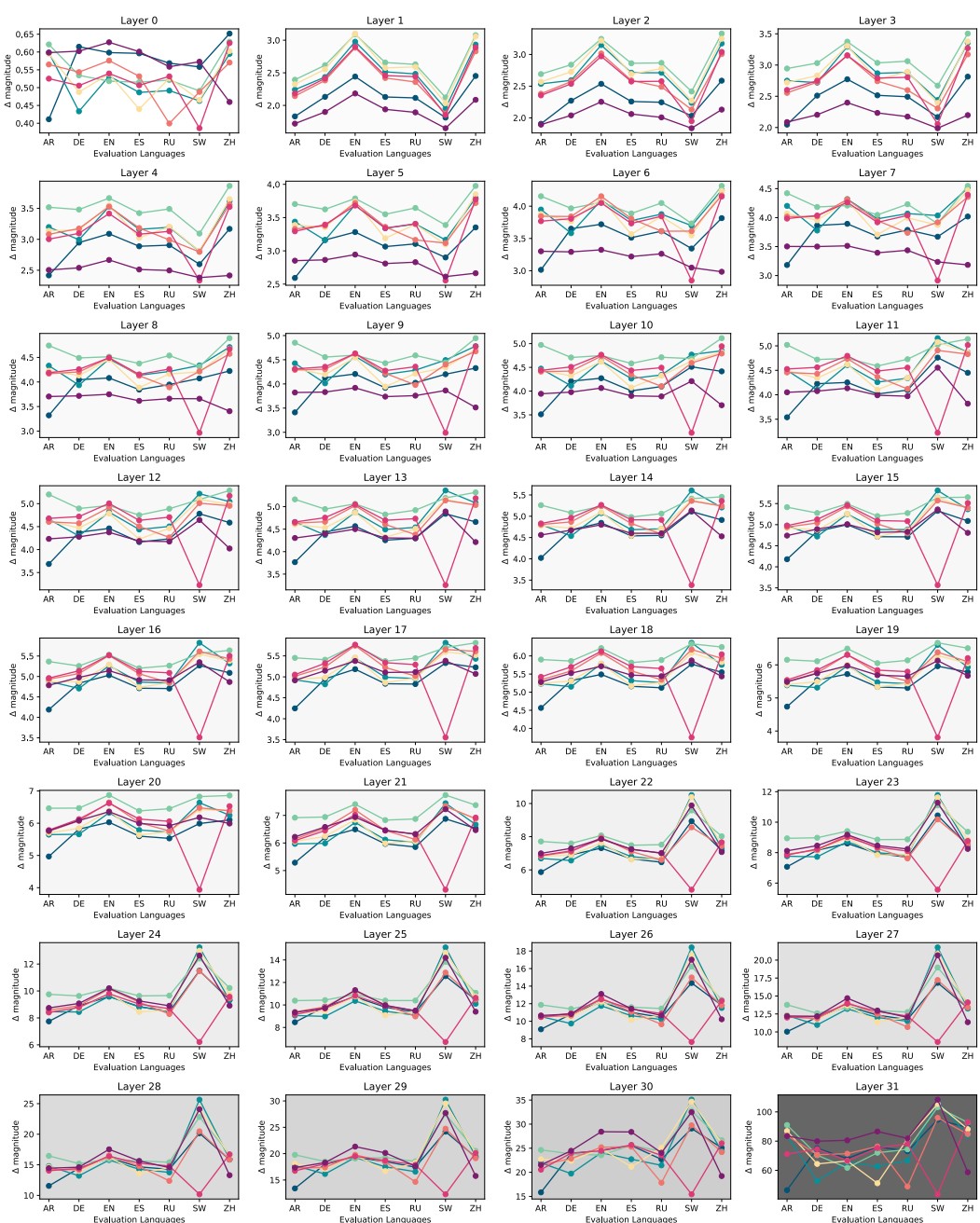

Figure 4: Language-wise mean magnitude of difference between the hidden states of a full-sized Llama3 8B base model and its 50% unstructured sparsity SparseGPT-pruned model for all calibration-evaluation language pairs. The test languages are shown on the x-axis, the calibration languages are color-coded (AR, DE, ES, EN, RU, SW, ZH). The background color indicates the magnitude of the maximum deviation. We use all the 900 samples from the fully parallel Belebele dataset (Bandarkar et al., 2023) as input, concatenating the context passage and question for each sample.

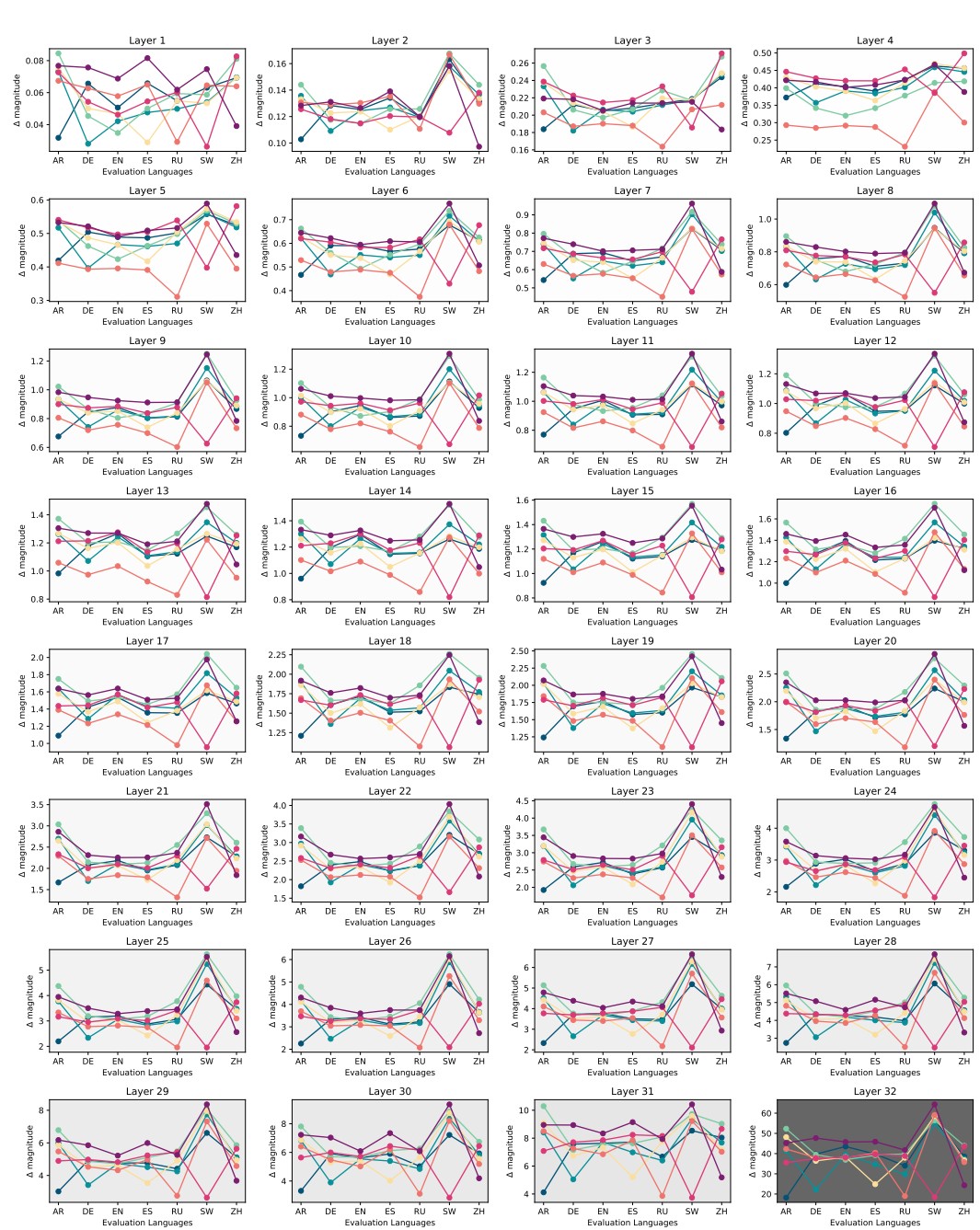

Figure 5: Language-wise mean magnitude of difference between the prompt-wise and layer-wise averaged hidden states of a full-sized Llama3 8B reference model and its 50% unstructured sparsity SparseGPT-pruned model for all calibration-evaluation language pairs. The test languages are shown on the x-axis, the calibration languages are color-coded (AR, DE, ES, EN, RU, SW, ZH). The background color indicates the magnitude of the maximum deviation. We use all the 900 samples from the fully parallel Belebele dataset (Bandarkar et al., 2023) as input, concatenating the context passage and question for each sample.

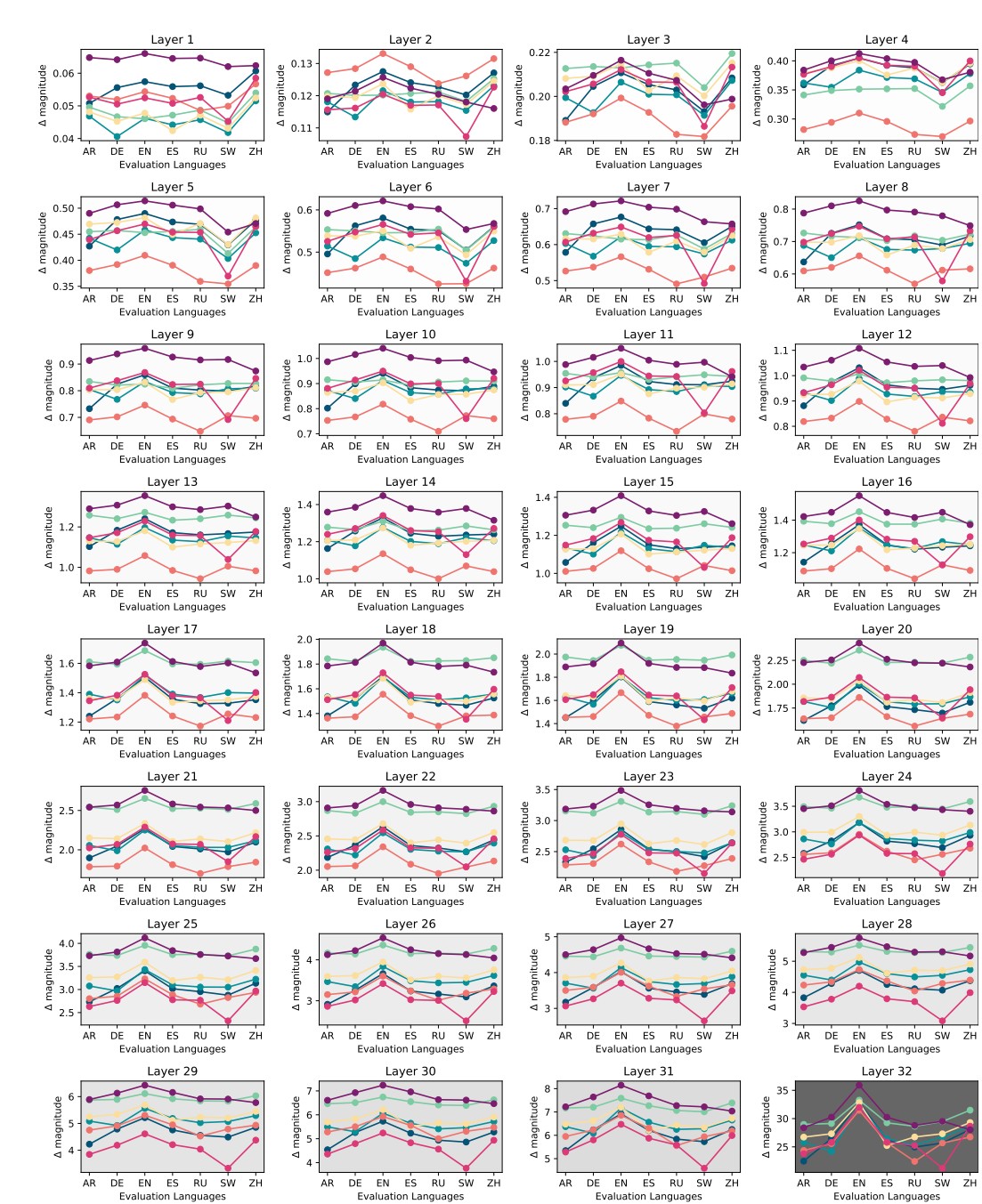

Figure 6: Language-wise mean magnitude of difference between the prompt-wise and layer-wise averaged *language-agnostic* features extracted with LSAR for an full-sized and pruned (50% unstructured sparsity, SparseGPT) Llama 3 8b model. Both, the LSAR projection matrix and the feature differences, were computed per model over all samples of the Belebele dataset for the seven calibration/test languages. The gray-scaled background represents the y-axis magnitude. The test languages are shown on the x-axis, the calibration languages are color-coded (AR, DE, ES, EN, RU, SW, ZH). The background color indicates the magnitude of the maximum deviation.

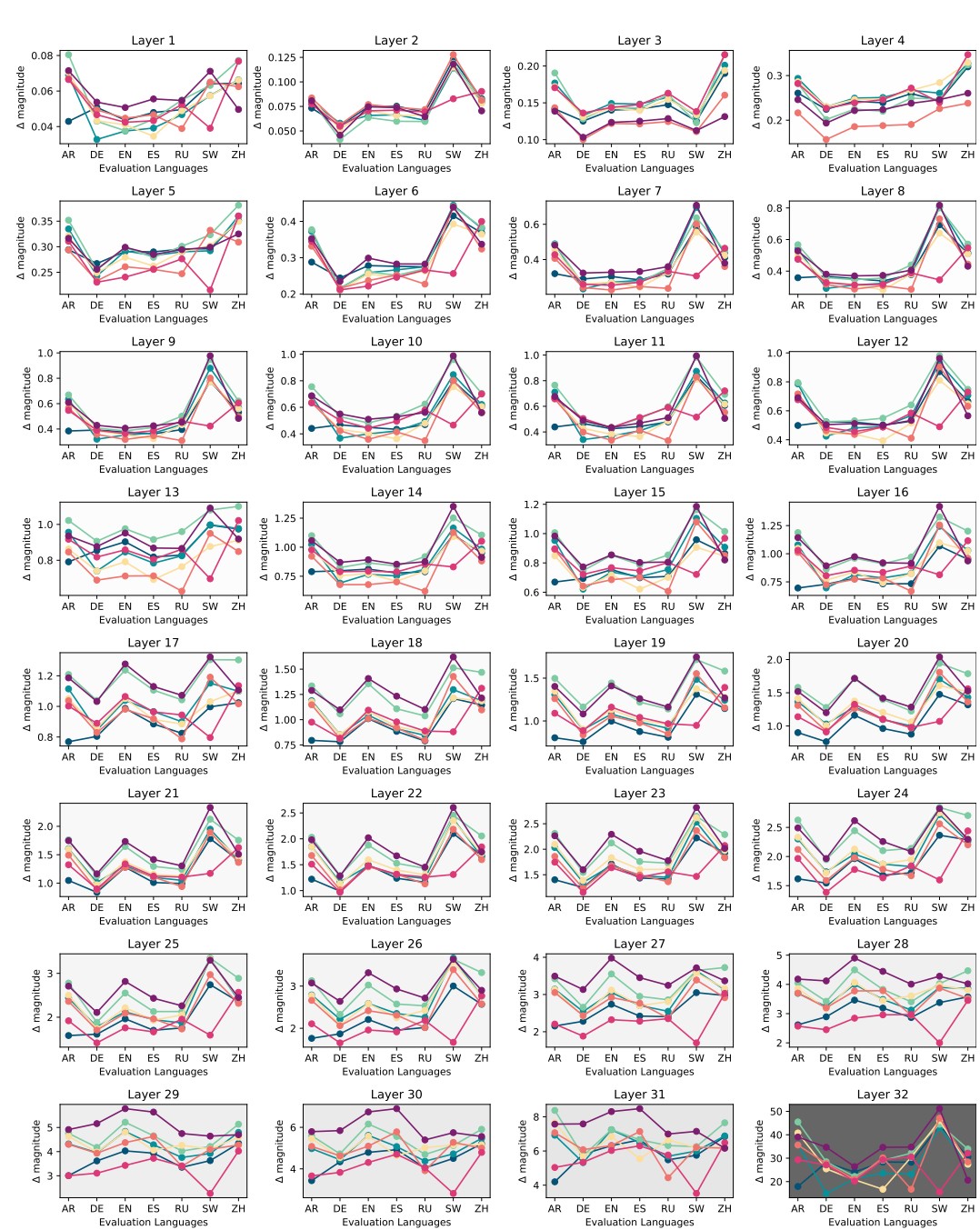

Figure 7: Language-wise mean magnitude of difference between the prompt-wise and layer-wise averaged *language-specific* features extracted with LSAR for an full-sized and pruned (50% unstructured sparsity, SparseGPT) Llama 3 8b model. Both, the LSAR projection matrix and the feature differences, were computed per model over all samples of the Belebele dataset for the seven calibration/test languages, concatenating the context passage and question for each sample. The test languages are shown on the x-axis, the calibration languages are color-coded (AR, DE, ES, EN, RU, SW, ZH). The background color indicates the magnitude of the maximum deviation.

| Subcomponent | DE | EN | ZH |
|---|---|---|---|
| **Q** | 0.890 | 0.880 | 0.874 |
| **K** | 0.891 | 0.880 | 0.875 |
| **V** | 0.882 | 0.871 | 0.865 |
| **attn.out** | 0.845 | 0.832 | 0.825 |
| **ffn.up** | 0.875 | 0.863 | 0.849 |
| **ffn.gate** | 0.873 | 0.862 | 0.847 |
| **ffn.down** | 0.870 | 0.852 | 0.843 |

Table 16: Average IoU of all layers for a specific subcomponent. IoUs are computed for the pruning masks of three pruning runs on the same calibration language (DE, EN or ZH).

| Subcomponent | EN-DE | ZH-DE | ZH-EN |
|---|---|---|---|
| **Q** | 0.886 | 0.855 | 0.857 |
| **K** | 0.886 | 0.855 | 0.857 |
| **V** | 0.879 | 0.846 | 0.848 |
| **attn.out** | 0.867 | 0.826 | 0.824 |
| **ffn.up** | 0.871 | 0.833 | 0.835 |
| **ffn.gate** | 0.870 | 0.832 | 0.833 |
| **ffn.down** | 0.846 | 0.808 | 0.811 |

Table 17: Average IoU of all layers for a specific subcomponent. IoUs are computed between the intersected pruning masks of three pruning runs for the same calibration language and the intersected pruning mask of another calibration language (EN-DE, ZH-DE and ZH-EN).

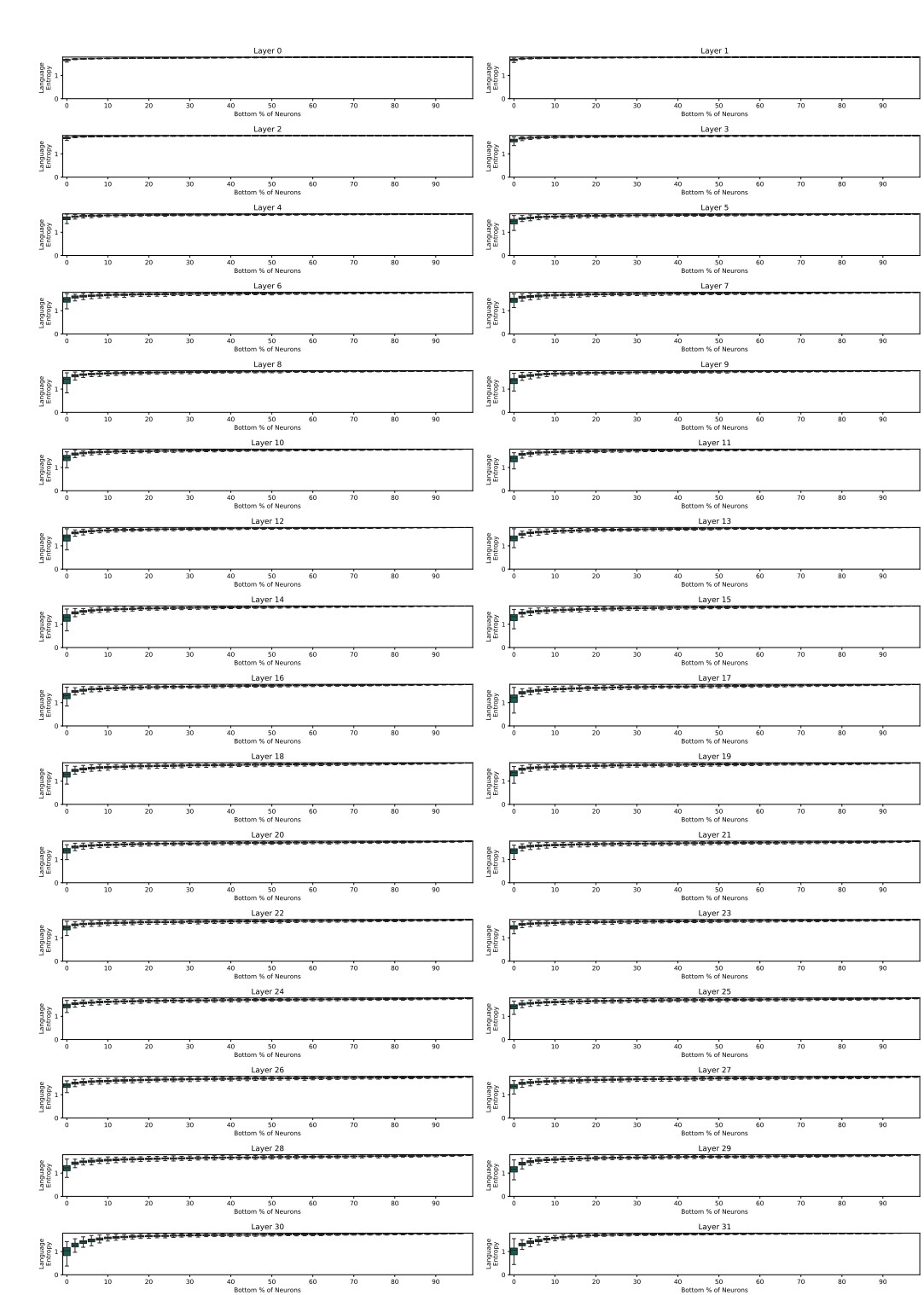

Figure 8: Layerwise LAPE score for (50% unstructured sparsity, SparseGPT) Llama 3 8b model.

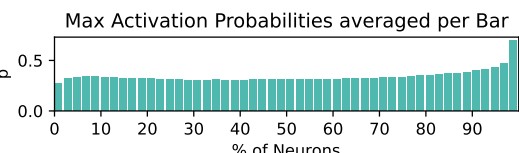

Figure 9: Maximum activation likelihood (i.e. for the language a neuron activates most often in) of FFN up projection neurons in the full-sized reference model, sorted in ascending order with respect to their LAPE score. The languages used for LAPE score computation are AR, DE, EN, ES, RU, and ZH, each represented by 1000000 tokens from samples in the mC4 validation set.

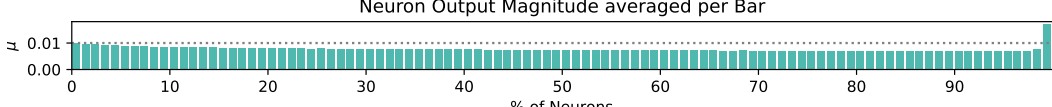

Figure 10: Mean absolute neuron outputs for FFN up projection neurons, sorted in ascending order with respect to their LAPE score. The languages used for LAPE score and neuron output computation are AR, DE, EN, ES, RU, and ZH, each represented by 1000000 tokens from samples in the mC4 validation set.

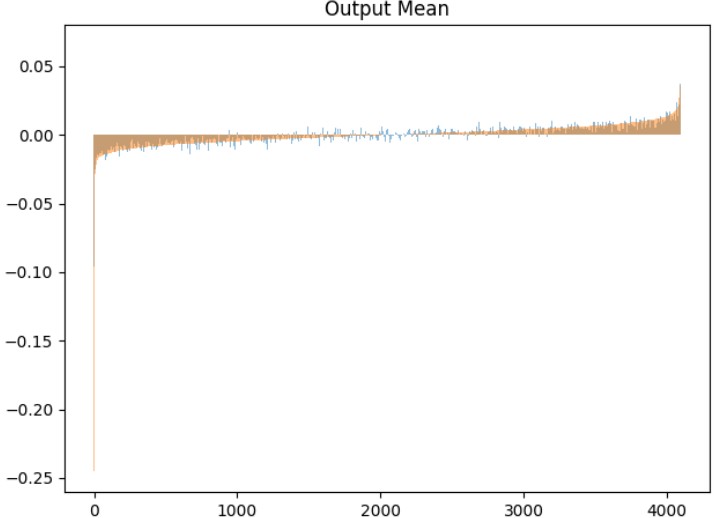

Figure 11: Mean activations of MLP down projection in the 5th layer for full-sized (orange) and pruned (blue) Llama 3 8b model (50% unstructured sparsity, Wanda)

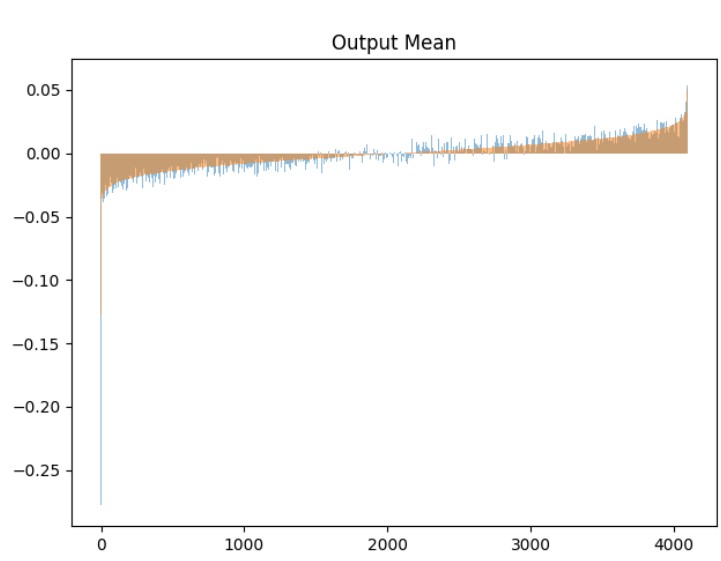

Figure 12: Mean activations of MLP down projection in the 15th layer for full-sized (orange) and pruned (blue) Llama 3 8b model (50% unstructured sparsity, Wanda)

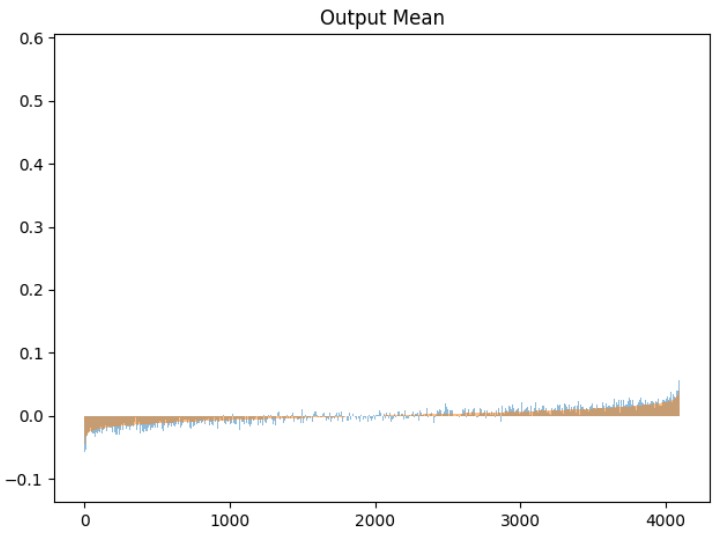

Figure 13: Mean activations of attention output projection in the 15th layer for full-sized (orange) and pruned (blue) Llama 3 8b model (50% unstructured sparsity, Wanda)

