# OpenReview forum: "Investigating Language-Specific Calibration For Pruning Multilingual Large Language Models"
_ICLR.cc/2025/Conference — ICLR 2025 Conference Withdrawn Submission_

### Official Review · Reviewer_FuS7 · 2024-10-27

**Soundness:** 4
**Presentation:** 4
**Contribution:** 4
**Rating:** 8
**Confidence:** 3

**Summary:**

This paper presents an empirical study on calibrating the pruning of multi-lingual language models. Different settings are investigated, which includes a comparison of languages, tasks, pruning techniques and models. The findings show that calibration in the target language for post-training pruning is beneficial for retaining the language modelling capability. However, it does not necessarily benefit downstream tasks. An analysis of the preservation of language-specific and language-agnostic features is conducted through latent subspaces and pruning masks.

**Strengths:**

- This paper addresses an important problem of language-specific calibration for pruning multilingual LLMs. To my knowledge, this hasn't been investigated in sufficient detail before.
- The experiments are thorough and cover different dimensions of the calibration problem on standard multilingual tasks (MKQA, MMLU).
- The paper is well-written and the results are presented clearly.
- The findings are significant, as they offer useful guidelines to practitioners for the selection of calibration data for post-training pruning to achieve better performance.

**Weaknesses:**

- The observation that performance patterns in smaller models do not consistently translate to larger models (e.g., Llama-3 70B and Aya-23 35B) suggests the presence of additional unexplored factors. Since pruning is potentially more useful in reducing the size of larger models and allows them to be used in resource-constrained environments, further analysis regarding this point would be important for the overall completeness of this study.

- Minor issues:
    - The question on 275-277 can be restructured for clarity.
    - Minor typo (line 346): poist -> posit

**Questions:**

- How does changing the number of inputs in the calibration set (which is 128 currently) affect the values in Table 1? Can we expect better performance as the calibration examples increase?
- Presenting a few qualitative examples (in the appendix) for the finding that 'pruning impairs the reasoning capability' would help understand the problem more clearly.
- Section 4.5 states that the performance patterns from the smaller models do not consistently hold true on their bigger counterparts. Can the authors hypothesize what additional factors might be contributing to this difference in performance on the bigger models?

---

> ### Author Response · Authors · 2024-11-19
> **Response to Reviewer FuS7 (Part 1)**
>
> We thank the reviewer for the assessment of our paper and valuable feedback. We will correct the typos and revise the suggested phrases for better clarity in the final version.
>
> **Comment on Weakness:**
> For the final paper version, we will add pruning mask similarity and LAPE score results for Llama 3 70B to the appendix. We hope this gives more insights into performance pattern differences between model sizes and addresses the reviewer's concern.
>
> **Q1:** _Impact of different amounts of calibration set samples_
>
> When increasing the amount of calibration samples we expect two outcomes.
>
> 1. Previous research suggests that adding more samples provides diminishing returns. We use a default number of calibration samples (128), yet we employ a longer sequence length (8192 * 128 tokens) for calibration. This is because prior studies relied on older models with shorter maximum context lengths (e.g., 2048 × 128 tokens for LLaMA 1) [1], [2].
>
> 2. Previous research on the impact of calibration data on the downstream task performance has observed fluctuating performance between different calibration sets [3]. As the amount of calibration data increases, this variance is reported to decreases slightly [4].
>
> The costs of increasing the amount of calibration samples would be disproportionate to the benefit, therefore, how fewer calibration samples impact the downstream-task performance, except from less performance with higher variance, remains an open research question.
>
> **Q2:** _Qualitative examples_
>
> Thank you so much for pointing it out and we fully agree that qualitative examples will be helpful (and interesting to examine). We will provide qualitative results in the appendix for the final version and give a first outlook below.
>
> Models pruned with different calibration languages retrieve different facts (correctness set aside) for the same prompt:
>
> __Prompt:__ “Who is the greatest ballerina of all time? Explain why.”
>
> >__Answer (EN-calibrated):__ “According to many ballet enthusiasts, the greatest ballerina of all time is considered to be Natalia Makarova.”
>
> >__Answer (RU-calibrated):__ “According to many experts and dance historians, the greatest ballerina of all time is Maria Callas (1923-1969). She was a Russian-born American ballet dancer and singer, known for her unique and expressive technique, and her iconic performances in roles like La Scala\'s 'La Bohème' and 'Eugene Onegin'.”
>
> __Prompt:__ “Bernice studied some organisms that live together in a field. In which way is a rabbit similar to the plant it eats?”
>
> >__Answer (EN-calibrated):__ “A clever question!\n\nBernice studied organisms that live together in a field, and she found that a rabbit is similar to the plant it eats in the way that both are **herbivores**. Just like the plant, a rabbit is a herbivore, meaning it eats plants and vegetation.”
>
> >__Answer (DE-calibrated):__ “Bernice is studying the organisms that live together in a field!\n\nIn this case, the rabbit is similar to the plant it eats in the following way:\n\nBoth the rabbit and the plant are heterotrophs, meaning they are organisms that obtain their nutrients from other organisms. Rabbits are heterotrophic because they eat plants and oth er animals, whereas plants are autotrophic, meaning they produce their own food from the energy they absorb from the sun.\n\nIn other words, both the rabbit and the plant are consumers that obtain their energy from sources outside themselves, rather than producing their own food.”
>
> Or skip the answer:
>
> __Prompt:__ “During which stage of its life cycle does a butterfly spend the most time eating plant leaves?”
>
> >__Answer (EN-calibrated):__ -
>
> >__Answer (DE-calibrated):__ “Butterflies, also known as Lepidoptera, typically spend the most time eating plant leaves during the larval stage, also known as the caterpillar stage.\n\nAs a caterpillar, the butterfly spends most of its time feeding on plant leaves, leaves, and other plant material to accumulate energy and nutrients. This feeding period is cruc ial for the caterpillar's growth and development.\n\nThe larval stage is typically the longest stage of the butterfly's life cycle, and during this stage, the caterpillar feeds on various plant parts, such as leaves, stems, and flowers, to grow and develop.\n\nAfter the caterpillar has reached its full size, it forms a pupa or chrysalis, and then emer ges as an adult butterfly, ready to start the next generation of its life cycle.\n\nSo, to answer your question, the stage where a butterfly spends the most time eating plant leaves is the larval stage, which is the caterpillar stage.”

---

> > ### Author Response · Authors · 2024-11-19
> > **Response to Reviewer FuS7 (Part 2)**
> >
> > Also, the amount of detail and length of the answer can vary significantly:
> >
> > __Prompt:__ “When an electric current is passed through water (H_{2}O) using a specific apparatus, hydrogen (H_{2}) and oxygen (O_{2}) gases form. How can this reaction be shown in a chemical equation?”
> >
> > >__Answer (EN-calibrated):__ “The reaction you are referring to is the electrolysis of water, which is a process where an electric current is passed through water to produce hydrogen and oxygen gases.\n\nThe chemical equation for this reaction is:\n\n2H2O → 2H2 + O2\n\nThis equation shows that the water (H2O) is decomposed into hydrogen (H2) and oxygen (O2) gases when an electric current is passed through it.\n\nNote that this reaction is often referred to as the electrolysis of water, and it is commonly used in applications such as fuel cells, hydrogen production, and water purification.”
> >
> > >__Answer (RU-calibrated):__ “The reaction can be represented as:\n\n2H2O → H2 + O2\n\nThis reaction is known as electrolysis of water, where an electric current is used to split the water molecules into hydrogen and oxygen gases.”
> >
> > **Q3 and regarding weakness**: _Reason for performance differences between models of different size_
> >
> > We agree with the reviewer, that finding the cause for performance differences between models of different size, such as Wanda becoming more efficient on larger LLMs, would be valuable.
> > A possible yet insufficiently explored explanation for this could be patterns that only emerge with growing LLM size. For example [5] have identified the emergence of increasingly structured outlier features for LLMs past a certain amount of parameters. More recently, [6] have identified a new scaling law for deeply quantized LLMs, indicating pretraining Llama models with more than 3 billion parameters with 8-bit activations and ternary weights could eventually outperform full-precision counterparts. Other research such as [7] demonstrates neuron activations becoming more sparse at increasing LLM scale.
> > While research towards properly explaining LLM behavior at different model sizes is ongoing, we believe that combining such insights with new pruning techniques could be promising to better adapt pruning to the LLM size.
> >
> >
> > We hope that these clarifications, together with the planned revisions, sufficiently address the reviewer’s concerns.
> >
> > [1] Wanda: A Simple and Effective Pruning Approach for Large Language Models, Sun et al, ICLR 2024
> >
> > [2] SparseGPT: SparseGPT: Massive Language Models Can be Accurately Pruned in One-Shot, Frantar et al, ICML 2023
> >
> > [3] On the Impact of Calibration Data in Post-training Quantization and Pruning, Williams et al, ACL 2024
> >
> > [4] Beware of Calibration Data for Pruning Large Language Models. Ji et al, https://arxiv.org/pdf/2410.17711
> >
> > [5] LLM.int8(): 8-bit Matrix Multiplication for Transformers at Scale, Dettmers et al, NeurIPS 2022
> >
> > [6] The Era of 1-bit LLMs: All Large Language Models are in 1.58 Bits, Ma et al, [https://arxiv.org/abs/2402.17764](https://arxiv.org/abs/2402.17764), 2024
> >
> > [7] Neurons in Large Language Models: Dead, N-gram, Positional, Voita et al, ACL RR 2024

---

> > > ### Comment · Reviewer_FuS7 · 2024-11-24
> > >
> > > Thank you to the authors for their detailed response. I'll keep my score at 8.

---

### Official Review · Reviewer_vgGp · 2024-10-31

**Soundness:** 3
**Presentation:** 2
**Contribution:** 3
**Rating:** 5
**Confidence:** 2

**Summary:**

This paper presents a comprehensive empirical study on language-specific calibration for post-training pruning of multilingual Large Language Models (LLMs), using LLaMA-3 and Aya-23 as the base models, and applying Wanda and SparseGPT as the pruning techniques. The authors found that calibrating in the target language effectively retains language modeling capabilities, though it does not necessarily improve performance on downstream tasks. Additionally, the paper provides a detailed analysis of latent subspaces, pruning masks, and individual neurons within the pruned models to examine the effects of pruning on both language-agnostic and language-specific neurons, where language-agnostic features failed to be retained.

**Strengths:**

S1. The paper presents comprehensive experiments to investigate both language-specific and language-agnostic features, covering multiple calibration languages, models (LLaMA-3 and Aya-23), and pruning techniques (Wanda and SparseGPT). This study evaluates a wide range of metrics—including perplexity, downstream task performance, signal-to-noise ratio, and pruning error—extending the analysis to neuron-level effects and latent subspaces.

S2. The paper provides insights not previously found in English-focused pruning studies, revealing the distinct impacts of pruning on multilingual LLMs.

**Weaknesses:**

W1. While the paper includes extensive experimentation, the analysis of results feels somewhat limited, as many findings are inconclusive, especially in Section 4. Specific examples include:
- Calibrating on the target language itself **generally** yields the best pruning performance and the lowest perplexity.
- Calibrating using the target language **typically** results in acceptable performance on downstream tasks, though not consistently the best.
- Pruning **can** shift which languages the model performs best or worst on
- Employing bilingual and multilingual calibration sets **occasionally** improves performance on downstream tasks, compared to monolingual calibration.
- The performance patterns and findings from the smaller models **do not consistently hold** true on their bigger counterparts.
- **No pruning technique consistently** performs best in all tasks.

The insights presented in the paper are informative but lack conclusive takeaways, which limits actionable takeaways for the readers. Going back to the introduction introduced in the paper, to my understanding, it still remains unclear how to calibrate pruning to optimize the post-pruning performance of multilingual LLMs on tasks in non-English languages. I think providing a more in-depth analysis by either explaining why certain findings remain inconclusive or making stronger, more definitive claims would significantly enhance the paper. For instance, identifying specific linguistic factors that influence when language-specific calibration is effective, or pinpointing consistent patterns under particular conditions, would add depth.

W2. The paper did not propose any method or framework to address the observed limitations from their empirical experiments. This gives the work a preliminary feel, as it primarily presents observations without actionable solutions. Introducing even a preliminary solution could have increased the practical impact of the findings, bridging the gap from exploratory analysis to offering actionable guidance for pruning multilingual models.

How about a systematic approach to language selection for calibration or a metric-based framework to guide pruning objectives could significantly elevate the paper’s contributions? For instance, defining an optimal selection strategy for calibration languages or samples, tailored to different model objectives (such as maximizing language-specific vs. language-agnostic features), would provide some practical value. Alternatively, developing a metric to assess the "quality" or effectiveness of pruning based on targeted goals (like preserving language modeling capability versus general reasoning) would offer structured guidance. Such methods could help refine pruning approaches for Wanda/SparseGPT in multilingual models, providing a roadmap for choosing calibration data and pruning techniques that align with specific goals.

**Questions:**

Aside from the issues raised in the weaknesses, I noticed some typos, and the tables presented are challenging to interpret. Here are some suggestions:
- In Section 4.1, the method for reading the table is not immediately explained. I only found instructions on interpretation in Section 4.2 (first paragraph), where it describes comparing entries "column-wise" based on evaluation languages, which are the "column headers". It would be helpful to mention this earlier to improve readability.
- Since much of the analysis compares the best and worst performances, how about highlighting only the most relevant entries in the main paper tables? For example, if most analyses are focused only on the best entry in each column, highlighting only these would guide readers instead of reading all different colors. The heatmap presentation could be moved to the Appendix for readers who want broader information.
- In Figure 1, the labels "Fig (b)" and "Fig (c)" appear to be intended as "Fig (a)" and "Fig (b)," respectively. Also, explicitly indicating which part is Fig (a) and Fig (b) would enhance clarity.
- Similarly, when discussing Figures 2 and 3, how about using alphabetical identifiers (e.g., (a), (b), (c), etc.) as in Figure 1? Referring to parts of the figures by identifiers rather than phrases like "left-most," "bottom-right," or "left side" would reduce potential ambiguity.

Additionally, regarding multi-language calibration, how were the calibration samples allocated? Was the sample size $128 * \text{the number of languages}$? Also, what sampling strategy was used, and were other sampling strategies considered?

---

> ### Author Response · Authors · 2024-11-19
> **Response to Reviewer vgGp (Part 1)**
>
> We thank the reviewer for acknowledging our “comprehensive experiments” and “informative insights”, providing “insights not previously found in English-focused pruning studies” and we embrace the feedback and would like to address all the raised concerns as outlined below.
>
> **W1:** _Feedback on clarity and conclusiveness_
>
> We appreciate the reviewers' feedback on clarity and conclusiveness and actually, we are on the same page. We followed the common pruning practices that were expected to show us consistent and conclusive results but they did not.  Based on those **mixed results: pruning on target languages can effectively maintain low perplexity but not downstream task performances**. We further investigate the **WHY?**, and we **analyze the internal learned representations on three different levels: subspace level (section 5.1), matrix level (section 5.2), and neuron level (section 5.3)**. All three analyses lead us to conclude that **model pruning preserves the linguistic capabilities of LLMs but distorts certain non-linguistic capabilities, such as reasoning and knowledge retrieval which are essential for downstream tasks**.  We will carefully rephrase or remove the unclear conclusions highlighted by the reviewer accordingly.
>
> **W2:** _Proposal of a method or framework to address the observed limitations_
>
> While our focus is on providing detailed empirical results and analysis of internal learned representations of current pruning practices, we recognize the importance of more practical insights. Thus, as the reviewer suggested, we introduce a preliminary solution: **calibration using task-specific data**. We evaluate the effectiveness of this method and show the results below: this approach improves pruning performance on specific downstream tasks but not on others, compromising the open-domain and general-purpose nature of the LLM. We will include the full results in the final version.
>
> Llama 3 8B, pruned with SparseGPT, calibrated on MMLU, tested on MMLU
>
> | calib \ test |   AR       |   DE       |   EN       |   ES       |   FR       |   RU       |   ZH       |
> | ------------ | ---------- | ---------- | ---------- | ---------- | ---------- | ---------- | ---------- |
> | AR           | **33.7**   |   38.3     |   50.6     |   37.1     |   32.8     |   33.0     |   32.9     |
> | DE           |   31.2     | **41.6**   |   49.0     |   38.6     |   37.0     |   35.7     |   35.2     |
> | EN           |   27.9     |   34.7     |   55.2     |   40.0     |   38.5     |   28.9     |   30.5     |
> | ES           |   26.8     |   36.1     |   52.8     | **42.2**   |   36.3     |   32.5     |   31.5     |
> | FR           |   27.6     |   37.6     |   54.6     |   40.5     | **39.5**   |   30.9     |   30.3     |
> | RU           |   30.8     |   36.7     | **55.4**   |   34.2     |   35.0     | **35.8**   |   33.8     |
> | ZH           |   30.9     |   37.1     |   54.2     |   39.1     |   33.1     |   34.6     | **37.4**   |
>
>
> Llama 3 8B, pruned with SparseGPT, calibrated on MMLU, tested on Belebele
>
> | calib \ test |   AR       |   DE       |   EN       |   ES       |   RU       |   SW       |   ZH       |
> | ------------ | ---------- | ---------- | ---------- | ---------- | ---------- | ---------- | ---------- |
> | AR           | **56.8**   | **58.1**   |   72.4     | **57.3**   | **55.2**   |   30.1     |   52.6     |
> | DE           |   49.0     |   55.8     |   55.1     |   44.3     |   49.8     | **31.1**   | **62.7**   |
> | EN           |   40.3     |   52.0     |   72.4     |   54.9     |   50.3     |   25.8     |   54.6     |
> | ES           |   45.3     |   56.0     |   67.6     |   52.1     |   53.2     |   30.0     |   57.4     |
> | RU           |   45.8     |   49.7     | **74.9**   |   42.1     |   50.8     |   27.9     |   51.9     |
> | ZH           |   34.2     |   51.3     |   73.8     |   55.9     |   41.0     |   26.6     |   53.7     |

---

> > ### Author Response · Authors · 2024-11-19
> > **Response to Reviewer vgGp (Part 2)**
> >
> > **Calibration using Aya-Collection instruction-tuning dataset**: We further test calibration with question-answering prompts. This method does not resolve the irregularities in downstream task performance, **underscoring the challenges in selecting optimal calibration sets and the limitations of current state-of-the-art pruning techniques.**
> >
> > Llama 3 8B, pruned with SparseGPT, calibrated on Aya-Collection, tested on MMLU
> >
> > | calib \ test | AR       | DE       | EN       | ES       | FR       | RU       | ZH       |
> > |--------------|----------|----------|----------|----------|----------|----------|----------|
> > | AR           | 27.6     | 31.1     | 51.1     | 34.2     | 35.6     | 28.5     | 28.2     |
> > | DE           | 30.9     | 40.3     | 50.6     | 36.8     | **37.2** | **35.9** | 34.8     |
> > | EN           | 27.5     | 35.2     | **53.2** | 38.0     | 36.0     | 27.4     | 29.9     |
> > | ES           | 25.3     | 28.7     | 50.2     | 34.8     | 30.5     | 26.3     | 27.4     |
> > | FR           | **32.0** | **39.1** | 51.7     | **39.9** | 35.3     | 35.3     | **35.7** |
> > | RU           | 29.2     | 37.1     | 49.9     | 36.7     | 33.0     | 31.8     | 32.3     |
> > | ZH           | 24.3     | 28.0     | 48.4     | 31.9     | 29.2     | 26.6     | 26.9     |
> >
> > Aya-23 8B, pruned with SparseGPT, calibrated on Aya-Collection, tested on MMLU
> >
> > | calib \ test | AR       | DE       | EN       | ES       | FR       | RU       | ZH       |
> > |--------------|----------|----------|----------|----------|----------|----------|----------|
> > | AR           | **39.6** | 43.0     | 47.3     | 45.1     | 44.2     | 42.2     | 41.0     |
> > | DE           | 38.3     | **44.4** | 47.4     | 44.9     | 45.0     | 42.3     | 41.3     |
> > | EN           | 37.9     | 43.7     | **48.4** | 44.1     | 44.1     | 41.4     | 41.3     |
> > | ES           | 38.7     | 42.9     | 47.8     | **45.2** | 44.7     | 42.7     | 41.1     |
> > | FR           | 39.1     | 43.2     | 47.5     | 45.0     | 44.4     | 42.8     | 41.8     |
> > | RU           | 39.1     | 42.5     | 47.7     | 45.6     | **44.8** | **43.0** | 41.6     |
> > | ZH           | 37.9     | 43.4     | 45.0     | 43.1     | 43.8     | 42.2     | **41.8** |
> >
> > Aya-23 8B, pruned with SparseGPT, calibrated on Aya-Collection, tested on Belebele
> >
> > | calib \ test | AR       | DE       | EN       | ES       | RU       | FR       | ZH       |
> > |--------------|----------|----------|----------|----------|----------|----------|----------|
> > | AR           | 61.1     | 68.2     | 77.1     | 63.9     | 62.9     | 72.7     | 65.9     |
> > | DE           | 52.6     | 68.3     | 74.6     | 59.3     | 59.8     | 66.0     | 61.7     |
> > | EN           | 54.8     | **70.2** | 76.9     | 67.6     | 66.8     | 72.3     | 68.3     |
> > | ES           | 60.9     | 69.9     | **79.7** | **71.3** | **68.0** | **72.8** | 65.7     |
> > | FR           | 61.3     | 68.7     | 75.0     | 68.2     | 67.1     | 72.4     | 66.4     |
> > | RU           | **64.1** | 69.0     | 77.4     | 69.0     | 65.9     | 72.2     | 65.8     |
> > | ZH           | 57.4     | 69.1     | 74.8     | 64.7     | 64.4     | 69.7     | **68.9** |
> >
> > Additionally, we experimented with two new pruning methods that were not included in the main paper, as their results are negative and distracting from our central findings:
> >
> > 1. **Restoring weights of language-specific neurons post-pruning:** This approach led to performance degradation presumably due to not accounting for the distribution shifts in neuron activations, as shown in the appendix.
> >
> > 2. **Excluding the first and last transformer layers from pruning:** Contrary to expectations, this strategy also worsened performance, suggesting that reasoning and retrieval capabilities encoded in the intermediate layers are critical for maintaining task performance post-pruning.
> >
> >
> > Lastly, we will thoroughly review the paper to address any remaining typos and errors. For example, as suggested by the reviewer, we will consider moving some of the heatmaps to the appendix, relocate the explanation for interpreting the tables from Section 4.2 to Section 4.1, and shift the footnote on the allocation of calibration samples for multi-language calibration to the main text for improved visibility. We believe all rising issues will be well addressed in our revision.

---

> > > ### Comment · Reviewer_vgGp · 2024-11-23
> > >
> > > Thank you for the authors' thorough response to my comments. I appreciate the effort to address the concerns raised.
> > >
> > > [W1] Thank you for your efforts to improve the clarity of the takeaways.
> > >
> > > [W2] I appreciate the detailed explanation and the preliminary solution provided, along with the extensive empirical evaluations. However, I am still unsure regarding the intuition behind using task-specific data for calibration. Could you elaborate on how this approach addresses the observed reduction in "non-linguistic capabilities" due to pruning?
> > >
> > > If I understand correctly, in your first example with Llama 3 8B pruned using SparseGPT, calibration was conducted with MMLU as both the calibration and test dataset, as opposed to using mC4 for calibration. Based on the comparison with the table in the paper, it appears that calibrating with in-domain, task-specific data (MMLU) generally resulted in better accuracy than mC4 calibration. However, when calibration was performed using the Aya-collection, the performance was somewhat similar to using mC4 (I think a side-by-side table would be nice). This outcome seems intuitive, as in-domain calibration naturally aligns more closely with the specific downstream task. That said, I would appreciate it if you could further explore the connection between this finding and the concern about "non-linguistic capabilities." Specifically, do you observe any notable differences in the internal representations after calibration, particularly with task-specific versus general-purpose datasets? Providing insights into this connection would help clarify the broader implications of your findings. I may consider increasing my score if these additional insights are addressed.
> > >
> > > Lastly, I appreciate your effort in experimenting with different pruning methods, even when the results were negative.

---

> ### Author Response · Authors · 2024-11-25
> **New experiments as the reveiwer suggested (images are in supplementary material)**
>
> Thank you for your reply. Yes, your understanding is correct.
>
> In response to the suggestion to explore  “the connection between [in-domain and general-purpose calibration]” and “[observing] any notable differences in the internal representations after calibration [...]”, we conducted a more detailed analysis.
>
> Specifically, we add new experiments for analyzing the internal representations in terms of pruning mask (Section 5.2) and feature-level analysis (Section 5.3), with the focus on comparing between task-specific calibration (MMLU dataset) and general-purpose calibration (mC4).
>
> Our findings suggest, that **in-domain calibration indeed benefits internal representations for non-linguistic capabilities** as initially indicated by the previous evaluation tables. Visualizations of these results are provided as supplementary material, and we will include the full results (covering more models or pruning methods) in the final version. We hope these additions address the reviewer's concern and further clarify the implications of our work.
>
> **Pruning Mask Similarity Analysis**
>
> We analyze the pruning mask similarity for the LLaMA 3 8B model pruned with SparseGPT and calibrated on MMLU, Aya-Collection, or mC4. For this analysis, we directly intersected pruning masks calibrated with different datasets, omitting intra-dataset intersections due to limited reruns. Please see the following PDF files in the supplementary materials:
> - pruning_mask_similarity_aya_calib.pdf
> - pruning_mask_similarity_aya_calib.pdf
> - pruning_mask_similarity_mc4_calib.pdf
>
> Our results show that calibration on a narrower domain (MMLU) produces a higher contrast between low and high pruning mask similarity compared to open-domain calibration (mC4). This suggests that task-specific calibration better focuses the pruning process on weights most relevant to that domain, consistent with the improved evaluation results on MMLU.
>
> **Feature-Level Analysis with LSAR Decomposition**
> To investigate the impact of calibration datasets on layer output features, we employed LSAR to decompose MMLU prompts into language-specific and language-agnostic components for models calibrated on mC4 or MMLU.
>
> - **Language-specific features:** No significant differences were observed between models calibrated on mC4 and MMLU. (lsar_specific_mc4_calib.pdf, lsar_specific_mmlu_calib.pdf)
> - **Language-agnostic features:** Models calibrated on MMLU demonstrated reduced deviation from the full-sized model for matching calibration and test languages. This trend was particularly evident in the early and later layers. Conversely, models calibrated on mC4 exhibited more consistent pruning error across language-agnostic features, regardless of the test language. (lsar_agnostic_mc4_calib.pdf, lsar_agnostic_mmlu_calib.pdf)
>
> These results suggest that in-domain calibration helps retain language-independent capabilities, such as reasoning and knowledge retrieval, which are critical for downstream tasks.
>
> We sincerely thank the reviewer for the insightful suggestions, challenging us toward a deeper investigation of these issues. We look forward to including these expanded results in the final version and hope they address the remaining concerns.

---

> ### Comment · Reviewer_vgGp · 2024-11-25
>
> Thank you for the authors' very detailed responses. I think my initial main concerns have been addressed. While the solution may seem preliminary, I believe the insights provided here can shed light on future research in this area.
>
> I am also hoping that you would organize the contents of the paper better, especially now that you have additional experiments/clarifications (including from other reviewers). Nevertheless, I believe the authors will be committed on that and therefore I have increased my score.

---

### Official Review · Reviewer_t9N5 · 2024-11-04

**Soundness:** 3
**Presentation:** 3
**Contribution:** 2
**Rating:** 5
**Confidence:** 4

**Summary:**

This paper examines the calibration of pruning multilingual language models for use in specific languages, focusing on the impact of choosing a calibration language. While recent LLM pruning techniques achieve high compression without retraining, they often rely on English-based calibration, which may not suit multilingual models used in non-English contexts.

Through a comprehensive study, the authors compared different calibration languages across tasks, models, and pruning techniques, finding that calibration in the target language preserves language modeling abilities but doesn't consistently improve downstream task performance. Their analysis reveals that pruning generally maintains language-specific features but struggles with complex, language-agnostic aspects needed for knowledge and reasoning tasks. They suggest avoiding reliance on perplexity or English performance metrics for assessing pruned models' performance in other languages.

**Strengths:**

* This paper tackles essential issues related to the model pruning in multilingual settings.

* The motivation of this paper is clearly described.

* This paper revealed several significant findings, such as calibrating in the target language can efficiently retain the language modeling capability but does not necessarily benefit downstream tasks.

* This paper conducts a deeper analysis of the inner representations of pruning masks and individual neurons, revealing that while pruning generally preserves prominent language-specific features, it may struggle to retain language-specific neuron activation patterns and subtle, language-agnostic features related to knowledge and reasoning, both of which are essential for complex tasks.

**Weaknesses:**

* This paper identifies several issues related to the choices of calibration language that influence the downstream performance of pruned models. However, it lacks discussion or proposals for alternative methods to address these issues. While the paper makes several contributions indeed, it would be more comprehensive if it included potential solutions to the issues it highlights. Currently, I am not very confident that this paper is eligible to be published in a high-standard conference like ICLR, as it primarily presents observations from their experiments.

**Questions:**

* This is not a significant weakness, but it is interesting if findings in this paper also appear in slightly different model architectures, such as GPT-2 (an older model) and MoE models like Mixtral. Do the authors have any thoughts on this perspective? If so, could they offer any insights or evidence on whether these findings might generalize to a broader range of architectures?

* From my understanding, this paper employs a 50% pruning setting across all experiments. Is there a specific reason for choosing only this pruning ratio? If not, how might results and findings vary with a different pruning ratio? For instance, for larger models (e.g., 70B), many researchers may be very interested in exploring higher compression rates.

* In Section 4.1, it states, "There are a few exceptions. For instance, when evaluating Llama-3 pruned with Wanda on Chinese, Russian calibration performs best in terms of perplexity (23.6), slightly outperforming Chinese calibration (24.4)." Do the authors have any hypotheses about these observations? For instance, Wanda appears to yield poorer results in the target language compared to SparseGPT. What aspect of Wanda might be responsible for this degradation?



* In the discussion in Section 5.3, I find the statement, "This indicates that pruning introduces LAPE noise, shifting the LAPE score distribution and creating new language-specific (low LAPE) and agnostic (high LAPE) neurons," somewhat unclear. The results for 20% sparsity appear more diverse than those for 50% sparsity. If my interpretation is correct, the statement may be overstated or could misinterpret the observation. If I misunderstand something, please let me know.

**Details Of Ethics Concerns:**

No concerns

---

> ### Author Response · Authors · 2024-11-19
> **Reply to Reviewer t9N5 (Part 1)**
>
> We thank the reviewer for the thoughtful assessment and we are encouraged by the recognition of our "contributions indeed" "tackling essential issues’" and "deep analysis". We would like to address the concern raised as outlined below.
>
> **Weaknesses:**
>
> We would like to signpost the highlight of our paper, the deep **analysis conducted on three levels of internal learned representations (subspace, matrix, and neuron) to explore WHY** common calibration practices do not consistently deliver optimal downstream performance. This is beyond mainly presenting only observations. Our triple-level analysis also leads to the same conclusion: **pruning preserves linguist capability but compromises reasoning capability and knowledge retrieval.** This is the first study for such an underexplored and multifaceted problem. Our findings highlight several surprising limitations of current pruning techniques, even using common pruning practices and metrics. Notably, our results also reveal that existing conventional metrics (e.g., perplexity and SNR) fail to reliably indicate downstream task performance. These are all valuable takeaways. The reviewer suggests that "it would be more comprehensive if it included potential solutions". In our revision, we plan to make our discussions more comprehensive by discussing different tradeoffs. For example, we can additionally present results for calibrating on specific downstream task text, improving performance on that specific task but not on others. Consequently, this compromises the open-domain and general-purpose nature of the LLM. Such alternative solutions that consider different tradeoffs will be further discussed in our revision, addressing their limitations and tradeoffs. For example, the following two tables will be included:
>
> Llama 3 8B, pruned with SparseGPT, calibrated on MMLU, tested on MMLU
>
> | calib \ test |  AR       |  DE       |  EN       |  ES       |  FR       |  RU       |  ZH       |
> |--------------|-----------|-----------|-----------|-----------|-----------|-----------|-----------|
> | AR           | **33.7** |  38.3     |  50.6     |  37.1     |  32.8     |  33.0     |  32.9     |
> | DE           |  31.2     | **41.6** |  49.0     |  38.6     |  37.0     |  35.7     |  35.2     |
> | EN           |  27.9     |  34.7     |  55.2     |  40.0     |  38.5     |  28.9     |  30.5     |
> | ES           |  26.8     |  36.1     |  52.8     | **42.2** |  36.3     |  32.5     |  31.5     |
> | FR           |  27.6     |  37.6     |  54.6     |  40.5     | **39.5** |  30.9     |  30.3     |
> | RU           |  30.8     |  36.7     | **55.4** |  34.2     |  35.0     | **35.8** |  33.8     |
> | ZH           |  30.9     |  37.1     |  54.2     |  39.1     |  33.1     |  34.6     | **37.4** |
>
> Llama 3 8B, pruned with SparseGPT, calibrated on MMLU, tested on Belebele
>
> | calib \ test |   AR       |   DE       |   EN       |   ES       |   RU       |   SW       |   ZH       |
> | ------------ | ---------- | ---------- | ---------- | ---------- | ---------- | ---------- | ---------- |
> | AR           | **56.8**   | **58.1**   |   72.4     | **57.3**   | **55.2**   |   30.1     |   52.6     |
> | DE           |   49.0     |   55.8     |   55.1     |   44.3     |   49.8     | **31.1**   | **62.7**   |
> | EN           |   40.3     |   52.0     |   72.4     |   54.9     |   50.3     |   25.8     |   54.6     |
> | ES           |   45.3     |   56.0     |   67.6     |   52.1     |   53.2     |   30.0     |   57.4     |
> | RU           |   45.8     |   49.7     | **74.9**   |   42.1     |   50.8     |   27.9     |   51.9     |
> | ZH           |   34.2     |   51.3     |   73.8     |   55.9     |   41.0     |   26.6     |   53.7     |

---

> > ### Author Response · Authors · 2024-11-19
> > **Reply to Reviewer t9N5 (Part 2)**
> >
> > **Answers to Dedicated Questions**
> >
> > **Q1:** _Generalization to other architectures (e.g., GPT-2, Mixtral MoE models)_
> >
> > We appreciate the reviewer’s interest in broader generalization. However, we kindly ask for consideration of the experimental burden involved. Given the numerous configurations **one more model requires nearly 4,000 experiments** where one experiment covers at least hundreds of data (e.g., 2 pruning methods × (7+12) calibration language sets × 5 tasks × 7 test languages × 3 seeds). When factoring in the pruning process, this number increases further.
> >
> > In our current study, we include Llama 3 and Aya-23 models (and the new Llama 3.1 8B instruct in the appendix), which cover two different but mainstream architecture types. Llama 3 aligns closely with traditional Transformer models like GPT-2 (not specifically multilingual itself), while Aya-23 employs parallel attention and MLP layers, similar to PALM-2. Since the calibration set and pruning technique are independent of architecture we anticipate that our findings will extend to other multilingual LLMs. We are happy to provide selective results for a multilingual LLM that the reviewer is specifically interested in.
> >
> > **Q2:** _Choice of 50% pruning ratio_
> >
> > We selected a 50% sparsity level as it is commonly used in the literature, enabling easy and fair comparison with existing work. This ratio also strikes a balance, revealing differences between calibration languages while preserving sufficient downstream performance.
> >
> > In addition, we experimented with other sparsity levels: 20% sparsity for Llama 3 8B and Aya-23 8B and 80% sparsity for Llama 3 70B and Aya-23 35B. At low sparsity, the pruning effects are minimal, hence, the 20\% pruning may be of less practical motivation. Conversely, at high sparsity levels, the model’s performance collapses, diminishing the benefits of calibration, echoing previous studies [1]. We will include these results in the revised version for completeness.
> >
> > **Q3:** _Differences between Wanda and SparseGPT (Section 4.1)_
> >
> > We believe the observed PPL gap between Wanda and SparseGPT stems from their different approaches to pruning. Unlike SparseGPT, Wanda does not readjust model weights post-pruning, prioritizing computational efficiency over error minimization. This leads to greater quantization errors and, consequently, higher perplexity values.
> >
> > The observation of Llama-3 pruned with Wanda on Chinese performing worse than calibration on Russian in terms of perplexity might be a problem of fluctuating performance as also reported in [2, 3]. In this case, averaging PPL over more pruning runs on calibration sets with different samples could fix this deviation. Further, this could point towards a more intricate explanation involving linguistic features which is explored in Section 5.
> >
> > **Q4:** _Clarification on LAPE noise (Section 5.3)_
> >
> > "This indicates that pruning introduces LAPE noise, shifting the LAPE score distribution and creating new language-specific (low LAPE) and agnostic (high LAPE) neurons," The LAPE noise refers to the increased spread in the LAPE score distributions, evident from the length of the boxplots, particularly the leftmost ones (note that each box represents the same neurons across different sparsities). As the pruning ratio increases, the variance of the LAPE scores for low-entropy neurons expands. This suggests a greater shift in neuron activations, with a broader range of low LAPE scores post-pruning. Additionally, there is a general increase in language-specific neurons (low LAPE scores), which will become more apparent when the height of Figure 3 is enlarged. We will revise this section for clarity.
> >
> >
> > We hope these clarifications and the planned revisions address the reviewer’s concerns.
> >
> > [1] Compressing LLMs: The Truth is Rarely Pure and Never Simple, Jaiswal et al, ICLR 2024
> >
> > [2] On the Impact of Calibration Data in Post-training Quantization and Pruning, Williams et al, ACL 2024
> >
> > [3] Beware of Calibration Data for Pruning Large Language Models. Ji et al, [https://arxiv.org/pdf/2410.17711](https://arxiv.org/pdf/2410.17711)

---

> ### Comment · Reviewer_t9N5 · 2024-12-03
>
> Thank you to the authors for providing detailed responses. However, while the responses outline potential directions for future revisions, I feel they do not fully and directly address my concerns. Therefore, I will maintain my original score.
> (If I have misunderstood anything or if the authors have already updated the manuscript to address my concerns, please let me know.)

---

> > ### Author Response · Authors · 2024-12-04
> >
> > We believe we have addressed all concerns with new experiments (in reply and supplimentary materials) and further clarifications. We would like to reiterate the focus of our paper: explaining the observation that common pruning metrics fail to reliably predict downstream task performance when calibrating pruning for a specific target language. Our three-level mechanistic analysis on internal model representations reveals that **calibrating pruning on the target language effectively maintains language-modelling capabilities but not non-linguistic reasoning and factual retrieval capabilities**. While we believe our findings already fill an important gap in research, we acknowledge the value of exploring practical solutions.
> >
> > Motivated by the discussions with Reviewer vgGP and in line with the papers focus on current pruning practices, we have conducted additional experiments for task-specific pruning calibration. These experiments reveal a tradeoff between enhancing task-specific performance and retaining open-domain capabilities in pruned LLMs, further validating our central conclusions. The corresponding figures, demonstrating these insights, have been included in the supplementary materials uploaded before the November 27th deadline for PDF revisions. We will include them in the appendix for the camera-ready version upon acceptance.
> >
> > We respectfully note that the reviewers' concern reflects a broader challenge with current pruning methods. Our work sheds light on this limitation and its implications of current pruning methods, we believe that it is not a weakness of our work but rather an evidence and motivation for further research opportunities. We hope these efforts clarify and strengthen the contributions of our work.
> >
> > Thank you all again for your time and thoughtful feedback.

---

### Note · Authors · 2025-02-25

I have read and agree with the venue's withdrawal policy on behalf of myself and my co-authors.

---

### Meta-Review · Area_Chair_3mFE · 2024-12-13

**Metareview:**

This paper investigates pruning calibration for multilingual language models applied in monolingual contexts. It reveals that calibrating pruning in the target language helps preserve language modeling capabilities but does not consistently enhance performance on downstream tasks. The analysis indicates that while pruning retains language-specific features, it may fail to preserve language-agnostic features essential for knowledge representation and reasoning in complex tasks.

After reviewing the discussion during the rebuttal phase, I have decided to **reject** this paper for the following reasons:

- The authors did not adequately address the reviewers' questions.
- The analysis of results is somewhat superficial and lacks depth.
- The paper fails to propose any methods or frameworks to address the observed limitations identified in their empirical experiments.
- Despite the authors' assurances that these issues would be addressed in their revisions, the paper has not yet been updated. (Reviewer vgGp)

**Additional Comments On Reviewer Discussion:**

Reviewer t9N5 noted that the paper lacks innovative solutions to the issues it highlights, and the authors' response did not fully address the reviewer's concerns.

Similarly, Reviewer vgGp emphasized that the paper requires significant revisions, including the addition of further experiments. However, the authors failed to fulfill their commitment to implement the suggested updates.

Therefore, I am inclined to reject this paper.

---

### Decision · Program_Chairs · 2025-01-22

Reject